# Bio-climatic factors drive spectral vegetation changes in Greenland

Tiago Silva[1,2], Brandon Samuel Whitley[3], Elisabeth Machteld Biersma[3], Jakob Abermann[1,2], Katrine Raundrup[4], Natasha de Vere[3], Toke Thomas Høye[5,6], Verena Haring[7], and Wolfgang Schöner[1,2]

[1]Geography and Regional Science Institute, University of Graz, Graz, Austria
[2]Austrian Polar Research Institute, Vienna, Austria
[3]Natural History Museum of Denmark, University of Copenhagen, Copenhagen, Denmark
[4]Department of Environment and Minerals, Greenland Institute of Natural Resources, Nuuk, Greenland
[5]Department of Ecoscience, University of Aarhus, Aarhus, Denmark
[6]Arctic Research Centre, University of Aarhus, Aarhus, Denmark
[7]Institute of Biology, University of Graz, Graz, Austria

**Correspondence:** Tiago Silva (tiago.ferreira-da-silva@uni-graz.at)

**Abstract.**

The terrestrial ecosystem (ice-free area) in Greenland has undergone significant changes over the past decades, affecting biodiversity. Changes in near-surface air temperature and precipitation have modified the duration and conditions of snowpack during the cold season, altering ecosystem interactions and functioning. In our study, we statistically aggregated the Copernicus Arctic regional reanalysis (CARRA) and remotely sensed data on spectral vegetation, spanning from 1991 to 2023. We used principal component analysis (PCA) to examine key subsurface and above surface bio-climatic factors influencing ecological and phenological processes, both preceding and during the thermal growing season in tundra ecosystems. Subsequently, we interpreted spatio-temporal interactions of bio-climatic factors with vegetation and investigated bio-climatic changes dependent on latitude and topographical features in Greenland. Ultimately, we described regions of ongoing changes in vegetation distribution.

Our results indicate that, particularly in West Greenland, vegetation has responded highly to prevailing weather patterns of past decades. The PCA effectively clustered bio-climatic indicators that co-vary with summer vegetation, demonstrating the potential of CARRA for use in biogeographic studies. Among the factors studied, the duration of the thermal growing season (GrowDays) was pivotal across all ecoregions, increasing by up to 10 thermal growing season days per decade. These increases, interacting with other bio-climatic indicators, further promoted summer vegetation growth. The earlier onset of GrowDays was driven by warming (up to 1.5°C per decade), reduced winter precipitation, earlier snowmelt (on the order of 20 days per decade), and significant in snow depth. We report that regions with shallower snowpacks melt more slowly during the ablation period and are linked with a higher soil water content in the spring season. This relation not only coincides with the greenest regions in West and Southwest Greenland, but also with regions where green vegetation has recently emerged. These processes occur prior to the onset of GrowDays and are later combined with summer weather conditions that favour warmer temperatures and clear skies, resulting in significant summer greening. In general, vegetation expanded northward and towards the interior of Greenland. For instance, vegetation in Northeast Greenland has expanded by 22.5%, leading to newly vegetated areas compared to the 1991—2007 period.

While our statistical outcomes and interpretations derived from reanalysis and remote sensing data include uncertainties, they are corroborated by in situ studies conducted in the tundra region. Our study highlights the applicability of bio-climatic indicators from climate models as a foundational way to assess future changes in vegetation, while also demonstrating the need to include such indicators into permafrost dynamics schemes. If integrated, these bio-climatic indicators will improve our understanding of the atmosphere-vegetation-permafrost-carbon feedback loops across terrestrial Greenland with the changing climate, leading to better predictions of their responses.

## 1  Introduction

The changing climate in the past decades has had profound and rapid effects on Arctic ecosystems, with regional warming in Greenland at nearly three times the global average (Rantanen et al., 2022). This rapid warming is causing significant changes in the region's climate patterns and ecosystems. Jansen et al. (2020) highlight that the current era of abrupt climate change in the Arctic is unprecedented in the past several thousand years, leading to complex and varied responses in Arctic vegetation. Myers-Smith et al. (2020) reveal the intricacies of "Arctic greening", where increased temperatures and earlier snowmelt drive changes in plant growth and species distribution. These changes affect ecological interactions, such as shifts in plant community composition and alterations in soil nutrient cycling, leading to feedback mechanisms involving snow cover and surface albedo. Similarly, Huang et al. (2017) discuss the rate of change in vegetation productivity across northern high latitudes, reiterating that the response to climate change is influenced by multiple factors including soil moisture and temperature variations, as well as the timing and extent of snowmelt.

Over the last three decades, plant community height has increased across the Arctic (Bjorkman et al., 2018). This largely results from changes in plant species composition within communities, particularly due to an increase in abundance and productivity of deciduous shrub species, causing "shrubification" of the tundra (e.g., Mekonnen et al. 2021; Sturm et al. 2001). A large-scale study on the interconnection between temperature, moisture, and various key plant functional traits at 117 Arctic locations over 30 years of warming revealed a strong relationship between temperature and a range of plant traits, with soil moisture being a strong factor in determining the strength and direction of these relationships (Bjorkman et al. 2018). Availability of soil moisture is an important factor governing changes in plant communities (e.g., Ackerman et al. 2017; Gamm et al. 2018; Power et al. 2024). Studies on future scenarios of dynamic tundra vegetation have suggested that the spatial expansion of deciduous shrubs is favoured by warmer summers, whereas graminoids are more likely to increase in wetter conditions (van der Kolk et al., 2016).

Due to high latitude and continentality, soil moisture levels in summer are particularly important in certain areas of Greenland. In these drier areas, higher temperatures and less precipitation during the summer can potentially cause desiccation and salt accumulation at the soil surface, leading to a negative effect on plant growth (Zwolicki et al., 2020). Therefore, it is expected that an increase in temperature is unlikely to lead to a striking increase in plant growth, mainly due to the lack of precipitation. For instance, a study on the growth responses of two widespread and dominant deciduous shrub species (*Salix glauca* L. and *Betula nana* L.) in western Greenland revealed that both species have declined in growth since the 1990s, likely

due to increasing water limitation (Gamm et al., 2018). However, increased herbivory also plays a role, evident from increasing moth outbreaks (Post and Pedersen, 2008), growing muskox populations in both West (e.g., Cuyler et al. 2022; Eikelenboom et al. 2021) and East Greenland (Schmidt et al., 2015), as well as increases in geese populations in East Greenland (Boert-mann et al., 2015). Such studies are, however, based on small-scale analyses and contrast with observations of increasing shrub growth in other parts of the Arctic (Metcalfe et al., 2018). Also, certain inland parts of Greenland are warmer and drier than most other areas of the Arctic, and will therefore respond differently to climate change, making their spatial representativeness unclear. The critical influence of soil water availability on future changes in tundra plant communities in Greenland should not be underestimated and may also serve as an indicator for other drier Arctic regions, which may experience similar changes in temperature and precipitation. Additionally, while certain plant communities are generally better adapted to drier conditions, and have been observed to have increased with recent warming in the colder and drier High Arctic (e.g., Heijmans et al. 2022; Opała-Owczarek et al. 2018; Weijers et al. 2017), it is likely that certain species also become decreasingly temperature- and increasingly soil moisture-dependent during the summer (Weijers, 2022).

Temperature, precipitation, as well as soil water availability during the growing season are a few of the climatic indicators contributing to vegetation changes (Migała et al., 2014). However, other climatic indicators, such as snowfall, snowmelt rate, snowmelt timing, and frost also play an important role even before the onset of the growing season (Cooper, 2014). Increased temperatures during the cold season likely have a different impact on vegetation than temperature increases during the growing season (Weijers, 2022). Increased snow depth during the cold season usually causes increased plant growth in the following summer, as more snow provides insulation, less frost damage, and depending on the snowpack characteristics, increase in water availability (e.g., Lamichhane 2021; Migała et al. 2014; Wang et al. 2024). A relevant characteristic of the snowpack is that deep snow requires more energy than shallow snowpacks to equalise the cold content and liquid water holding capacity, an equalisation needed to subsequently initiate and sustain melt (Colbeck 1976; Musselman et al. 2017). As a result, deep snow often persists for extended periods, potentially delaying the start of the growing season and hindering plant growth (Schmidt et al., 2015). On the other hand, the insulation provided by deep snow has been demonstrated to promote increased microbial decomposition, enhancing the nutrient supply for the following growing season (e.g., Cooper 2014; Pedron et al. 2023; Xu et al. 2021). The greater energy input required to melt deep snow means that it melts later but more quickly, potentially causing nutrient loss due to increased runoff. Concurrently, meltwater from relatively shallow snow percolates the soil more efficiently during the ablation period, in contrast with fast snowmelt that quickly saturates the soil surface and runs off (Stephenson and Freeze, 1974). Slow snowmelt rates allow water to remain in the soil for extended periods, which is critical for activating soil microbe communities. These microbes then produce nutrients that are vital for vegetation growth (Glanville et al., 2012). However, if snow is limited and precipitation is falling as rain rather than snow, the resulting ice conditions can have damaging effects on the vegetation (increased branch mortality and vegetation damage, Weijers 2022) and on soil nutrient cycling. Additionally, in exceptional years like that of 2018, the High Arctic experienced unusually large amounts of snow, resulting in extraordinarily delayed snowmelt. This made it very difficult for plants to grow and for animals to access resources (Schmidt et al., 2019). Such conditions will strongly influence the growth of plants and have impacts throughout the food chain, such as for ruminants like the Svalbard reindeer (*Rangifer tarandus platyrhynchus*, Le Moullec et al. 2020) and caribou (*Rangifer*

*tarandus*) in West Greenland (Cuyler et al., 2022). Though the amount of snow and the coupling with temperature are important for plant growth and plant community composition in the Arctic, a Greenland-focused study assessing bio-climatic changes has not yet been conducted.

Grimes et al. (2024) recently showed that the doubling of vegetation cover across ice-free Greenland is linked with warming. The warming observed in Greenland over recent decades has been associated with increasingly frequent and intense weather patterns, promoting widespread clear-sky conditions and the advection of relatively warm air masses from southern latitudes along West Greenland (Barrett et al., 2020). Large-scale weather patterns can be inferred from climate oscillation indices by analysing specific atmospheric variables over time and space. For instance, the North Atlantic Oscillation is driven by surface

pressure configurations in the North Atlantic (Hurrell et al., 2003), and the Greenland Blocking Index measures geopotential height in the mid-troposphere over Greenland (Hanna et al., 2016). Both indices are commonly utilized in climate studies to deduce influences on various components of the climate system in Greenland and its vicinity (e.g., Bjørk et al. 2018; Olafsson and Rousta 2021). However, the way in which warming impacts other interlinked bio-climatic indicators through large-scale weather patterns requires further investigation.

In order to properly assess changes in bio-climatic indicators in Greenland, it is important to consider that soil water sources in the region are mainly from precipitation, snowmelt, and permafrost thaw. The combination of hard local geology with scouring by the ice sheet has shaped the landscape, resulting in thin soil cover leading to less prevalent thermokarst and low water retention properties (Anderson, 2020). Therefore, meltwater flows rather freely, eventually gathering in low-lying areas to form lakes or draining towards the sea. As a product of this, tundra vegetation often develops in regions adjacent to such water

bodies, eventually colonizing recently drained lakes and regions (e.g., Chen et al. 2023). Due to climate warming, not only runoff, but also subsurface runoff has increased in the Arctic (Rawlins and Karmalkar, 2024). Given the heterogeneity of soil properties and sources of water availability, subsurface bio-climatic indicators, such as volumetric soil water and, potentially, subsurface runoff, should be considered.

In this study, we analyse 32 years (1991–2023) of remotely-sensed Normalized Difference Vegetation Index (NDVI) data to

gain a deeper understanding of the spatio-temporal patterns of spectral vegetation changes across ice-free regions of Greenland, extending beyond the boundaries of point-scale studies. We examine the associations among bio-climatic indicators ranging from subsurface factors (such as soil water availability) to above-surface factors (such as the thermal growing season, heat stress, and frost) with summer spectral greenness. We also extend our study of bio-climatic changes beyond the summer by examining indicators from the preceding winter and spring seasons, assessing their combined interactions with summer spectral

greenness. Additionally, we individually explore historical trends of bio-climatic indicators and investigate their latitudinal and topographical sensitivity. Finally, we identify regions with significant temporal changes in summer spectral greenness and spatio-temporal changes in summer spectral greenness distribution.

## 2 Data

### 2.1 Copernicus Arctic regional reanalysis

The Copernicus Arctic regional reanalysis (CARRA) system predominantly relies on the non-hydrostatic numerical weather prediction model HARMONIE-AROME (Bengtsson et al., 2017), laterally forced by ERA5. CARRA, with a spatial resolution of 2.5 km, assimilates the same observational datasets as ERA5 (Hersbach et al., 2020), but is supplemented by additional station data from the national meteorological services within the CARRA domain. This study employed the CARRA-West domain, which encompasses Greenland. For the ice-free Greenland domain, the additional station data that CARRA assimilates

are sourced from the Danish Meteorological Institute and Asiaq-Greenland Survey networks. However, snow depth observations are not provided and are therefore not assimilated by CARRA. According to the CARRA Full System documentation (Schyberg et al., 2020), ice cover extent remains constant throughout the reanalysis period (1991–2023). The Leaf Area Index (LAI) climatology in CARRA is updated based on the multi-year mean values from the Moderate Resolution Imaging Spectroradiometer (MODIS) MCD15A2H C6 (Yang et al. 2006; Yuan et al. 2011), and these have been used to update the

ECOCLIMAP cover types for Greenland. ECOCLIMAP-I (Masson et al., 2003) is the global database utilized to initialize the Surface Externalisée (SURFEX, Masson et al. 2013), the soil–vegetation–atmosphere transfer scheme within CARRA. SURFEX is a multi-layer surface model that computes specific schemes dependent on the surface type (e.g., vegetation, soil, snow), allowing soil water phase changes and enabling runoff over frozen and unfrozen soil. This helps to better represent areas with permafrost and ice surfaces in Greenland, as they are not well described in the present version of HARMONIE-AROME.

Since CARRA does not represent permafrost, lacking inter-annual classifications of its types and extent, we used the term frozen surface instead of permafrost when discussing the CARRA output.

The snow and frozen soil parameterizations from the ISBA (Interactions between Soil, Biosphere, and Atmosphere) scheme, as described by Noilhan and Planton (1989), are implemented in the SURFEX. They have been tested in model intercomparison campaigns across northern Europe (e.g., Luo et al. 2003; Slater et al. 2001), high latitudes (Decharme and Douville, 2006),

and the Alpine regions (e.g., Decharme et al. 2016). The physical parameterizations within the ISBA have seen progressive developments over the past decades, particularly in its snowpack scheme, Crocus (Vionnet et al., 2012), which accounts for various snowpack features such as thickness, temperature, density, liquid water content, and grain types. Moreover, it incorporates important physio-geographical attributes like the surface slope. Crocus has been consistently coupled with global reanalysis like ERA5 (e.g., Ramos Buarque et al. 2025) and other atmospheric models (e.g., Luijting et al. 2018). When integrated with

the atmospheric model AROME, Crocus accurately reproduced the evolution of the snow surface temperature over Dome C (Antarctica) during an 11-day period (Brun et al., 2011), and it has effectively represented snowpack features in the French Alps (Vionnet et al., 2012) for more than a decade. Regarding surface and subsurface parameterizations, the ISBA scheme explicitly calculates the actual ice and water content in the soil to determine the heat capacity and thermal conductivity of the ground. The ground thermal conductivity depends on the surface and soil heat fluxes, which in turn are dependent on the soil scheme.

For soil schemes with vegetation, ISBA allows roots and organic matter to favour the development of macropores, which can lead to enhanced water movement near the soil surface (Masson et al., 2003). To our knowledge, data regarding the accuracy

for SURFEX schemes coupled with HARMONIE-AROME are not yet available. Soil properties in CARRA are derived from the Harmonized World Soil Database (Nachtergaele et al., 2010). The CryoClim project has generated a satellite-derived product of snow extent, which provides access to data collected on a daily basis from 1982 to 2015. CryoClim is a worldwide,

optical snow product that utilizes the historical Advanced Very High-Resolution Radiometer - Global Area Coverage (AVHRR GAC) data (Stengel et al. 2020). In the context of CARRA, the CryoClim data is ultimately used due to its comprehensive coverage for the entire period up to 2015. The data providers assure that the data for the period post-2015 have been produced and arranged in collaboration with the CryoClim developers at the Norwegian Meteorological Institute. Despite the fact that neither snow depth nor snow extent is assimilated, van der Schot et al. (2024) demonstrate in a recent study that the agreement

is strong between the snow water equivalent modelled by CARRA and a snow model utilizing in situ observations in both the West and East coastal regions of Greenland. They report that CARRA is capable of successfully representing snow-related indicators, with correlation coefficients exceeding 0.8 and mean absolute percentage errors less than 30%.

The derived precipitation from CARRA was taken from its underlying model forecast system and is not an assimilated product. Hence, to minimize the impact of the spin-up, we followed the CARRA Full System documentation (Schyberg et al.,

2020), which suggests combining 12 h accumulated precipitation by the difference of precipitation at lead times 18 and 6 h from forecasts initiated at 00 UTC and 12 UTC, respectively. This procedure was used for determining liquid precipitation (time integral of rain flux) and total solid precipitation (time integral of total solid precipitation flux).

## 2.2 NOAA Climate Data Record for Normalized Difference Vegetation Index

Phenology studies in remote sensing utilize data collected by satellite sensors, which determine the spectrum of light absorbed

and reflected by predominantly green vegetation. Specific pigments present in plant leaves exhibit a pronounced absorption of visible light wavelengths, particularly those in the red spectrum. Conversely, the leaves exhibit a strong reflection of near-infrared (NIR) light wavelengths. While numerous vegetation indices exist, one of the most prevalent is the Normalized Difference Vegetation Index (NDVI), which uses red and near-infrared bands. NDVI serves as a measure of spectral vegetation health and spans from -1 to 1. Biologically, NDVI values close to +1 suggest a high density of greenness and robust vegetation

health, while values near zero indicate barren land or surfaces with little to no vegetation, such as rocks or sand. Negative NDVI values are typically associated with water, clouds, or snow, with no spectrally visible vegetation.

The National Oceanic and Atmospheric Administration (NOAA) Climate Data Record (CDR) using the AVHRR (Vermote et al. 2018) NDVI, Version 5 (hereafter AVHRR NDVI) and NOAA CDR using the Visible Infrared Imager Radiometer Suite (VIIRS, Vermote et al. 2022) NDVI, Version 1 (hereafter VIIRS NDVI) are jointly used in this study from 1991 to

185 2023 on a daily basis with a grid resolution of 0.05 degrees (approx. 5.5 km in latitude and around 2.5 and 0.5 km between 60 and 85 degrees North, respectively). AVHRR NDVI is available until the end of 2013, and is thereafter continued by its successor VIIRS NDVI. The surface reflectance and the associated AVHRR and VIIRS NDVI take into consideration atmospheric corrections (e.g., total column of atmospheric water vapour, ozone, and aerosol optical thickness). According to AVHRR and VIIRS technical reports, the NIR channel is centred at different wavelengths (830 nm vs. 865 nm). As there

is no overlapping period available in the NOAA CDR, potential mismatches between AVHRR and VIIRS NDVI cannot be

discarded. However, AVHRR NDVI uses the MODIS Land-Sea mask, and its cloud mask is spectrally adjusted using 10 years of MODIS data, with a 90% match accuracy over land (Franch et al. 2017). As VIIRS will eventually replace MODIS for land science, MODIS is also used to calibrate VIIRS NDVI estimates (Skakun et al. 2018).

In addition to NDVI, both products provide quality control flags. While the AVHRR NDVI flags the entire domain for latitudes above 60 degrees as polar latitudes, the VIIRS NDVI implements more stringent quality control measures, effectively flagging clouds and snow cover at polar latitudes.

### 2.3 Climatic oscillation indices

A variety of analytic approaches, including principal component analysis (PCA) and $k$-means clustering, are often used to characterize the North Atlantic Oscillation (NAO), with input data sourced from reanalysis datasets or station records. The NAO
provided by the National Center for Atmospheric Research/University Corporation for Atmospheric Research (NCAR/UCAR) (Hurrell et al., 2003) is derived by applying PCA to sea-level pressure measurements, with the NAO index calculated using the leading principal component derived from sea-level pressure anomalies within the Atlantic domain (20°N–80°N, 90°W–40°E). This product is posited to provide a more comprehensive representation of spatial patterns of the NAO compared to indices based on specific terrestrial stations. However, it should be noted that the dynamic nature of PCA-based NAO indices is subject
to ongoing refinement with the integration of new data.

The Greenland Blocking Index (GBI) is derived from the 500 hPa geopotential height over the region (60°N–80°N, 80°W–20°W), retrieved from the NOAA Physical Sciences Laboratory/Earth System Research Laboratories (PSL/ESRL) (Hanna et al., 2016). Both the NAO and GBI indices originate from the NOAA National Centers for Environmental Prediction (NCEP/NCAR) reanalysis dataset (Kalnay et al. 1996). Consequently, these climatic oscillation indices have undergone sea-
210 sonal standardization against the baseline period of 1950–2000.

## 3 Methods

### 3.1 Spectral greenness

Arctic regions, characterized by sparse vegetation, typically exhibit markedly low NDVI values, often as low as 0.15 (e.g., Gandhi et al. 2015; Liu et al. 2024), with areas of dense shrubs above 0.5 (e.g., Walker et al. 2005), and signal saturation at
215 around 0.7 (e.g., Myers-Smith et al. 2020).

As estimates integrated through time are less likely to be influenced by temporal sampling artefacts at high latitudes than metrics based on maximum NDVI (e.g., Myers-Smith et al. 2020), we started by calculating monthly integrated NDVI. Since the focus is on green vegetation, only daily NDVI pixel values greater than or equal to 0.15 are considered. Then, we divide the monthly integrated NDVI by the total number of monthly observations (n, see Figure S1 for the interannual variability
of n) to obtain the monthly NDVI. However, before 2014 and as described in Subsection 2.2, the AVHRR algorithm was less strict in its data quality control compared to VIIRS from 2014 onward, resulting in higher n before 2014 that lowers

monthly NDVI. To address temporal heterogeneities, we adjusted n from the AVHRR period with the number of monthly observations acquired during the VIIRS period. From 2014 to 2023, we identified the minimum, maximum, and average number of observations for each month. Hence, using these three quantities, we generated a consistent variability range from 1991 to 2013 to recalculate monthly NDVI, considering a similar number of observations as from 2014 to 2023. This procedure assumes that the environmental conditions (i.e., snow-cover, clouds, and shadow) between 1991 to 2013 are similar to those between 2014 and 2023. Figures S2 to S5 show the average number of monthly observations and the associated standard deviation for the AVHRR and VIIRS periods, both before and after adjusting n.

Pixels exhibiting a monthly NDVI of 0.15 or greater are indicative of monthly greenness. The area derived from this monthly greenness is defined as the greenness extent. Additionally, we calculated the average summer greenness (see subsection 3.2 for the definition of seasons), which we will refer to as greenness hereafter. We also assessed spatio-temporal changes in the greenness extent between the periods of 2008–2023 and 1991–2007. We described these comparisons as changes in the greenness distribution, where an increase in greenness distribution is characterized as an expansion and a decrease as shrinkage. In addition, we analysed temporal changes in greenness (more details about trend analysis provided in subsection 3.4), wherein positive trends denote greening, and negative trends denote a reduction in greenness.

As the gridded products in this study have different spatial resolutions, the study interpolates the greenness from the NOAA NDVI to match the finer 2.5 km CARRA grid resolution.

## 3.2 Bio-climatic factors

The set of chosen bio-climatic indicators was inspired by previous work from Aalto et al. (2023) and Rantanen et al. (2023), who proposed and investigated bioclimatic indices in Finland and across the Arctic. Our study focusses on Greenland, and considers adapted thresholds and additional climatic factors.

As shown in Table 1, certain bio-climatic indicators are based on seasonal statistics using the definition of the meteorological seasons: winter spans from December to February (DJF), spring from March to May (MAM), summer from June to August (JJA) and autumn from September to November (SON), respectively. Greenness, 2-m air temperature ($T_{2m}$), RainRatio, the volumetric soil water (i.e. the volume concentration of liquid water in the soil) and ice (SoilWater and SoilIce), and vapour pressure deficit (VPd) are seasonally averaged, whereas precipitation, snowfall (Snow), and rainfall (Rain) are seasonally accumulated. The volume of cryospheric variables, like SoilIce and Snow, are in water equivalent (w.e.) correspondent to the volume that the liquid water would have if the ice melted.

**Table 1.** Brief description of the bio-climatic indicators derived in the study.

| Bio-climatic Indicator | Description | Units |
|---|---|---|
| $T_{2m}$ | seasonally averaged air temperature at the height of 2 m above the surface | °C |
| $SWE_{MAX}$ | annual maximum mass of liquid water from melting the snow per unit area | mm w.e. |
| $SWE_{MAX}DOY$ | day of the year for $SWE_{MAX}$ | day of the year |
| SnowDays | annual number of snow-covered days when SWE is higher than 10 mm w.e. | days |
| GrowDays | annual number of days with daily $T_{2m}$ higher than 1 °C that does not belong to SnowDays | days |
| DegreeDays | sum of daily $T_{2m}$ during GrowDays | K days |
| Onset | first day of GrowDays | day of the year |
| End | last day of GrowDays | day of the year |
| MeltRate | mean melt rate for ablation days between $SWE_{MAX}DOY$ and Onset of GrowDays | mm w.e. day$^{-1}$ |
| Greenness | seasonally averaged monthly NDVI, as described in Section 3.1 | unitless. |
| Snow | seasonally accumulated mass per unit area of snow and ice particles falling on the surface | mm w.e. |
| RainRatio | seasonally averaged fraction of liquid precipitation out of the total precipitation | % |
| RainOnSnow | number of days with RainRatio higher than 50% in SnowDays | days |
| Rain | seasonally accumulated mass per unit area of rain falling on the surface, when RainRatio≥50% | mm w.e. |
| VPd | seasonally averaged vapour pressure deficit as the difference between the amount of water vapour in the air and the amount of water vapour the air could hold when it is saturated | hPa |
| SoilIce | seasonally averaged water equivalent of volumetric soil ice content | % |
| SoilWater | seasonally averaged volumetric liquid water in the soil | % |
| FrostDays | number of days when SWE is less than 10 mm w.e. in spring with negative daily $T_{2m}$ | days |
| DroughtDays | number of days with precipitation lower than 1 mm w.e. lasting for more than 10 consecutive days | days |
| HeatDays | number of days exceeding the seasonal $T_{2m}$ climatology for the period 1991–2023 by 2 SD | days |
| Longitude | distance east or west of the Greenwich meridian | degrees |
| Latitude | distance north of the equator | degrees |
| Elevation | vertical elevation above sea level | m a.s.l. |
| Surface slope | the inclination of the surface | degrees |
| Surface aspect | the slope direction | degrees |

Surface slope is transformed into sine-aspect (west-east orientation) and cosine-aspect (north-south orientation), given its circular orientation. Positive values in sine-aspect (cosine-aspect) indicate the degree to which a slope faces east (north), whereas negative values indicate the degree to which it faces west (south).

From the CARRA daily-averaged snow water equivalent (SWE), we derived the maximum SWE ($SWE_{MAX}$), the day of the year $SWE_{MAX}$ occurs ($SWE_{MAX}$ DOY), and snow-covered days (SnowDays) when SWE > 10 mm of water equivalent (w.e.). It is important to note that SnowDays are not necessarily continuous, with sporadic snow events occurring along the hydrological year (1st October to 30th September of the following year). van der Schot et al. (2024) provide a thorough validation of CARRA SWE with in situ observations across Greenland. They report that CARRA is capable of successfully representing snow-related indicators such $SWE_{MAX}$ and $SWE_{MAX}$ DOY.

We calculated GrowDays by considering days that do not belong to SnowDays, using daily-averaged 2 m air temperature ($T_{2m}$) > 1 °C. The onset (i.e., the first) and termination (i.e., the last) day of the year of GrowDays are also derived. The indicator DegreeDays is obtained by summing up $T_{2m}$ during the previously defined GrowDays. The daily rain ratio (RainRatio) is defined as the fraction of liquid precipitation out of the total precipitation. SnowDays, in combination with RainRatio higher than 50%, are used to derive days with rain-on-snow (RainOnSnow) between January and July to investigate potential snowpack warming before the thermal growing season onset. $SWE_{MAX}$ DOY and thermal growing season onset are used to determine the length of the snow melting period. During the snow melting period, we calculated daily changes of SWE from which we derived days with negative SWE changes (SWEmeltDays) and the mean of the negative SWE changes (MeltRate).

The vapour pressure deficit in summer (VPdJJA), which is the difference between the water vapour pressure of saturated air and the actual water vapour pressure in the air, was calculated to represent continentality. Continentality in summer is expressed by high temperatures and lower humidity due to the large distances from moisture sources. This lack of moisture availability contributes to lower water vapour pressure, which, when combined with high temperatures, leads to higher VPd. A high VPdJJA indicates a strong drying potential in the atmosphere, which can significantly influence evaporation rates and plant water stress (e.g., Grossiord et al. 2020; Yuan et al. 2019).

DroughtDays, the number of days with precipitation lower than 1 mm w.e. lasting for more than 10 consecutive days, is seasonally aggregated in spring and summer. HeatDays are also seasonally aggregated in spring and summer, and they consist of the number of days exceeding the seasonal $T_{2m}$ climatology for the period 1991–2023 by two standard deviations (2SD). As the 32-year period is fairly normally distributed, +2SD are approximately equivalent to the 97.5[th] percentile. FrostDays in spring are derived from days without snow cover, jointly with negative $T_{2m}$ days.

Spectral greenness was compiled for the summer, in order to capture the period with maximum solar radiation in Greenland, thereby avoiding snow-covered patches. Given the fact that shadow areas heavily impact reflectance, latitudes higher than 75°N are not considered due to low sun elevation. We restricted our study area to West and Northeast Greenland, as steep mountains, deep fjords, expansive glaciers, and extensive ice caps inhibit the method's applicability in Southeast Greenland.

### 3.3 Ecoregions

Greenland extends for approximately 23 degrees of latitude, with temperature and precipitation rates varying considerably across latitudes and coasts (Westergaard-Nielsen et al., 2020). Due to the semi-permanent Icelandic Low and the steep topography, the Southeast coast receives more precipitation than the Southwest coast (e.g., Ettema et al. 2010; Fettweis et al. 2017). In general, the West and East coasts exhibit different topographic features, from a topographically complex East contrasting

with predominantly glacially eroded regions in the West (e.g., Karami et al. 2017; Anderson 2020). Nevertheless, both coasts comprise diverse fjord systems that often channel the wind and shield inland areas against storms. Consequently, the north-facing slopes and the leeward side of these inland mountain systems receive reduced precipitation. Such coast-inland gradients are therefore complex, also influencing the distribution of permafrost and freshwater systems (e.g., Westergaard-Nielsen et al. 2018; Abermann et al. 2019). Both precipitation and temperature tend to decrease with latitude. Other factors known to shape the coastal climate are prominent ocean currents (e.g., East Greenland and North Atlantic current) as well as sea ice and fjord ice conditions (e.g., Westergaard-Nielsen et al. 2020; Shahi et al. 2023).

The Arctic tundra ecosystem, including Greenland, is typically separated into Low Arctic and High Arctic at around 70°N based on climatic and vegetation differences (Bliss et al., 1973). Greenland has also been mapped according to hydrology, soil pH, percentage of water cover, floristic provinces, and bio-climatic subzones (e.g., Walker et al. 2005). The former mapping partly relies on mean July temperature thresholds and positive degree monthly temperatures to classify subzones. However, the $T_{2m}$ JJA has warmed at a median rate of approx. 1°C per decade since 1991 (Fig. S6), likely shaping plant community structure and distribution. Eythorsson et al. (2019) also showed that the Köppen-Geiger climate classification and snow cover frequency in the Arctic have changed and will continue to change in the Arctic this century. To avoid climate-sensitive metrics, we split ice-free Greenland into five ecoregions (Fig. 1) based on physio-geographic features, such as adjacent seas, ocean currents, and ice caps, with direct and indirect control on heat and moisture transport.

In our study, we have defined Ecoregion 1 as the narrow coast along Baffin Bay in Northwest Greenland, including Sigguup Nunaa, Uummannaq fjord, Nuussuaq Peninsula, and Disko Island. Disko Island is known as the transition region between the High Arctic and theLow Arctic, with a smooth transition of High Arctic to Low Arctic vegetation types in between ecoregions 1 and 2. Ecoregion 2 stretches from Ilulissat to the Maniitsoq Ice Cap. This ice-free part is particularly widely stretched from West to East, with climates ranging from maritime at the coast to continental in the dry interior. Ecoregion 3 encloses mainly Southwest Greenland along the Labrador Sea to Nunarsuit, curving from the Labrador Sea to the North Atlantic. Ecoregion 4 comprises the mountainous and southernmost end of Greenland, facing the North Atlantic. Southeast Greenland, a very narrow coast composed of steep slopes, is the meeting point between the relatively cold East Greenland current and the relatively warm Irminger Current, leading to very foggy conditions during the warm season (e.g., Gilson et al. 2024; Laird et al. 2024). The combination of this region's complex topography with frequent cloud cover resulted in its exclusion from the analysis. Finally, ecoregion 5 spans from Kangertittivaq (Scoresby Sound) to the North coast of Young Sound, including Daneborg and Zackenberg. The coast of ecoregion 5 is also commonly affected by fog conditions. However, the coastal topography usually shelters inland regions. The Stauning Alps, a large system of mountain ranges west of Kangertittivaq, are excluded due to their very rugged and complex topography, with numerous rocky peaks and active glaciers in most valleys and only minor vegetation growth.

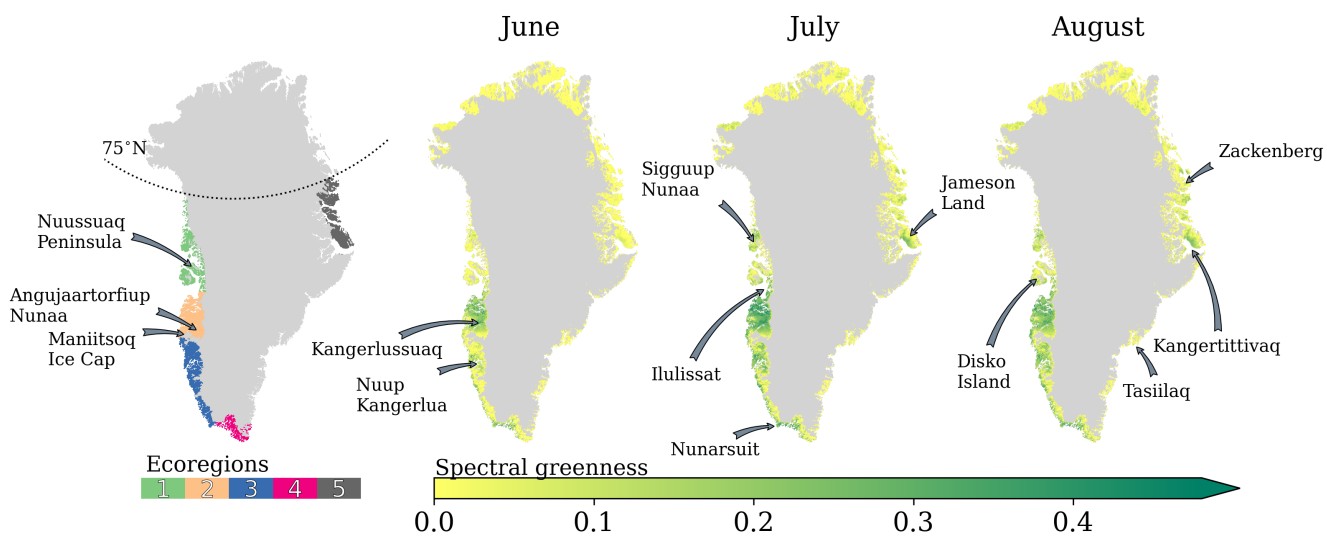

**Figure 1.** Ecoregions in ice-free Greenland, overlaid with June, July, and August averaged spectral greenness for the period 1991–2023. Place names referenced in the study are indicated.

The onset of the thermal growing season is inherently linked with distance to the coast, local elevation, and specific latitude. While the distance to the coast and elevation influence precipitation and snow depth, latitude governs the duration of sunlight and near-surface air temperature. Due to less elevated and less topographically complex terrain, the thermal growing season starts earlier in the West (on average [5th percentile, 95th percentile], DOY: 140 [103, 172]) than at the East Coast (DOY: 171 [145, 204]). Moreover, both latitude and elevation are crucial in cooling the atmosphere, allowing snowfall to occur, which in turn marks the end of the thermal growing season. Ecoregion 4, with the most GrowDays (140 [76, 198] days), is followed by ecoregions 2 (119 [75, 145]) and 3 (121 [77, 164] days). Ecoregion 1 (97 [56, 132]) and 5 (78 [47, 108]) typically have less than 100 GrowDays. Due to the proximity to the Atlantic cyclone track, ecoregion 4 receives the most precipitation, accumulating to a $SWE_{MAX}$ of 432 [114, 984] mm w.e. In contrast, ecoregion 2 receives about 25% of the precipitation received by ecoregion 4, with a $SWE_{MAX}$ of 143 [64, 325] mm w.e. This pattern arises because the interior of ecoregion 2 is surrounded by high peaks in the south, such as the Maniitsoq Ice Cap, which serve as a barrier to poleward moisture transport.

In Figure 1 we also show the 32-year monthly averaged greenness for summer months. As mentioned in subsection 3.1, the typical NDVI analysis that consists of averaging either the entire NDVI range or selecting the maximum NDVI is more prone to artifacts. Therefore, the 32-year monthly averaged greenness shown here is not necessarily based on 32 values in every pixel. This is reflected by the monthly averaged greenness over 32 years being lower than 0.15 in many regions. While the 32-year monthly averaged greenness spatial variability can be assessed with Figure 1, direct quantification of greenness saturation should be taken with care, given the interannual variability in greenness. Maps with the correlation coefficients between greenness and NAO index and the GBI between 1991 and 2023 are shown in Figure S7.

## 3.4 Statistical methods

Principal Component Analysis (PCA, Pearson 1901; Lorenz 1956), often used on remotely sensed and environmental data (e.g., Mills et al. 2013; Yan and Tinker 2006), was employed to investigate the combined interactions among bio-climatic indicators with summer greenness. The PCA (Pedregosa et al., 2011) solver was selected based on the input data shape. As the number of features in the input data are much less than the number of samples (geographic pixels), a classical eigenvalue decomposition on the covariance matrix was run. The classic PCA approach operates upon several assumptions, including I. linearity, which assumes that the relationships between variables can be adequately described by linear transformations; II. that there are no significant outliers in the data; and III. that there is homoscedasticity, meaning that variables have equal variance. In order to overcome heteroscedasticity, we standardized all variables for each ecoregion, centring the distribution around 0 and scaling it to a standard deviation of 1. We used quartiles and the interquartile range (IQR) to filter out values beyond the upper ($Q_3 + 3 \times IQR$) and lower outer ($Q_1 - 3 \times IQR$) fence, with $Q_1$ and $Q_3$ as first and third quartile, respectively. Finally, we ran a PCA for a set of bio-climatic indicators in every ecoregion between 1991 and 2023 until at least 90% of the cumulative explained variance was reached, omitting components contributing to minimal explained variance in order to accelerate the computation process.

As the classic PCA requires the variables to be linearly related, we calculated Pearson correlation coefficients to investigate bio-climatic indicators by ecoregion. However, Pearson correlation assumes that the data are stationary; that is, their statistical properties do not change over time. In order to avoid serial autocorrelation, we transformed the data into non-stationary time series by linearly detrending the data before performing correlation. The calculated correlations are displayed in a correlation matrix, and bio-climatic indicators with similar correlations are sorted with hierarchical clustering. This helped to visually discern bio-climatic indicators with comparable statistical relationships and supported the empirical reduction of indicators accounting for the relevant physical and ecological processes influencing the tundra ecosystems later used as part of the PCA. Certain bioclimatic indicators exhibited high correlations to one another, primarily due to physical reasons. Other bioclimatic indicators corresponded to complementary quantities. Consequently, the selection of bioclimatic indicators for the PCA was made on an arbitrary basis, further detailed in subsection 4.1. This aimed to diminish "noise", redundancy, and ultimately boost the clarity of interactions across the atmosphere-biosphere-cryosphere.

Due to a change of satellite sensor from 2014 onwards, we also investigated how the PCA performs interannually and whether there was a statistically significant change in the explained variance for years before and after 2014. The result is shown in Figure S8 for a set of 16 bio-climatic indicators, displaying that the two independent samples of explained variance have identical averages in all ecoregions, with a 95% confidence level, as determined by a two-sample t-test. Additionally, we performed correlation and trend analysis in three periods: AVHRR (1991–2013), VIIRS (2014–2023), and the full period (1991–2023) between greenness and climate oscillations to assess their statistical strength and direction as dependent on the sensor period.

We attempted a careful causal interpretation of the loading vectors from the first two principal components (PCs) of the PCA through biplots (Gabriel, 1971). Although these PCs account for most of the explained variance, their interpretation in

terms of causality is limited by the nature of PCA as a descriptive statistical technique. For a cautious interpretation of the PCs, we examined not only the magnitude and direction of the loading vectors but also trend maps of the involved bio-climatic indicators and relevant literature on experimental studies.

We attempted to use the preceding autumn bio-climatic indicators to understand whether the start of the snow period could have played a role in the following growing season. However, the explained variance in PCA changed little (decreases of approx. 2-3% per ecoregion) and the relative importance of all loadings remained similar. Additionally, we correlated the interannual explained variance of the first two principal components with averaged climate oscillations (NAO and GBI) during the warm season (from March to September), spring, and summer. We observed that the year-to-year variability in explained variance does not significantly correlate with seasonal climate oscillations. In other words, a particular NAO or GBI phase does not boost the explained variance of the first PCs, maintaining similar values interannually.

We used the non-parametric Mann-Kendall (M-K) trend test (Hussain and Mahmud, 2019) to assess trend monotonicity and significance among bio-climatic indicators. However, to acknowledge autocorrelation in the greenness data, we computed the Hamed and Rao modified M-K test (Hamed and Rao, 1998), with a variance correction approach considering all significant lags to improve trend analysis. The trend magnitude retrieved over decadal timescales corresponds to the Theil-Sen (T-S) estimator, a robust regression method that does not require the data to be normally distributed, making it less vulnerable to outliers than conventional methods. Under the null hypothesis that the slope is equal to zero, trends exhibiting confidence levels higher than 95% are highlighted and treated as significant trends.

## 4  Results

The greenness extent progressed at different rates across summer months and ecoregions (Fig. 2). Given the colder temperatures in ecoregions 1 (first column) and 5 (last column), considerable greenness extent was generally not evident until July, contrasting with the more southern ecoregions. However, in recent years, greenness extent has started to increase already in June, particularly noticeable for 2019. In the southern ecoregions, the thermal growing season onset (Onset) is much earlier, with greenness extent occurring already in the spring months. This is especially pronounced in ecoregion 2, which typically has a shallow snow cover (see subsection 3.3 for more details). By June, greenness becomes quite developed, reaching its peak in July. In the years 2015 and 2016, ecoregion 2 exhibited the largest extent of greenness, covering approximately 80% of its area.

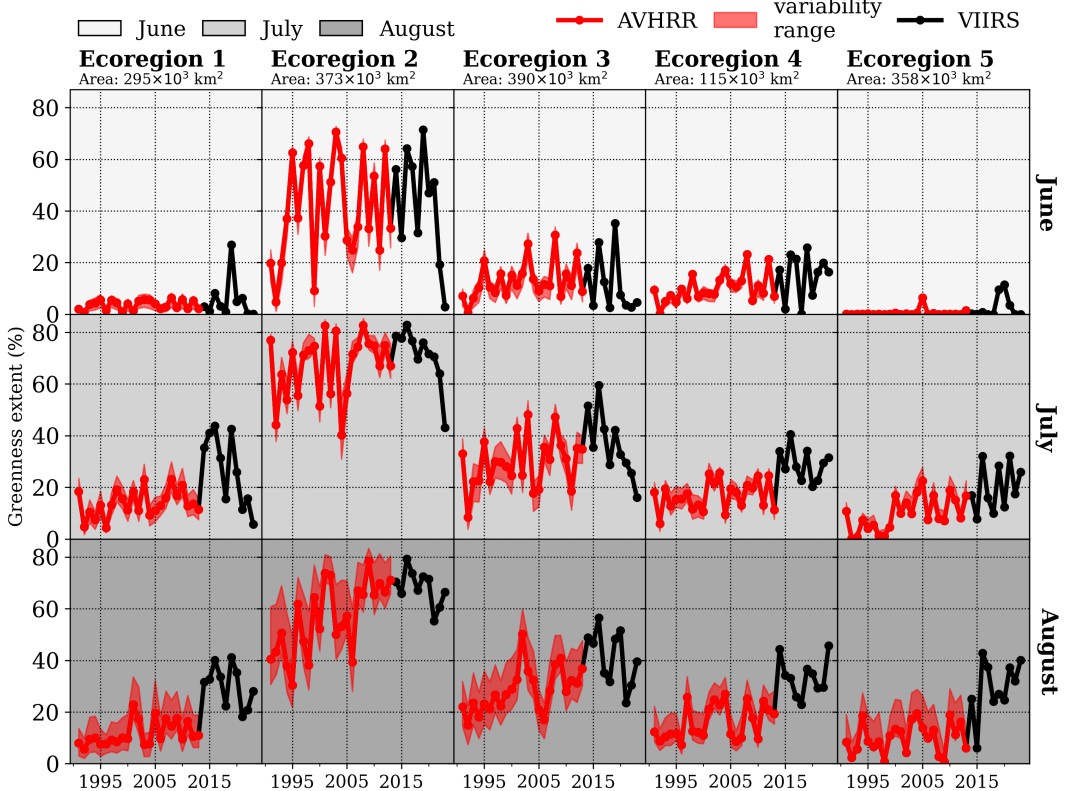

**Figure 2.** Development of greenness extent between 1991 and 2023 for June (upper row), July (middle row), and August (bottom row) across ecoregions based on AVHRR (red) and VIIRS (black). The AVHRR variability range is shaded in red, given the monthly minimum and maximum number of observations within VIIRS period (2014–2023).

It should be noted that prevailing weather patterns during summer months, like the North Atlantic Oscillation (NAO) and the Greenland Blocking Index (GBI), are highly correlated with greenness (Fig. S7). Therefore, summer weather patterns can accelerate or delay the maximum greenness extent given their link with temperature and precipitation. Correlations between greenness extent and summer GBI are investigated for three periods: AVHRR (1991–2013), VIIRS (2014–2023), and the full period (1991–2023), and are shown in Table S1. Positive and significant correlation coefficients ranging between 0.5 and 0.8 are found between ecoregions 1 and 4, generally with higher correlations for the VIIRS than for the AVHRR period. Greenness extent in ecoregion 5 is poorly correlated with the prevailing weather patterns during summer.

While the AVHRR 22-year trend showed increases in greenness extent, the VIIRS 9-year trend indicated a decrease, particularly in West Greenland (Table S2). However, due to high variability and small sample size, most trends in both periods are not significant. Significant and positive long-term trends range from 2% per decade in ecoregion 1 to approximately 6% per decade in ecoregion 4.

## 4.1 Interconnectedness among bio-climatic indicators

The detrended Pearson correlation coefficients for ecoregion 2 are shown in Figure 3. While the magnitude of the correlation coefficients varies across ecoregions, their direction is generally consistent.

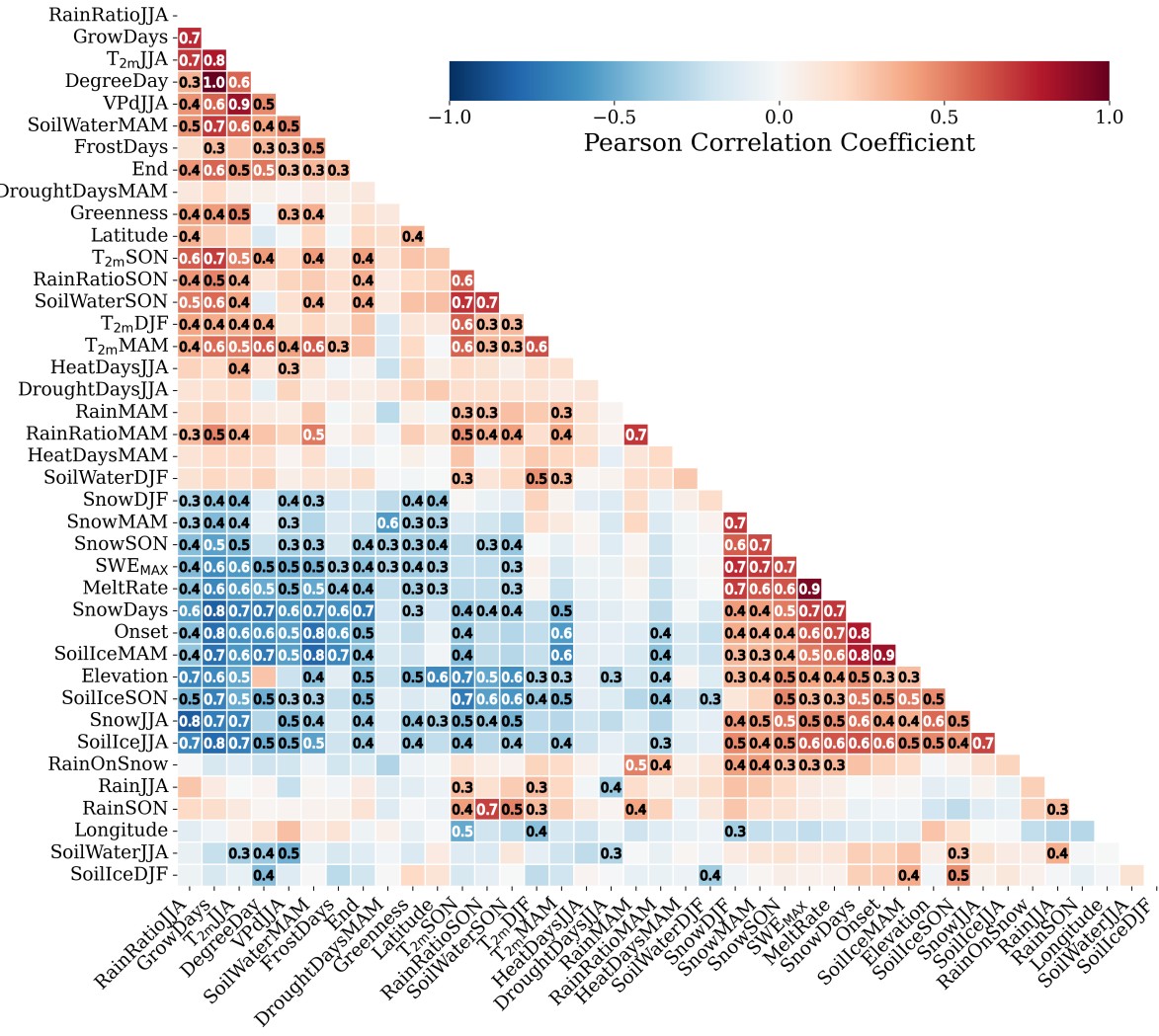

**Figure 3.** Correlation matrix for the bio-climatic indicators in ecoregion 2, including Elevation, Longitude, and Latitude. The correlation coefficient is colour-coded, and the absolute value is noted for absolute correlation coefficients higher than 0.3. The abbreviations of the bio-climatic indicators are described in Section 3.2 and in Table 1.

We investigated the correlations among all the bio-climatic indicators, including physical features like elevation, latitude, and longitude. These physical features relate to climate attributes across ecoregions. For instance, the higher the elevation and

latitude, the lower the precipitation rates. Note that these physical features are constant through time and were not considered when investigating the combined associations among bio-climatic indicators with greenness in the PCA.

A few bio-climatic indicators, such as DroughtDaysJJA, HeatDaysMAM, HeatDaysJJA, generally show correlations lower than 0.3, with the exception of DroughtDaysMAM, which has a negative correlation with SnowMAM and with $SWE_{MAX}$.
HeatDaysJJA correlates with $T_{2m}$JJA, indicating that higher near-surface air temperatures in summer are associated with the increased number of summer heat days. We excluded drought and heat indicators from the subsequent analysis, as greenness correlates more strongly with seasonal temperatures and precipitation amounts. Furthermore, drought and heat variability are respectively explained by seasonal temperature and precipitation. However, it is important to highlight that increases in Heat and DroughtDays during spring and summer reflect elevated near-surface air temperatures. RainOnSnow seems to be statistically linked with the increase of the RainMAM and RainRatioMAM, but also with SnowMAM, as RainRatioMAM represents a mix of rain and snow. In our trend analysis, it is noticeable that RainOnSnow is increasing along East Greenland, and FrostDays are locally increasing in West Greenland. However, summer greenness is not statistically related to these two bio-climatic indicators. Therefore, we removed them from further analysis. Interestingly, FrostDays is positively correlated with spring near-surface air temperature. The increase in FrostDays is also correlated with the early disappearance of the snow cover, partly related to shallower snowpacks and highly correlated with the decreasing volume of ice in the soil in spring (SoilIceMAM). SoilIce is largely negatively correlated with the volume of water in the soil (SoilWater). Therefore, we decided to arbitrarily use SoilIce in winter (SoilIceDJF) and summer (SoilIceJJA) and SoilWater in spring (SoilWaterMAM) and autumn (SoilWaterSON) in further analyses. Additionally, SnowDays and DegreeDays are not used since both are highly explained by GrowDays. While DegreeDays accumulate $T_{2m}$ during GrowDays, SnowDays complement FrostDays and GrowDays as together they represent snow-free occurrences when daily $T_{2m}$ is negative and higher than 1°C, respectively. Strong correlations between Rain and RainRatio are found in spring and autumn, but not in summer. Consequently, we will retain both Rain and RainRatio variables exclusively for the summer. Finally, MeltRate is removed as it is physically explained by the snowpack depth.

## 4.2 Bio-climatic indicators interlinked with greenness

As described in the subsections 3.4 and 4.1, 16 bio-climatic indicators were chosen to account for relevant physical and ecological processes on the tundra ecosystem. Figure 4 displays the combined influence of the 16 bio-climatic indicators based on the first two principal components across ecoregions. These two principal components account for most of the variability, ranging from 52% in ecoregion 2 to 65% in ecoregion 4 (Fig. S9). It requires six to seven principal components to account for an additional 30 to 40% of the explained variance.

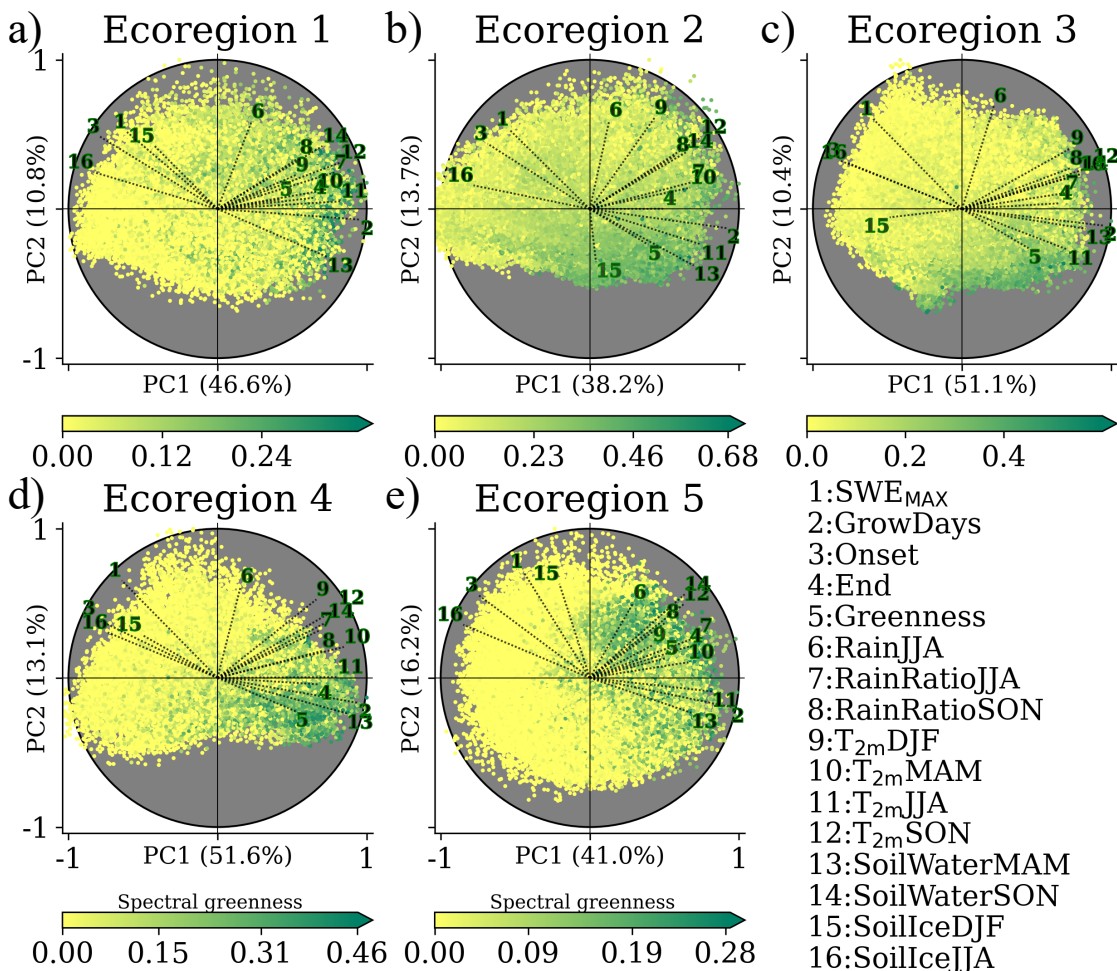

**Figure 4.** Biplot for scores between 1991 and 2023 for each ecoregion. The loading vectors are labelled and scaled by the maximum of each principal component. The scores are colour-coded based on the summer greenness, with different scales to enhance greenness. The explained variance of the first (PC1) and second (PC2) components is labelled in the corresponding axis of the subplot. The 16 bio-climatic indicators are 1: maximum snow water equivalent ($SWE_{MAX}$); 2: total number of thermal growing days (GrowDays); 3 and 4: start (Onset) and termination (End) of GrowDays; 5: summer greenness (Greenness); 6: rain in summer (RainJJA); 7 and 8: averaged rain ratio in summer (RainRatioJJA) and autumn (RainRatioSON); 9, 10, 11, 12: averaged 2-m air temperature in winter ($T_{2m}DJF$), spring ($T_{2m}MAM$), summer ($T_{2m}JJA$) and autumn ($T_{2m}SON$) 13 and 14: volumetric soil water in spring and (SoilWaterMAM) autumn (SoilWaterSON); 15 and 16: volumetric soil ice in winter and (SoilIceDJF) summer (SoilIceJJA). The abbreviations of the bio-climatic indicators are described in Section 3.2 and in Table 1. The spatial pattern of the averaged 1991–2023 scores for both components in every ecoregion, including their corresponding loadings, is shown in Fig. S10-S14.

According to the spatial maps of the first (PC1) and second component (PC2, Fig. S10-S14), PC1 is found to be highly controlled by the topography of the ecoregion, and is consequently related to temperature (and through that on elevation),

making GrowDays the bio-climatic indicator with the highest loading in all ecoregions, and therefore, the most significant contributor to the pattern represented by PC1. Through the analysis of the trend map for RainJJA (Fig. S15) and the spatial maps of PC2, we found that PC2 relates to precipitation and snow patterns, with $SWE_{MAX}$ and RainJJA having the highest explanatory power.

These two principal components together largely capture Greenness distribution, as seen by scores with high summer greenness often clustered in one specific quadrant of the biplot. Given that GrowDays has the most significant loading, demonstrated by the longest vector across ecoregions, and its minor loading on PC2, it evidences little dependence on precipitation and snow patterns.

In ecoregion 2, the ecoregion with the widest East-West coverage, summer greenness is suggested to depend considerably on the snowpack of the preceding cold season ($SWE_{MAX}$ loading vector opposite to Greenness loading vector). The decreasing trend of seasonal accumulated snow (SnowDJF, Fig. S16 and SnowMAM, Fig. S17) has led to $SWE_{MAX}$DOY to occur earlier (Fig. S18), which resulted in lower melting rates of the snowpack (MeltRate) as shown in Fig. S19. These shallow snowpacks are statistically linked to more water content in the soil in spring (with SoilWaterMAM loading vector opposite to $SWE_{MAX}$ loading vector). Additionally, the earlier snow depletion and thus earlier onset of the thermal growing season relate to enhanced greenness (with the Onset loading vector opposite to the Greenness loading vector). A wide atmospheric warming is shown by increases in $T_{2m}$JJA in most ecoregions (Fig. S6), which is also reflected in the increases of RainRatioJJA (Fig. S20). These increases in RainRatioJJA do not seem to be linked to RainJJA between 1991 and 2023 (Fig. 4 and Fig. S15). Likewise, RainJJA does not seem to be related to higher greenness (see the orthogonal loading vectors in Fig. 4) across ecoregions. It is also worth noting that RainJJA in the northern ecoregions is not in alignment with SoilIceJJA. In turn, the increase in $T_{2m}$JJA is generally aligned with less SoilIceJJA (opposed loading vectors). This is particularly evident in the northern ecoregions. The remaining ecoregions show localized increases in SoilWaterJJA, which is in contrast with the significant decreases in ecoregion 2 (Fig. S21). The same areas in ecoregion 2 show significant increases in VPdJJA (Fig. S22). Additionally, the increased $T_{2m}$SON is in alignment with the increase in RainRatioSON and SoilWaterSON, particularly in the southern ecoregions where the end of the thermal growing season occurs later.

## 4.3 Coastal, latitudinal and altitudinal dependence on trends

The significant increase in length of the thermal growing season (GrowDays) across ice-free Greenland is shown in Figure 5. A pronounced increase in the number of GrowDays occurs in Southwest Greenland at low-lying regions below 600 meters above sea level (asl). At the local scale, significant increases are also found at elevations above 500 m asl, more specifically within Nuup Kangerlua (east of Nuuk) and Angujaartorfiup Nunaa (in between Maniitsoq Ice Cap and Kangerlussuaq). Such areas are located in precipitation shadows with reduced snow depths, but close to glaciers and ice caps. Along the narrow ice-free strips of land in the Southeast, there is a modest increase of GrowDays (approx. 5 days per decade), at several elevations around Tasiilaq. The most pronounced increase in the number of GrowDays occurs along the coast facing the Denmark Strait (approx. 8 days per decade).

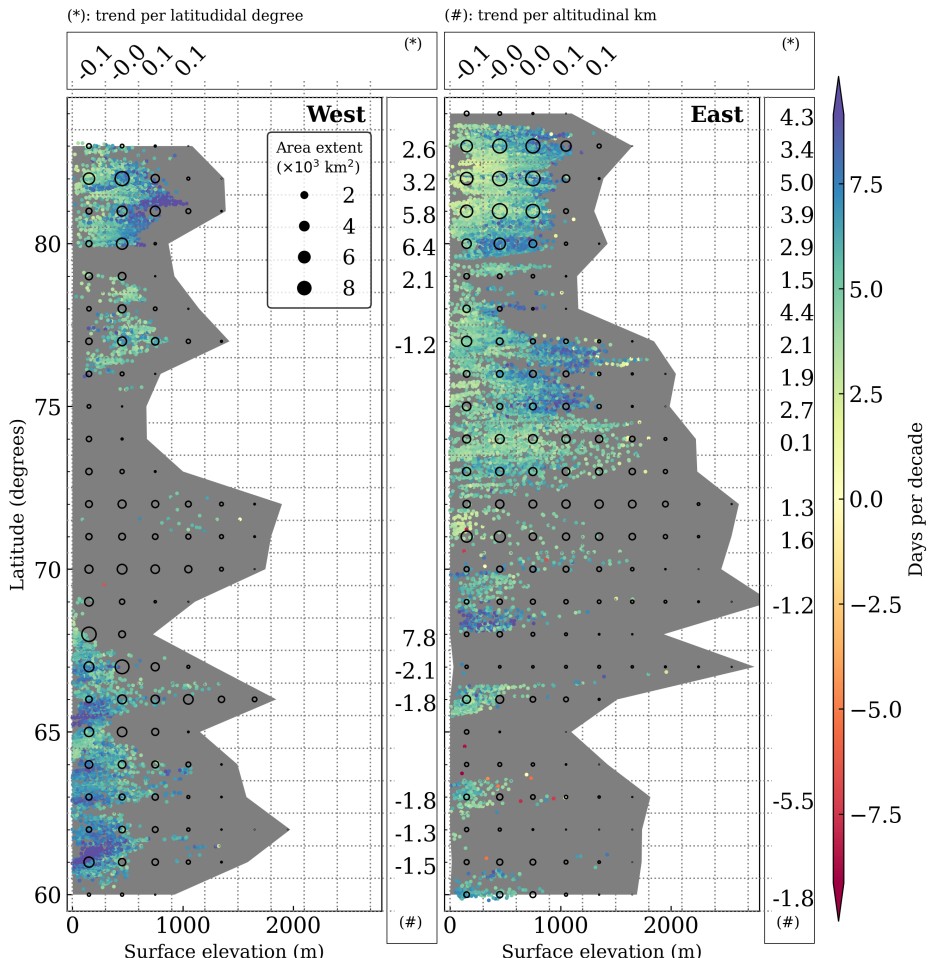

**Figure 5.** Significant trends for GrowDays (in days per decade) in the ice-free part of West (left panel) and East (right panel) Greenland. The trend's elevation dependency (in days per decade per altitudinal km) is binned in one degree of latitude and shown in vertical boxes marked with (#). The trend's latitudinal dependence (in days per decade per latitudinal degree) is binned for every 300 m and is shown in horizontal boxes marked with (*). The background grey shade displays the altitudinal extent of ice-free Greenland in the respective degree latitude, and the black circles represent the area extent by altitudinal and latitudinal bins. At least 50 pixels (approx. 312 km$^2$) are required within each bin to compute its regression, which is otherwise not displayed. Trends are considered significant for confidence levels in the Mann-Kendall trend test when they are higher than 95%, with the null hypothesis that the slope is equal to zero.

The vast and relatively flat, ice-free Jameson Land (east of Ittoqqortoormiit, between 70° and 72°N) shows little evidence of change in GrowDays over the past three decades. At the northernmost part of ecoregion 5 (75°N), areas at low elevation reveal the smallest increase of GrowDays in the ecoregion. This feature becomes even more pronounced in Greenland's north-ernmost regions, exhibiting the highest GrowDays elevation sensitivity (approx. 5 days per decade per km elevation), which is a contrasting elevation dependence in comparison with Southern Greenland. This tendency is modestly evident for the latitu-

dinal sensitivity, mainly driven by trends at high latitudes and elevations. Notably, GrowDays trends decrease with latitude in low-lying areas (< 300 m asl), while GrowDays trends increasing with latitude at higher elevations across North Greenland.

Trends for the onset of the thermal growing season resemble the trends in GrowDays, with earlier starts (approx. 8 days per decade) in southwest coastal Greenland and in the interior of Northeast Greenland. This is a consequence of shallower snow depths, which, combined with warming, have promoted longer snow-free periods. Thus, some areas of these ecoregions show increased trends in the number of frost days in spring.

The relationship between GrowDays and topographical features such as slope and aspect were further explored. As the surface slope is highly correlated with surface elevation, trends in GrowDays tend to significantly decrease with steepness. The dependence between GrowDays and surface aspect is rather complex, without a predominant slope orientation promoting GrowDays, in general. However, latitudes immediately south of Maniitsoq Ice Cap show increases of GrowDays in slopes with southwest orientation. On the East coast, a western slope orientation is particularly pronounced along Jameson Land, whereas northeast exposure appears favourable north of ecoregion 5. The dependence of greenness changes on slope orientation (Fig. S23 and S24) is partially consistent with the dependence of GrowDays on slope orientation. Greenness trends increased in two latitudinal bands facing southeast in ecoregions 1 and 2. In Jameson Land, a similar tendency toward more greening is observed on southwest-facing slopes, while northeast-facing slopes are favoured in the northern part of ecoregion 5.

## 4.4   Greening and greenness expansion

Trends in summer greenness are shown in Figure 6a. Significant greening occurs throughout Greenland, with pronounced greening across all ecoregions. Marked greening in ecoregion 1 is found in East Disko and northeast of Disko Bay. In ecoregion 2, the most pronounced greening is along the inland part of the Kangerlussuaq Fjord. The interior of Nuup Kangerlua shows the highest greening in ecoregion 3. The coastal Kujalleq municipality, facing Labrador Sea, exhibits substantial greening. There are two greening clusters in ecoregion 5, namely Jameson Land in the south and the interior of King Christian X Land in the north. In contrast, decreases in summer greenness are shown along the Southwest coast from ecoregion 1 to ecoregion 3, and in the interior of ecoregion 2.

In order to assess which regions became greener due to greenness expansion, we detected whether a pixel met the summer greenness criterion annually from 1991 to 2023. A detailed explanation on how the study period was split into two to investigate changes in greenness distribution is found in subsection 3.1. A considerable part of summer greening (Fig. 6a) results from greenness expansion (Fig. 6c). With the support of such maps, we can discern that the relationship between changes in greenness and its distribution is not linear. For instance, the central part of ecoregion 2 had as many summers meeting the criterion of greenness in the second half as it had in the first half of the study period. Therefore, the trends in greenness are either related to the greening of the existing vegetation or plant community change in ecoregion 2. The observed greening in the remaining areas seems to be the result of greenness expansion, likely due to the colonization of previously bare ground. The decreasing trend in greenness along the Southwest coast suggests that vegetation is not as dense in the second sub-period as it used to be. Also, greenness seems to be emerging directly adjacent to the ice sheet.

Figure 6b integrates information from both maps by displaying significant temporal changes in greenness as functions of latitude and elevation, colour-coded by the spatio-temporal changes in greenness distribution. Ecoregions 3 and 4 exhibit a shrinkage in greenness at elevations below 500 meters, whereas ecoregion 2 shows a shrinkage up to 1000 meters at specific latitudes. In contrast, expansion in greenness is not only shifting upward but also northward across all ecoregions, with notorious inland advancements south of Kangerlussuaq towards Angujaartorfiup Nunaa in the west and in Jameson Land in the east.

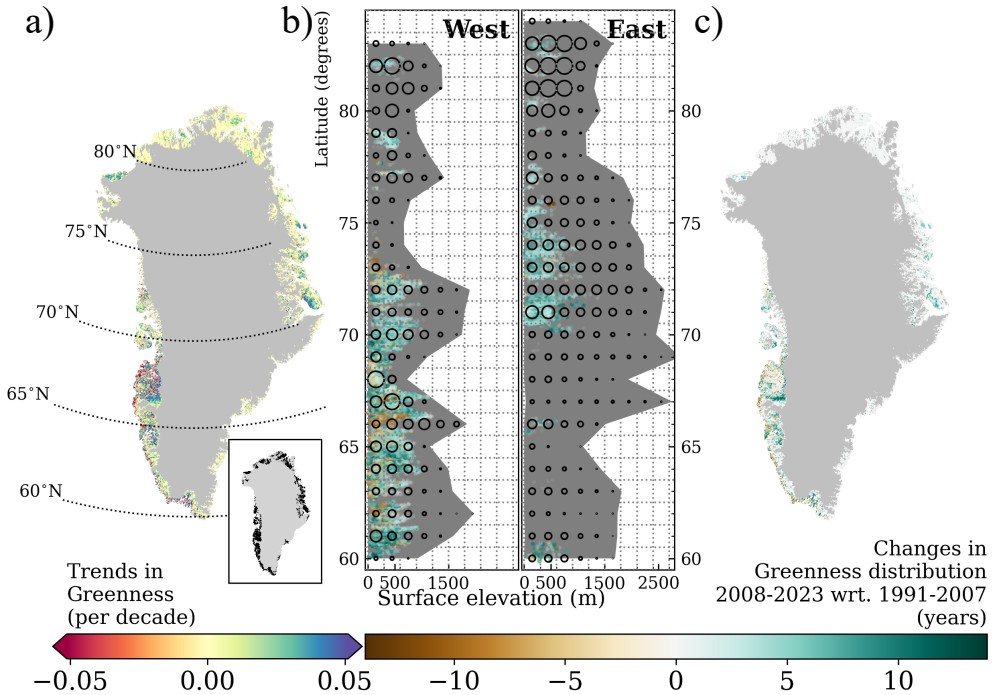

**Figure 6.** Trends in Greenness in Greenland (a). Significant trends are shown in the inset of (a). Trends are considered significant for confidence levels in the Mann-Kendall trend test higher than 95%, with the null hypothesis that the slope is equal to zero. Changes in greenness distribution as a function of latitude and elevation at locations with significant trends in greenness in West and East ice-free Greenland (b). The background grey shade displays the altitudinal extent of ice-free Greenland in the respective degree latitude, and the black circles qualitatively represent the area extent by altitudinal and latitudinal bin. Greenness distribution (c), as the difference in summer greenness counts for the period 2008–2023 with respect to (wrt.) the period 1991–2007

According to Table 2, we show that ecoregion 2 experienced the highest greenness expansion (positive greenness distribution) at 44.2%, along with the highest greenness shrinkage (negative greenness distribution) at 33.4% between 2008–2023 compared to 1991–2007, resulting in an overall increase of 10.7% in greenness distribution. Ecoregion 1 saw the largest increase in greenness distribution at 22.2%, with a greenness expansion of 30.6%. Ecoregion 5 had the lowest greenness shrinkage at 2.7% and an overall increase in greenness distribution of 19.8%. Ecoregions 3 and 4 also experienced increases in greenness distribution, at 18% and 20%, respectively.

**Table 2.** Percentage of expansion and shrinkage of greenness distribution, and ratio (fraction of expanded by shrank area) between 2008 and 2023 with respect to the period 1991–2007 in % of the total ecoregion area

|  | Ecoregion 1 | Ecoregion 2 | Ecoregion 3 | Ecoregion 4 | Ecoregion 5 |
|---|---|---|---|---|---|
| Expansion | 30.6 | 44.2 | 38.6 | 28.0 | 22.5 |
| Shrinkage | 8.4 | 33.4 | 20.5 | 7.9 | 2.7 |
| Ratio | 3.6 | 1.3 | 1.8 | 3.5 | 8.3 |

The southernmost and northernmost ecoregions experience the highest expansion ratios, ranging from three times to eight times more expansion than shrinkage in ecoregions 4 and 5, respectively. Overall, greenness expansion in ice-free Greenland increases two times faster than greenness shrinkage between 2008–2023 compared to 1991–2007.

## 5    Discussion

### 5.1    Key findings and interpretation in the context of the current literature

#### 5.1.1    Changes in greenness extent and distribution

Greenness extent has increased over time across Greenland, with an increasing rate of 2% per decade in ecoregion 1 to up to almost 6% per decade in ecoregion 4 (Figure 2). When comparing the recent half of the time-series (2008–2023) to the earlier half (1991–2007), the distribution of greenness has also changed. In ecoregions 3 and 5, the distribution of greenness expanded to nearly double and eight times the size of the areas that shrank, respectively (Table 2). Within the time-series, 535 maximum greenness extent was observed in 2019, aligning with the end of a period of frequent, long-lasting, and intense summer atmospheric blocking conditions in the vicinity of Greenland, conditions which promoted advection of relatively warm and humid air from the North Atlantic along West Greenland (Silva et al., 2022). By investigating greenness distribution, we disentangled greening of already-green areas from the emergence of new green areas (Fig. 6). Additionally, this analysis allowed us to identify areas associated with a shrinkage in greenness. Ultimately, we investigated the associations between bio- 540 climatic indicators and greenness, going beyond the statistical links between greenness and its extent with large-scale weather patterns

#### 5.1.2    PCA performance and basis for interpretation

To better understand how bio-climatic indicators co-varied with greenness between 1991 and 2023 across ice-free Greenland, a set of bio-climatic indicators, including greenness, were statistically aggregated. This was achieved by applying principal com- 545 ponent analysis (PCA) to remote-sensing data of greenness and the output from a polar-adapted reanalysis. We demonstrated that the first two principal components account for most of the variability among the 16 combined bio-climatic indicators (Fig. 4). Given the fact that the chosen indicators are interlinked with supplementary indicators, we extended the interpretation to

over 30 bio-climatic factors, with the support of climatology and trend maps. PCA effectively clustered bio-climatic indicators that co-vary with summer greenness based on data from 1991 to 2023. Numerous indicators closely co-vary with near-surface air temperature and topography (PC1) and, to a lesser extent, with precipitation patterns (PC2). The rank of relative importance of individual bio-climatic indicators depends on ecoregion, with the number of days of the thermal growing season (GrowDays) being the most relevant across all ecoregions, followed by soil ice during summer (SoilIceJJA) in the northern ecoregions and soil water in spring (SoilWaterMAM) in the southern ecoregions.

### 5.1.3 Bio-climatic changes in the northern ecoregions

Upon investigating the bio-climatic factors driving greenness changes in the northern ecoregions, we found that areas related to greenness expansion appear to be associated with a rise in SoilWaterMAM along with declines in both spring soil ice content trends (SoilIceMAM) and maximum snow depth (SWE$_{MAX}$). Notably, there are regional exceptions to these patterns. For instance, in Northwest Greenland (including ecoregion 1), coastal areas exhibit increasing SWEMAX and delays in the onset of the thermal growing season (Onset). This suggests that snow depth may buffer greenness in these regions. Conversely, areas related to greening are statistically linked with rising SoilWaterMAM, accompanied by higher spring temperatures (T$_{2m}$MAM) and earlier Onset.

Despite regional trends of higher summer rainfall amounts (RainJJA) in northern Greenland (Niwano et al., 2021), we did not find a clear link between greening and changes in RainJJA. Interestingly, trends in summer soil water content (SoilWaterJJA) and soil ice content (SoilIceJJA) are both negatively correlated to near-surface air temperatures in summer (T$_{2m}$JJA). This association could result from surface thawing and subsequently increased evaporation caused by higher vapour pressure deficits in these northern areas (Fig. S22). The greening of the recently emerged vegetated areas in the northern ecoregions responds to different seasonal soil water contents. Greening in ecoregion 1 demonstrates a stronger positive correlation with SoilWaterMAM patterns, similar to the remaining southwestern ecoregions. In contrast, ecoregion 5 is more closely connected with SoilWaterJJA, likely due to a later Onset (Fig. 1).

To better understand the reduction in SoilIce, which highly correlates with greening, we investigated the relationships among changes in SWE$_{MAX}$, MeltRate, SoilWaterMAM, SoilIceMAM, and greenness, and examined the levels of SoilIceDJF. We found no significant trends in SoilIceDJF. This suggests that, to a certain extent, the proportion of frozen surface has been restored during the cold season. Changes in SoilWaterMAM are moderately proportional to changes in SoilIceMAM, indicating that the increase in the SoilWaterMAM primarily originates from snowmelt. Subsequently, the presence of liquid water in soil with higher thermal conductivity, coupled with shallow snow depths (and eventually snow-free conditions), allows for a more efficient exchange of energy between the surface and the atmosphere, consequently leading to thawing. These changes create a more favourable setting for vegetation growth, enabling some plants to expand or establish in areas where frozen conditions previously limited their presence (e.g., Shijin and Xiaoqing 2023; Yang et al. 2024).

We report little to no change in the GrowDays length and Onset along the coast in Northeast Greenland. In situ long-term measurements (e.g., Schmidt et al. 2023) state that some taxa may have reached their phenological limits despite ongoing warming. Assmann et al. (2019) suggests that temperature and snowmelt explain the effects on spring phenology in Zackenberg,

contrary to sea-ice break up in the Greenland Sea. However, the continuous southward transport of cold waters, frequently with sea ice, through the East Greenland current likely stabilizes the onset of the GrowDays at the coast. This seems to be corroborated by our study given the GrowDays increase towards the interior and high elevations in Northeast Greenland, resulting in elevation sensitivity, as shown in Figure 5. The lengthening of GrowDays in these inland areas could facilitate the colonization of new vegetated regions.

### 5.1.4 Bio-climatic changes in the southern ecoregions

The observed trends in $SWE_{MAX}$ across southern ecoregions reveal important dynamics influencing the timing of the Grow-Days. Southern ecoregions with significant decreases in $SWE_{MAX}$ show early $SWE_{MAX}$ day of the year (DOY, Fig. S18) that leads to early Onset. However, the underlying mechanisms driving these trends differ between West and East Greenland. Despite an increase in fresh snow accumulation and a reduction in drought days during the spring, the observed declining trend in $SWE_{MAX}$ for West Greenland is linked to a decrease in winter snowfall (Fig. S16); conversely, for East Greenland, it is attributed to reduced spring snowfall.

The drier conditions in the interior of ecoregion 2 have led to substantial losses in SoilWaterJJA (Fig. S21) with minimal changes in SoilIceJJA. We attribute the decrease in SoilWaterJJA to higher rates of evaporation in the more continental areas of the ecoregion, supported by the significant increase in vapour pressure deficit (VPdJJA, Fig. S22). The energy necessary to convert liquid water into water vapour (latent heat) cools down the soil. This is shown in our results, where the increase in $T_{2m}$JJA and VPdJJA results in little changes in SoilIceJJA in ecoregion 2. These drying processes collectively force and constrain vegetation expansion toward areas closer to water bodies, where soil moisture levels can better support vegetation (e.g., Chen et al. 2023; Gamm et al. 2018).

While no significant trends in $SWE_{MAX}$ are observed within ecoregion 2, the presence of subsurface runoff from snow and ice likely plays a critical role in maintaining or increasing SoilWaterMAM in the region. Episodic rainfall events in spring can also contribute to increased SoilWaterMAM. However, no significant changes in accumulated rain (RainMAM) or rain ratio in spring (RainRatioMAM) were found in the region, particularly linked with greenness changes. While the spatio-temporal heterogeneity in these processes challenges direct comparisons between regions, it also emphasizes the critical role of local bio-climatic drivers in influencing greenness changes.

### 5.1.5 Bio-climatic changes across ecoregions

For most ecoregions in ice-free Greenland, we find that snowpacks are becoming shallower and consequently melt slowly, but earlier in the season. This feature was mentioned by Musselman et al. (2017) and is attributed to global warming. Musselman et al. (2017) explains that in Western North America regions with shallower snow are experiencing snow season contractions. Shallow snow is susceptible to snow season contraction because shallow snow requires less energy to initiate melt than deeper snow. This earlier start of the ablation period occurs at a slower rate due to a combination of near-surface warming with relatively low solar altitude angles. In contrast, for deep snowpacks that require more energy to initiate runoff, it is also more likely for the snowmelt water to refreeze within the snowpack (Dingman, 2015). Therefore, early season slow snowmelt rates

in shallow snowpacks allow for efficient soil water percolation and subsequent water storage (Stephenson and Freeze, 1974). The successful percolation of liquid water into soil plays a key role in tundra regions during the snow ablation period and the start of the growing season, as during this time, soils are generally dry due to high drainage (Migała et al., 2014). Increased water availability in the soil could stimulate dormant microbial communities and thus increase the decomposition of soil organic matter, releasing soil nutrients (e.g., Glanville et al. 2012; Salmon et al. 2016; Xu et al. 2021). This, in turn, could prime the soil for earlier and more efficient vegetation growth and colonization. The increased SoilWaterMAM, $T_{2m}$MAM, and GrowDays indicated in our results could thus improve conditions for plant growth and colonization, especially in the southern ecoregions. Therefore, it is expected that plants will develop more in early summer. Such conditions in conjunction with favourable weather patterns in summer, associated with increased $T_{2m}$JJA and longer periods of solar radiation (Barrett et al., 2020), allowed for higher greening and more greenness expansion. The same weather patterns also brought more drought and heat days, but apparently without an immediate negative impact on greenness.

The early Onset has also contributed to local increases of freezing days (FrostDays) and rain on snow days (RainOnSnow). However, the effect of FrostDays is not reflected in greenness levels. This could be related to freezing episodes occurring during the dormant phase, rather short freezing episodes which are not long enough to inhibit plant growth, or to a demonstration of certain plant community resilience (e.g., Gehrmann et al. 2020; Körner and Alsos 2008). Nevertheless, the early onset of GrowDays allows vegetation to be potentially more active and responsive to solar radiation, particularly in the ecoregions at lower latitudes with longer sun exposure (Opała-Owczarek et al., 2018).

### 5.1.6   Spatio-temporal changes in greenness and bio-climatic factors reported in literature

Grimes et al. (2024) investigated land cover changes across Greenland by using Landsat images from the late 1980s to the late 2010s, and found spatial patterns of vegetation change similar to our findings (Figure 6c). For instance, they showed increased vegetation coverage southwest of Kangerlussuaq. They attributed this increase to receding lakes, a process happening since at least 1995 (Law et al., 2018). Similar to our findings, Grimes et al. (2024) detected increased vegetation cover in the northeast of Kangerlussuaq. This has been shown and modelled in other parts of the Arctic tundra (e.g., Bosson et al. 2023; Jones and Henry 2003). Specifically, in ecoregions 2 and 5, greenness expansion is not only occurring toward the inland regions, but also upward (Fig. 6b). The interior of Greenland, less exposed to frontal systems developing over the Atlantic and with meltwater availability, seems to be a more favourable area for greening and expansion, as shown in ecoregions 1 and 2 (Figure 6c). Increasing greenness levels were also found with a tendency for slopes facing southeast in ecoregions 1 and 2. According to Grimes et al. (2024), the retreat of vegetation in front of the Maniitsoq Ice Cap is leading to the exposure of bedrock. Additionally, the less dense summer vegetation in coastal ecoregion 2 and along ecoregion 3 is suggested by Grimes et al. (2024) to be related to increases in freshwater, likely due to increased river discharge. A small-scale study, north of Kangerlussuaq, reports declining growth of deciduous shrubs (Gamm et al., 2018) since the 1990s. A similar signal is seen regionally in our results. They reported that the decrease is likely due to water soil scarcity, a markedly pronounced negative trend for SoilWaterJJA in the region. The derived spatio-temporal patterns of summer rainfall (Fig. S15) and rain ratio (Fig.

S20) are also in agreement with literature (e.g., Huai et al. 2022; van der Schot et al. 2023), especially on the significant increase of the rain ratio in North and West Greenland in summer and autumn.

The widespread summer greening (Fig. 6a) could be due to encroachment of vegetation on previously bare surfaces and changes in plant community composition at certain sites (Grimes et al., 2024). Greenness correlates best with biophysical properties, such as leaf area index (Myers-Smith et al., 2020). Therefore, we may argue that the greening is generally related to tundra vascular green vegetation expansion throughout the past three decades, as proposed by Sturm et al. (2001).

## 5.2 Significance and implications

Between 1991 and 2023, Greenland has experienced longer GrowDays. These longer seasons, combined with higher near-surface air temperatures, have favoured the greening and greenness expansion during the studied period. However, further investigation is needed to understand the impacts of longer seasons of GrowDays on vegetation and ecosystem functioning, as some regions experience freezing conditions due to reduced snow cover and an earlier Onset, and other regions face heat stress and changes in precipitation patterns throughout the growing season. The significant decreases in snow cover reported have led

to extended periods of low surface albedo, allowing more energy absorption and contributing to increased surface warming. The observed changes in greenness distribution enhance the surface albedo feedback, with varying effects that extend beyond the growing season and which depend on the vegetation type (e.g., Blok et al. 2011; Loranty et al. 2011). A surplus in the surface energy budget results in surface warming and promotes surface thaw, especially in Greenland's northern regions. However, this excess surface energy may also contribute to latent heat release to cool the surface, depending on the vapour pressure

deficit and the vegetation canopy (Heijmans et al., 2022). Initially, increased vegetation leads to greater carbon sequestration. However, substantial permafrost thaw potentially caused by the vegetation increase could also release carbon, offsetting the compensation from vegetation-based carbon sequestration (Glanville et al., 2012).

The terrestrial Arctic biosphere is an important regional source of primary biological aerosol particles (PBAPs), highly correlated with near-surface air temperature and vegetation. These aerosol particles were found to play an important role in

cloud formation, specifically in the Arctic where there are low aerosol concentrations (e.g., Pereira Freitas et al. 2023; Sze et al. 2022). Therefore, increased near-surface air temperature and changes in vegetation can significantly impact cloud properties, such as cloud phase, radiative properties, cloud lifetime, and precipitation patterns, which in turn impact the surface conditions, including the surface energy budget. Additionally, low clouds and fog are also very likely to become more frequent in certain coastal parts of Greenland due to decreasing sea ice (Song et al., 2023). Water droplets from fog can effectively be retained

by tundra vegetation and are not accounted for as a water source. This interaction between fog, vegetation, and soil conditions should be better investigated, particularly for coastal tundra vegetation. The potential warming and shading conditions were shown through an experimental study in West Greenland to reduce carbon sequestration from vegetation (Dahl et al., 2017). These PBAPs can be carried towards snow- and ice-covered regions such as the Greenland Ice Sheet, contributing to the surface darkening and enhancing algae growth (Feng et al., 2024), which again leads to increased melt, particularly of the ice bodies

in the vicinity of densely vegetated regions.

Longer thermal growing seasons in areas with shallow soils could have significant implications for biodiversity on a large scale. Prolonged warmth may foster the proliferation of shrubs, leading to increased "shrubification" and potentially resulting in the homogenization of species compositions across these landscapes (Myers-Smith et al., 2011). In permafrost regions with deeper soils, the deepening of the active layer may benefit graminoids, as they possess deeper root systems that allow them to exploit water and nutrients at greater depths (Wang et al., 2017). Similarly, in the event of permafrost degradation leading to deep water infiltration (Liljedahl et al., 2016), graminoids would gain an advantage for the same reason. These ecological shifts might also affect animal communities such as birds (Boelman et al., 2015) and arthropods (Høye et al., 2018). Moreover, longer growing seasons could foster conditions conducive to the establishment and spread of invasive species, further threatening the native biodiversity and altering the delicate balance of these unique environments (e.g., Elmendorf et al. 2012; Pearson et al. 2013).

Our study identifies a set of bio-climatic indicators relevant to understanding interconnectedness with greenness. These statistical interconnectedness and spatio-temporal changes in environmental indicators have been corroborated by experimental studies across the Arctic (e.g., Chen et al. 2023; Gamm et al. 2018; Grimes et al. 2024; Huai et al. 2022; Migała et al. 2014; Musselman et al. 2017; Opała-Owczarek et al. 2018; Schmidt et al. 2023; Stephenson and Freeze 1974; van der Schot et al. 2024). Such findings enabled us to extend our interpretation of the associations among bio-climatic indicators with greenness to larger scales, with apparent features dependent on the ecoregion and latitude. This consistency with other studies demonstrates the potential of the Copernicus Arctic regional reanalysis (CARRA) for use in biogeographic studies by extending insights from experimental studies into large-scale analyses. Such insights can now be used to validate whether the same bio-climatic indicators interdependence is captured during the historical period of global climate models. This would guarantee more confidence in the use of these indicators for the study of future vegetation changes across Greenland under a changing climate.

### 5.3 Study limitations and future research directions

The use of NDVI has limitations in characterizing changes in plant communities, as noted by Myers-Smith et al. (2020) . For example, while NDVI effectively captures plant communities with a high composition of shrubs (e.g., Blok et al. 2011), it struggles to detect communities with low infrared reflectance or those that are sparsely vegetated. Combining our methods with the approaches Karami et al. (2018) and Rudd et al. (2021) used to categorize tundra vegetation across ice-free Greenland will likely yield an optimal assessment of spatio-temporal changes among plant communities. However, high spatial resolution optical satellite images from Landsat 8 and Sentinel 2 have only been collected for approximately one decade.

Another limitation of NDVI analysis is that pixels representing certain vegetation types, such as wet tundra, may be erroneously influenced by adjacent water bodies, potentially affecting the measured greenness, particularly in places like ecoregion 2. Additionally, certain low-lying strips near fjords are very narrow, potentially causing errors in pixel reflectance calculations due to limited spatial resolution. Remote-sensed NDVI products are highly dependent on weather conditions to accurately retrieve surface reflectance. Occasions with snow, shadows, and clouds are thus assumed to be evenly distributed through time. The NDVI datasets used in this study come from two NOAA satellite products, each employing a different sensor type. The

absence of overlapping temporal datasets limited our uncertainty assessment, and the potential for mismatches between the datasets cannot be discarded. This lack of a common calibration period raises concerns about the reliability of long-term time-integrated NDVI analysis. Additionally, greenness is highly responsive to prevailing atmospheric circulation patterns. Between 2010 and 2019, an exceptional increase in frequency and intensity of anti-cyclonic activity promoted the advection of relatively warm and humid air from the North Atlantic towards Southwest Greenland (Silva et al., 2022). Such periods have favoured exceptional vegetation growth across western ecoregions, as shown in our results. However, surface reflectance retrievals may have been impacted by cloudiness, partly hindering the spatio-temporal changes in greenness. Despite the frequency of prevailing atmospheric circulation patterns, there is a superimposed warming signal, with less cold conditions likely promoting vegetation growth that is poorly captured due to cloudiness.

Soil nutrient distribution is also highly influenced by topography – not only elevation but also relief and aspect – though these factors are not entirely reflected in the greenness changes observed on a large scale in our study. According to Anderson (2020), organic-rich soils in Greenland generally accumulate on north-facing slopes, with little to no accumulation on the south-facing slopes as a result of precipitation patterns, whereas in valley bottoms and at slope breaks, thicker fen-like, organic-rich deposits accumulate. Even though we have investigated how vegetation and bio-climatic indicators are changing as a function of latitude, elevation, slope, and aspect, potential influences due to relief and aspect are apparent in our results, potentially more evident at the local scale due to less spatial heterogeneity.

Our results show that summer greenness appears statistically unresponsive to changes in rain on snow days (RainOnSnow) and below-zero temperatures (FrostDays) during spring. Future work could focus on the analysis of extreme events and their impacts on greenness.

Although CARRA can capture spatio-temporal changes in relevant bio-climatic indicators interacting with greenness, it likely misses certain interactions, such as the effect of shrub canopies on ground conditions. For instance, shrub growth and greenness expansion may lead to increased snow trapping during winter, thereby enhancing winter soil insulation (Lamichhane, 2021). This process could, in turn, promote increased microbial activity (Wang et al., 2024). Additionally, greater shrub cover may result in enhanced shading during the subsequent summer (Blok et al., 2010). This implies the existence of potential feedback loops, as suggested by previous studies (e.g., Hallinger et al. 2010; Barrere et al. 2018), that cannot yet be properly assessed. Additionally, vegetation type has been recently posited as a strong predictor of summer surface latent and sensible heat fluxes (Oehri et al., 2022). A better representation of the permafrost extent and active layer thickness, together with the inclusion of dynamic tundra vegetation models within CARRA, could be beneficial to deepening our knowledge on interactions among atmosphere, vegetation, carbon and nitrogen cycling, water, and permafrost dynamics.

Permafrost areas will likely continue to be locations for future vegetation spatial expansion (Chen et al., 2023), especially under the current trend of decreased summer precipitation. Moreover, permafrost thawed areas are also susceptible to fast drying (Liljedahl et al., 2016) and potentially sudden vegetation changes. Ultimately, plants can fixate along streams and small lakes as future land ice melt will continue to provide sediments and nutrients through runoff (Migała et al., 2014).

# 6    Conclusions

Our study aimed to better understand the long-term, large-scale relationships among various bio-climatic indicators and their collective associations with summer greenness in ice-free Greenland. We utilized remote sensing Normalized Difference Vegetation Index and bio-climatic indicators from the Copernicus Arctic regional reanalysis between 1991 and 2023. Bio-climatic changes are influenced by a complex set of factors, not only centered in summer, but also dependent on winter and spring atmospheric temperatures, precipitation, solar radiation, soil properties, and soil water availability.

We conclude that regions under greenness expansion in ice-free Greenland are associated with reductions in winter precipitation. The resulting shallower snowpacks melt earlier in the season but more slowly. We interpret that this slow snowmelt rate allows the ground to retain more liquid water during the ablation period. Experimental studies suggest that such conditions, occurring before the thermal growing season commences, facilitate vegetation growth. Longer thermal growing seasons, accompanied by prevailing summer weather patterns – reaching their peak in 2019 – that promoted warmer and clear-sky conditions, also contributed to vegetation growth.

The spatio-temporal changes in summer greenness distribution depend on ecoregion, elevation, and latitude. Overall, the bio-climatic changes during the study period led to more greenness expansion, particularly towards the interior and northward. Ultimately, to enhance our understanding of the intricate interactions among the atmosphere, vegetation, and cycles of carbon and nitrogen – as well as water and permafrost dynamics – this study underscores the need for integrating dynamic tundra vegetation schemes, especially for future projections.

*Data availability.* The Normalized Difference Vegetation Index CDR used in this study was acquired from NOAA's National Center for Environmental Information (http://www.ncei.noaa.govhttp://www.ncei.noaa.gov). This CDR was originally developed by Eric Vermote and colleagues for NOAA's CDR Program.

Schyberg et al. (2020) was downloaded from the Copernicus Climate Change Service (2024). The results contain modified Copernicus Climate Change Service information 2024. Neither the European Commission nor ECMWF is responsible for any use that may be made of the Copernicus information or data it contains.

The North Atlantic Oscillation and Greenland Blocking Index data were obtained from the NCEP/CPC and the PSL/ESRL, respectively. Both climate oscillations were seasonally standardized relative to the period 1950–2000.

*Author contributions.* The inspiration for the paper was brought by BSW, EMB, JA and NdV, the concept and methodology was developed by TS, the original paper draft was written by TS, the data were processed and analysed by TS, all authors contributed to the interpretation of results as well as reviewing and editing the final paper draft.

*Competing interests.* The authors declare that they have no conflict of interest.

*Acknowledgements.* The University of Graz is acknowledged for support of publication costs. Brandon S. Whitley, Elisabeth M. Biersma and Natasha de Vere have received funding from the Carlsberg Foundation. The main author would like to acknowledge the use of OpenAI's ChatGPT for assisting in the writing and editing of this manuscript. The chatbot was utilized to enhance the clarity and readability of the text. A special thanks to Inger Greve Alsos and Therese Rieckh for their valuable suggestions.

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
