# Peer review of "Bio-climatic factors drive spectral vegetation changes in Greenland"

_EGUsphere, 2024_

## Author Comment (AC1)

We would like to thank the editor for the opportunity to publish our manuscript at EGUsphere and to the nine referees for their thorough evaluations with constructive comments that will improve the manuscript greatly. In the following, we will address the referees' comments point by point. We mark the comments given by the referee in red, provide our answers and comments in black, and indicate how we will address the amendments in the manuscript that we plan to submit upon editor's decision in green. Note, that we add the concrete amendments planned not at all points as this in parts make the replies less readable and only where we will add/adapt a section directly and concisely. In many parts we only conceptually explain how we will adapt content, while the exact implementation remains until the revised version.

We would like to announce that should the manuscript advance to the revision stage; Verena Haring from the Department of Biology at the University of Graz will be included as a co-author in recognition of her valuable assistance with the point-by-point responses.

Tiago Silva, on behalf of all co-authors.

**RC1**: 'Comment on egusphere-2024-2571', Rúna Magnússon, 20 Oct 2024

Dear authors, dear editor
Thank you for inviting me to review "Bio-climatic factors drive spectral vegetation changes in Greenland", by T. Silva et al. for Biogeosciences. The manuscript explores the role of a wide range of bio-climatic factors in explaining satellite-derived vegetation dynamics in Greenland. The authors aimed to identify which sub-surface and above-surface climate factors were associated with greening in Greenland, and how such associations differed among ecoregions and latitudinal and altitudinal gradients. They report that increases in the duration of the thermal growing season show the strongest association with greening, with additional influences of snowpack dynamics, and differential strength of association across regions and altitudinal and latitudinal gradients.
This study is relevant in the context of rapid ongoing climate change in the Arctic, observed dynamics of "Arctic greening" and their implications for the future functioning of tundra ecosystems. The authors analyze a substantial amount of data for a large region, using various sources and environmental disciplines in a holistic and, generally, appropriate way. The manuscript is well within the scope of Biogeosciences and presents a relevant and timely case. I do, however, have several major concerns about some of the methodological choices and the structuring and argumentation of the work. I advise a round of thorough revision and rewriting of substantial parts of the manuscript before it can be considered for publication. This has resulted in a rather lengthy review report, but I would also like to stress that many of the points I raise are interrelated or specific examples of the major points, so I hope it is not discouraging. I am sure the ms will find a good home in a respected journal.
I have performed this review together with 7 MSc students for an open review course assignment at Wageningen University. Their help has been valuable, and they appreciated the opportunity to learn from this ambitious and relevant paper, and to contribute to the scientific publishing process. We have all enjoyed this activity and

we wish you all the best as the manuscript comes to full maturity!

Rúna Magnússon,
with input from Annika Robben, Djordy Potappel, Aron den Exter, Muriël de Vries, Rikuto Shinagawa, Yente Reniers and Yorick Kwakkel.

**Major comments**
1. I hope the authors can make clarify how the potential mismatches between AVHRR and VIIRS NDVI products (e.g. masking differences) have been accounted for during statistical analysis and trend detection. Explanations on how this was done are sparse and not sufficiently clear to understand the implications.
Beside adding the shaded min-max range in Fig. 2 (that I also don't fully understand the procedure behind, can this be clarified?),
how did you prevent the use of two different records and sensors from affecting your temporal trends? And especially, how do you prevent this from unduly influencing the comparison between 2008-2023 and 1991-2007, that you describe in L. 380-392? This appears to be based on counts of NDVI > 0.15, where differences in bandwidth and snow/water/cloud detection easily become problematic. Miura et al. (2012) may be an appropriate source to evaluate the validity of trend detection across two satellite platforms, and you may want to statistically test for absence of trend breaks coinciding with the switch from one platform to another.

The NOAA Climate Data Record (CDR) of AVHRR NDVI - Version 5 and the NOAA CDR of VIIRS NDVI - Version 1 are developed by Eric Vermote and colleagues (Vermote et al. 2018 and 2022) for NOAA's CDR Program. Both records have been processed considering the same atmospheric features as in Miura et al. (2012) and both processed records are posterior to Miura et al. (2012) proposed correction. Also, the correction proposed by Miura et al. (2012) is not assessed in polar regions, which may contribute to additional uncertainties.

We follow similar approaches of recent literature (e.g., Madson et al. 2023, Pourmohamad et al. 2024) that make use of the full AVHRR NDVI and VIIRS NDVI without additional corrections. As stated in Section 2.2, Vermote and colleagues for NOAA's CDR Program use MODIS to spectrally calibrate AVHRR (Vermote et al., 2018) and VIIRS (Skakun et al., 2018). As NOAA does not provide an overlapping period for AVHRR and VIIRS, we are unable to compare both processed products and quantify biases in polar regions. Nevertheless, we will make sure that we add to the discussion that the potential mismatches between AVHRR and VIIRS NDVI products cannot be discarded and, in a revised version we will provide the greenness trends before and after the sensor change in order to assess potential mismatches between sensors, bearing in mind that differences can also rise from other sources such as the interannual variability of the atmospheric conditions before and after 2014. In addition, we address the variability range during the AVHRR period in Figure 2 and discuss it in detail under a minor comment (point 9) below.

2. Your methodology is ambitious and extensive, which is laudable. It does however lead to many choices during the processing of the data, and not all of these have been properly backed or described yet. Examples include (1) the use of 0.15 as an NDVI

threshold without a reference, (2) the described use of CryoClim data that only go to 2015 without any visible inclusion of these data throughout later analyses, (3) why has only altitude, and not for example slope aspect, been included into the study? These, and further examples, are given in the minor comments. I suggest that the authors critically go through every step in the methodology and check whether all choices are described in sufficient detail for an independent reader to reproduce the study, and that choices are back-up either by literature, data or statistics. If needed, details on processing can be described in a supplementary methods section to prevent disruption of the flow of the main text.

Thanks for your kind work endorsing our ambitions!

Indeed, we will carefully review the clarity of the Methods to assure reproducibility in future interdisciplinary studies. Some specific comments below:

(1) Liu et al. (2024) is cited in the first sentence of Section 3.1, which shows sparse vegetation, with NDVI of approximately 0.15 at the start of the growing season at Disko Island. Additional online source is provided in a minor comment (point 7) below.

(2) long-term reanalysis products commonly use several datasets that do not cover the full period of the reanalysis. Additional examples are provided in a minor comment (point 5) below.

(3) for a matter of simplicity, the investigation of vegetation distribution as a function of slope and aspect was not considered so far. We will follow the advice and will include similar charts as for Figure 5 on surface slope and aspect in the supplementary material, whenever needed to support our Results. More information is provided in a minor comment (point 40) below.

3. From L. 227 onwards, it reads as if the distinction between methods, results and interpretation of results (discussion) is lost. For example: results and maps are presented in the methods in L. 227-247. New information on choices of processing, variable selection and statistical tests (Pearson correlations) are introduced in the results in L. 275-305, L. 311-314 L. 380-384 and many other places. Throughout the entire (lengthy) results section, interpretations are added that go beyond the statistical results of your own methods. Lines 320 -350 for instance are very speculative for a results section, and other paragraphs and show similar interpretation or speculation. These would be better suited for the discussion and require backing by references. Please rewrite the methods-results-discussion in such a way that: (1) all methodological choices and tests are explained in the methods (2) only numerical and statistical outcomes are presented in results (with a minimum interpretation to make the results understandable, e.g. writing out abbreviations and description of patterns) and (3) interpretation and relation to unmeasured mechanisms such as permafrost, latent heat processes or photosynthesis are only kept for the discussion.

Thank you for pointing this out. We will revise the Methods, Results and Discussion to make the differentiation between these sections more evident.

4. In the results section and abstract, observed greenness dynamics are attributed to processes such as nutrient dynamics and permafrost. This gives the reader the impression that such variables were included or that you can at least confidently

attribute greening dynamics to such processes. Given the set of bioclimatic factors that were included, however, I doubt whether you can make such claims. These processes can be touched upon in the discussion, with support from literature, but should not be presented in a way that readers might think that these are actual conclusions from this study. I also think that to properly discuss their role in the discussion, you will need to evaluate several lines of reasoning more critically: are the subsurface products (soil water and soil ice) that you include, given the limited representation of subsurface dynamics in the used reanalysis products, actually representative of permafrost conditions or hydrology? How can you better argue the role of snowmelt rates in relation to microbial activity and nutrient dynamics, especially to an audience that may not be familiar with works such as musselman et al.? Because at first it is very counterintuitive that shallower snowpacks melt more slowly and with the current explanation provided, this line of argumentation is very hard to follow. I suggest you evaluate to what extent your bioclimatic variables are representative of processes such as permafrost dynamics and melt rates and nutrient dynamics, discuss their potential roles in the discussion section, and refrain from making any hard statements about their role in the abstract/results/conclusion sections.

The Copernicus Arctic Regional Reanalysis (CARRA) -- Full system documentation (Schyberg et al., 2020) reports that areas with permafrost in Greenland are not fully described in the present model version. Nevertheless, the HARMONIE-AROME regional numerical weather prediction model (Bengtsson et al., 2017), uses SURFEX (Masson et al., 2013) a multi-layer surface model that accounts the interaction between soil-biosphere-atmosphere, computing specific models dependent on the surface type (e.g., vegetation, soil, snow), allowing soil water phase changes and enabling runoff over frozen and unfrozen soil. This implies that even though the reanalysis utilized does not integrate a surface model incorporating permafrost or nutrient dynamics, there is still an accounting for energy and mass exchange, achieved through different modelled processes that are influenced by the surface type.

Given the limited access to nutrients in the High and Low Arctic, organic matter decomposition is inferred due to microbial activity. The water availability due to permafrost and snow melt could stimulate decomposition of organic soil. We will discuss and review further in-situ studies supporting our interpretation of the modelled results.

We acknowledge that our reference to the work by Musselman et al. (2017) has not been clear enough. In the revised version we will add: Due to rising temperatures, snow begins melting earlier in the year. This results in the maximum snow depth being reached earlier than in the past. However, early in the year, the available energy from the atmosphere is still relatively low, which means the snow may melt slower than it used to be before the warming. This slow melting is especially noticeable in thinner layers of snow, which need less energy to start melting. On the other hand, thicker layers of snow persist until late in the year when conditions like higher sun angles provide more energy, leading to a quicker and more intense melting process.

**Minor comments**

1. The writing could be improved by splitting up some very long compound sentences into shorter ones. I provide some examples in the "technicalities", but I recommend a thorough re-reading for writing style and grammar.

Thank you for this very relevant point. We will thoroughly review and revise our sentence construction in order to improve readability and understanding.

2. The abstract ends with the conclusion that you "identify a set of bioclimatic variables" and that you provide a "basis to validate bioclimatic indicators from climate models". Your conclusions section states more or less the same. I suggest that you reflect more specifically on how exactly your findings help to achieve this (more to the point). This will hopefully also better explain how you advance the field, since the role of growing season onset and snowmelt timing are already well established in Arctic ecological studies.

Thank you for pointing out that we need to further develop our conclusions to show how our work advances the field. In our study, we show the potential of a polar-adapted, high-spatial resolution reanalysis product on capturing the combined role of a set of bio-climatic indicators with spectral greenness over a period of more than 30 years across the Greenland scale. We report the potential of a large-scale product for biogeographic research and discuss the application of a set of bioclimatic indicators in the validation of historical data (1991-2023) from global climate models that aim to capture future vegetation dynamics. We will make the core findings clearer and more concise in the conclusion of a revised manuscript.

3. 35 – 43. Several references seem out of place in this paragraph. I suspect you mean Bjorkman et al. (2018) instead of Metcalfe et al. (2018), since Metcalfe et al. (2018) does not deal with the type of findings you describe at all, and Anne Bjorkman's paper does. Sturm et al. (2001) is a rather old and case-specific (albeit popular) reference for shrubification of the Arctic. ITEX papers (e.g. Elmendorf et al., 2012) or syntheses (e.g. Mekonnen et al., 2021; Martin et al., 2017; Myers-Smith et al., 2011) would be more appropriate.

Thank you for identifying misplacement of references by the reference management. This will be corrected in the revised version. We also appreciate the suggestion of more up-to-date references regarding shrubification.

4. 66 – 68, this proposed increase in nutrient availability under deeper snow is at odds with your statements in the abstract, results and discussion that shallower snowpacks should melt more slowly. It should be clear from the introduction onwards which snowpack properties can be expected to facilitate faster or slower melt, and how would relate to nutrient cycling. If the literature on the influence of snow dynamics on microbial turnover and nutrient availability is ambiguous in itself, then I would refrain from making any statements about nutrients as a mediating effect between snow dynamics and greenness.

Increased snow during the cold season could allow more vegetation growth in the following warm season, as more snow provides insulation, less frost damage and an increase in water availability. However, the rate of snowmelt is essential for efficient

meltwater percolation. This is particularly important in tundra regions, where soils are dry due to high drainage or low precipitation. In contrast, rapid melt will saturate the soil surface layer and run off. The interpretation of our results is that a considerable part of the dense/greenest vegetation is correlated with relatively shallow snowpacks and slow melt rates. This is particularly true in the southern ecoregions, where deep snowpacks are at higher elevations. Due to the decreasing trends in snow water equivalent (SWE_MAX) in the southern ecoregions, snowmelt starts earlier in the year, as they require less energy to melt, and the melt rates are also becoming slower. Ecoregion 2, which already has shallow snowpacks and does not show substantial trends in SWE_MAX, also displays a pattern of decreased snowmelt rates. We will expand on this complexity in a revised manuscript.

5. 111, here you mention the use of CryoClim data, that was chose to represent daily snow cover rather than the CARRA dataset. I do not see how this could be done since the data only goes to 2015, and this data product is not mentioned anywhere anymore in the remainder of the ms. Did you actually use it and if so, how? Perhaps it is a nice addition to incorporate data sources directly into Table 1 to resolve unclarities like this.

The assimilation of CryoClim data (ending in 2015) and, for example, MODIS (starting in early 2000s) on CARRA (from 1990 to present) is described in CARRA's Full System Documentation (Schyberg et al., 2020). The same often happens for automatic weather stations in Greenland. Although CryoClim has not been available since 2015, van der Schot et al. (2024) reports how CARRA performs against in situ measurements until 2023 across Greenland. Accuracy metrics are going to be provided as suggested in point 10.

6. 128-153: Can you give an indication of the match between AVHRRR and VIIRS? Calibration against MODIS does not seem to be the most relevant thing to mention here, since you do not use MODIS. See Miura et al. (2012), there seem to be some structural NIR differences and non-linear NDVI relationships between VIIRS and AVHRR?

There is no objective indication of the match between AVHRR NDVI and VIIRS NDVI computed by NOAA. AVHRR NDVI and VIIRS NDVI technical reports from NOAA as well as Miura et al. (2013), state different NIR and R bandwidths, additional to different algorithm corrections. As shown in Figure 2, AVHRR NDVI is generally lower than VIIRS NDVI. We attribute these differences in the first part of the time-series to less favourable environmental conditions for vegetation greening and NDVI retrievals. The linear relationships proposed by Miura et al. (2012) to correct AVHRR/2 ($y = 0.0412396 + 0.939953x$) and AVHRR/3 ($y = 0.0515123 + 0.872332x$), where y is VIIRS NDVI and x is AVHRR NDVI, would decrease the "corrected AVHRR NDVI" signal, consequently leading to higher differences between AVHRR and VIIRS NDVI. In order to get an indication whether interannual changes in green vegetation extent (more information about green vegetation extent in point 8) during VIIRS period are significantly different from the AVHRR period, we will statistically test the interannual changes in green vegetation extent for both periods.

**7. 143, why did you use an NDVI threshold of specifically 0.15?**

In contrast to other studies that either use the entire NDVI range or simply the maximum NDVI, we briefly explained in Section 3.1 that we only consider the NDVI that represents spectral greenness (NDVI > 0.15). While trends using the full NDVI range are heavily influenced by drying (NDVI transitioning from negative to positive values), trends using maximum NDVI in the high latitudes are likely to be influenced by temporal sampling artefacts (Myers-Smith et al. 2020).

In addition to Liu et al. (2024), who report NDVI at around 0.15 at the start of the growing season on Disko Island (in central West Greenland), we find that the United States Geological Survey states that "Areas of barren rock, sand, or snow usually show very low NDVI values (for example, 0.1 or less). Sparse vegetation such as shrubs and grasslands or senescing crops may result in moderate NDVI values (approximately 0.2 to 0.5)." We will add this more explicitly in a revised version.

**8. 146, can you provide a sharper definition of "interannual extent of vegetation"? To ecologists, this may be confusing since extent almost always refers to spatial extent.**

The interannual extent of vegetation refers to the variations in the area covered by green vegetation (NDVI > 0.15) over the study period (1991-2023). We apologize for the misunderstanding, as vegetation does not typically exhibit large changes in extent from year to year. What varies is whether the vegetation is greening up. To clarify the misunderstanding, we will specify and refer to it as 'interannual extent of green vegetation'.

**9. 147-155. It is very difficult for readers who are not intimately familiar with the AVHRR and VIIRS datasets to follow this paragraph, even though it is quite important for the quality of the results. Terms like "flag" and "n" may be unclear. Please provide more explicit description of exactly how the monthly max/mean/min nr. of valid pixels was used and how this translates to the CI's in Fig. 2. From reading this several times I still did not understand if any correction was applied before further analysis (and looking at Fig. S1 I would expect for that to be necessary).**

We plan to add a more comprehensive explanation of the procedure in the revised version:

To calculate the NDVI for each month, we started by averaging the NDVI retrievals that we obtained each month ($monthly\ NDVI = \frac{\sum_{i=1}^{n} NDVI_i}{n}$), when NDVI > 0.15. However, before 2014, the AVHRR algorithm was less strict in its data quality control compared to VIIRS from 2014 onwards, which results in more data points (n) before 2014. With n representing the total number of data points per month for NDVI calculation (see Figure S1 for n interannual variability), a higher n previously leads to lower monthly NDVI values.

To address temporal heterogeneities, we adjusted the data from the AVHRR period with the number of data points acquired during the VIIRS period. From 2014 to 2023, we identified the minimum, maximum, and average number of good quality data points for each summer month. Using these three numbers, we were able to generate a consistent variability range for calculating monthly NDVI. Hence, the NDVI values from 1991 to 2013 were recalculated by considering a similar reduction of data points as from 2014 to 2023. Figure 2 illustrates the effect of the range of NDVI values using

these recalculations to estimate the interannual vegetation extent. This procedure assumes that the environmental conditions influencing the number of data points between 1991 to 2013 are similar to those between 2014 and 2023.

10. 163, it would be useful to report an accuracy metric here.

Thank you for pointing this out. We will add an accuracy metric between CARRA SWE and in situ SWE for a set of locations representing south, east and west Greenland.

11. 169, why only from January onwards and not in autumn-winter previous year?

Thank you for reflecting on the definition of rain-on-snow days. Our intention with the provided period in this indicator is to solely consider rain-on-snow days preceding the onset of the thermal growing season. This means that our aim with the chosen period is to assess whether rain-on-snow could warm the snowpack and enhance early melting.

12. 169-171, I have a slight doubt about the way that the melt rate is calculated here. If this basically represents the time that passes between the peak SWE and moment of complete snowmelt, and peak SWE occurs early in the winter-spring season, how representative is this timeframe really for the spring melt season and water release? Especially if heavy snowfall occurs later in spring and is followed by warming, this automatically leads to a situation where deep snow appear to melt more rapidly. As a reader, it is hard to fully grasp how such nuances in the choice of processing influence the results.

(Heavy) Snowfall events can indeed happen after the SWE_MAX and may slow down the snowpack melt. We acknowledge that the snowpack melting rate does not follow a linear decline. What is important to retain here is that the snowpack is providing the soil with meltwater from the moment SWE_MAX is reached, even if the snowmelt rate is not a constant as estimated. An alternative way to improve this indicator that we intend to implement for the revised version is to calculate the mean of daily mean melt rates between SWE_MAX DOY and the onset of the growing season.

13. 176, you mention rain, but rainfall does not seem to be included as a bioclimatic variable as far as I can see (Table 1, Fig. 3), while snowfall was, and rain fraction too. You refer to Fig S10 for statements on the role of rain, but this figure refers to "solid precipitation" which suggests that this is about snow. Since you discuss the role of rain regularly, why not include rain (total summer season liquid precipitation) as a bio-climatic variable explicitly? This would make your conclusions and discussion points on the role of rain more explicit and justifiable.

We described in Section 3.2 that rain is a bioclimatic indicator used to calculate rain ratio. As CARRA allows mixed precipitation, a certain volume of rain that corresponds to a low rain ratio will have a minor influence on the snowpack. Therefore, we have considered the absolute volume of accumulated rain not as relevant as rain ratio. Additionally, rain in summer is common in the southern ecoregions and less

14. You could statistically back up your choice for PCA and its assumption of linear relations. You could do this by reporting axis lengths, for instance.

Thanks for the remark! We will add to Section 3.4: As we standardized all variables prior to analyses, we opted for unimodal and linear species response, as PCA is better suited for low variance, small gradients and more intuitive for the interpretation of the biplots.

15. Fig. 1, here results are presented, and completely new information comes in (NAO / GBI), so perhaps the figure should be presented later, in the results. I also miss a scale bar for greenness and it is unclear what "greenness" represents here (is this one the extent variables you calculated, or a mean, and are pixels < 0.15 included or not?).

We apologize for the misunderstanding. Monthly NDVI in Figure 1 is described in Section 3.1. Hopefully the explanation now on point 9 facilitates the interpretation of Figure 1. We opted to remove the values of the scale from these subpanels since they correspond to a 32-year average as it would not be useful for any further interpretation. However, the figure can be simplified and split into two. The delineation of the ecoregions and the greenness evolution during summer will remain in the Methods, as they will support the readers to understand the geography of the ecoregions and to recognise the greenness dynamics across Greenland from June to August. The remaining subpanels can be moved to Supplementary Material. That way we hopefully will increase readability.

16. 227-247 seem to be combined methods and results. The source for the climate oscillation data, and the rationale for including them, have not been properly covered earlier in the methods. It is also unclear how the use of oscillations relates to your study aim and research questions.

We argue, that the paragraph mentioned is a description of the variability among bioclimatic indicators across ecoregions. We find this section in the Methods to be the most suitable place to briefly make the readers aware that there is substantial variability among bioclimatic indicators across ecoregions by indicating a few statistics, and then in the Results we actually dive into the goals of the paper.

17. 250, Pedregosa et al does not seem like the most appropriate reference for the use of PCA. I advise to find papers that specifically deal with the considerations and strengths of using PCA in a pixel-based remote sensing context.

Thank you for the comment. We will add key references there (e.g., Pearson 1901, Lorenz et al. 1956) to refer to PCA and keep Pedregosa et al. for the sake of reproducibility.

18. 259, the use of Mann-Kendall tests is state of the art, but it appears that later on you only show results for growdays and greenness, not all bioclimatic factors as suggested here? Perhaps mention only growdays and greenness then?

We performed regression and the Mann-Kendall trend test to all bioclimatic indicators in study. However, due to limited space, we only displayed GrowDays and Greenness in the main manuscript. Although other bioclimatic trends are not shown their results are referred to throughout the manuscript (e.g., Fig S2 and S10). We can specifically add the name of the bio-climatic indicators used in Section 3.4 that

supported the interpretation of the results. This will include the additional supplementary figures used to back the results as requested in point 23.

19. 262, please explain the use of a 90% confidence interval rather than 95%. With the vast amount of pixels at your disposal, and the relatively long timespan of the study, I would expect that the generally accepted 95% CI would be fine and I would be curious to know why you deviated from this standard.

Given the power of the Mann-Kendall test for a limited sample size and the associated temporal variability, we applied the test level at 10% in order to increase the probability of correctly rejecting the null hypothesis, when it is false. However, we will decrease the test level to 5%, as commonly used in ecology.

20. 271-273, the statements made here need backing; how did you test whether significant long-term trends in vegetation extent were evident? Mann-Kendall test? Could sensor discrepancies play a role here?

Thank you for your comment. We will provide the results the Mann-Kendall trend test for vegetation extent separated for the AVHRR and VIIRS periods as well as the combined period in a (supplementary) table of the revised manuscript. Although the VIIRS period is rather short, we can nevertheless apply a statistical test to assess whether the two (independent) samples are significantly different, as we did for the interannual explained variability of the principal components across ecoregions (Fig. S4).

21. 275-279, reads like methods and introduces a whole new aspect of the methodology. I would also provide some more explanation of why the use of detrended Pearson correlations is an appropriate method to evaluate linearity assumptions for a PCA.

Thank you for your comment. Pearson correlation will be appropriately introduced in Section 3.4 of the revised manuscript. An explanation of why the use of detrended Pearson correlations is an appropriate method to evaluate linearity assumptions for a PCA will be included in the revised manuscript: Pearson correlation assumes that the data are stationary; that is, their statistical properties do not change over time. Therefore, detrending supports in transforming a non-stationary time series into a stationary one, where mean, variance, and autocorrelation structure should not change over time.

22. 290 & 296, you describe how specific variables were removed from analysis a priori. This is essential information that should go into methods, and it seems at odds with your earlier statement that variables were excluded from PCA based on contribution to cumulative explained variance. I would recommend to present a single, unambiguous criterium for the inclusion of variables into PCA and figures, in the methods. Especially since the identification of useful bio-climatic indicators was an explicit aim of the study.

Thank you for your comment. We will improve our wording. Our approach was first to calculate and analyse Pearson correlations among linearly-detrended bioclimatic indicators (Section 4.1). The strong Pearson correlations are described, and the weak

linear correlations (absolute linear correlations lower than 0.3) are removed before performing PCA. In this way, we assure that the bioclimatic indicators considered meet the linear relationship criterium for PCA. Also, bioclimatic indicators with redundant information (e.g., SnowDays and GrowDays or SoilWater and SoilIce in the same season).

23. 291-292 & L. 294-295, examples of interpretation of results, and no backing (figure, reference) provided to support these interpretations.
Thanks for pointing this out. This stems from an attempt to keep the manuscript concise. While we have the results ready, we did not include all of them in the manuscript. We will make sure to include the relevant information in the Supplementary Material.

24. 311-314, I had to read this section a few times to understand the rationale and approach. So if I read correctly, you applied the PCA for all years and ecoregions separately, and then tested whether the variances explained by PC1 and PC2 were similar across the two time periods. I am not fully sure how this would demonstrate that the two NDVI records are comparable and valid in this context. The variances may be similar, but the greenness dynamics, and the associations between different variables and PC axes may not be (do I understand this correctly)? Sidenote: a lot of this information again reads like methods and not results.
Thank you for the remark. The PCA is applied to all years for the main interpretation of the results (Fig. 4). In addition, and in order to assess whether changes in the satellite sensor impact the PCA result in Figure 4, we applied PCA interannually (Fig. S4). If the interannual variability of Greenness would not be properly retrieved by both sensors, its co-variability with the remaining components would be affected and reflected in the explained variance of each component. As we do not detect significant changes in the explained variance of the first two components, we inferred that the PCA outcome is comparable despite the sensor shift.

25. This is a nice figure! Also here, a scale bar for greenness would help the reader understand what kind of magnitudes we are talking about, across regions.
Thank you for appreciating our charts! We will add a scale bar in each subplot to display the range of greenness for the years between 1991 and 2023.

26. 318-319, "PC2 is heavily shaped by continentality, permafrost extent and precipitation patterns, meaning that snow-related indicators, like SWEMAX and MeltRate have the highest explanatory power". I struggle to see how your variables and methods could allow you to conclude anything about continentality or permafrost. This needs to be either backed up better, or (ideally) kept for the discussion. I also do not see how this means that snow related indicators are most important (snow is something different than permafrost and continentality?).
Our interpretation is based on the calculated climatology maps for vapour pressure deficit (measure of continentality), soil ice (measure of permafrost) and snowfall patterns from several seasons. The snowfall patterns are shaped by continentality as

snows amounts are relatively low (e.g., the interior of ecoregion 2). We will rephrase the above statement and move our interpretation to the discussion.

27. L 320 – 350 are altogether quite speculative and many of the claims here need to be supported either by a figure, statistics or literature (and in the latter case, it is better suited for the discussion). I would advise to back up your statements much more. And please carefully evaluate whether reported drivers are really drivers, or just represent the overall role of warming (e.g. increases in rainratio cannot really be teased apart from warming effects so I do not see how you would attribute change to rainfall patterns specifically, especially if total rainfall is not included in the analysis). I think this paragraph needs a thorough rewriting.

Thank you for your comment. We will thoroughly review our manuscript and back up our statements with evidence, either from our results or the literature. We will also rewrite the paragraph mentioned here.

28. 349-350, please consider how this relates to the aims of the study (oscillations are not introduced anywhere), report the approach in the methods, and report the test statistics either here or in the appendix.

Thank you for the comment. We will add the relevance of climate oscillations to the study in a revised version, stressing the weather pattern configuration that promoted the enhanced atmospheric warming of the past decades in Greenland. These weather pattern configurations coincide in a simplified manner with indices such as the North Atlantic Oscillation Index and the Greenland Blocking Index. In our view it was important to assess whether a certain principal component was largely shaped by climate oscillations.

29. 380, at this point the different terms used (here: spectral vegetation expansion) become a bit confusing. It would be nice to have a single, consistent term for each of the various manifestations of greening that you study in this paper, and present all of these early on.

Thank you for the notice. We will carefully and consistently review the used terms related to temporal and spatial vegetation change.

30. 382, see also major comments, here I was very unsure whether the differences in bandwidths and quality filtering might introduce artefacts into the comparison. Perhaps also good to remind the reader that 'greenness' here refers to the 0.15 threshold (related to comment above).

Thank you for the comment. We will remind the reader that Fig. 6c refers to years where the greenness threshold is met. We tested the sensitivity of the sub-periods' interval and the pattern in Fig. 6c did not significantly change, especially on showing that part of the greenness extent in ecoregion 2 did not change from the first to the second sub-period.

31. 417-420, How can you demonstrate that soil ice has an additional role, additive to warming and rainratio? Aren't they just all sides of the same coin? Could it also be, for instance, that the northern regions still feature most frozen ground conditions in

summer and that in southern regions, soils were already mostly above 0 degrees in the summer season, and that hence this dynamic is mostly evident in northern regions? I would carefully read this part of the discussion and evaluate which claims can be made with certainty, and which ones just reflect collinearity within the bio-climate variables.

The increase in SoilIce thawing in the northern ecoregions (latitudes above 70°N) is a combined consequence of warm and moist air advected from lower latitudes that contribute to warm and melt the surface (Niwano et al., 2021). Therefore, reductions in SoilIce in these ecoregions allow vegetation expansion that would not be possible with frozen soil. This is evident from the literature (e.g., Schmidt et al., 2019) that shows the rather reduced thermal growing season in 2018 with limited warm and moist air from lower latitudes.

32. Overall, the discussion would really benefit from a thematic subdivision, for instance into different sets of climate variables, or into driving mechanisms and a section on how they differ among regions? Right now the reader easily gets lost between different lines of argumentation.

Thank you for your suggestion. We will include more subsections in the Discussion to keep the line of argumentation clear and streamline the content.

33. 426 – 435, I found the descriptions of slower melt of shallower snowpacks very difficult to follow (and frankly, counterintuitive, but then I am not a snow physics expert). Even if the melt rate is lower, wouldn't the timing of complete snowmelt still be earlier for shallow snow than for deeper snow? What then is the exact role of the slower melt rate and potentially better water absorption within the context of your findings? I have a feeling that similar claims could be made about the role of deeper snow and its impact on soil temperature and microbial activity (as you also state in the introduction), so I am still in the dark about the role of melt rate in nutrient availability. I would recommend rewriting this in a way that is more accessible to readers without a background in snow physics and staying closer to your own results.

We hope that the improved explanation on point 4 clarifies the relationship between snow depth and snowmelt rate better. The referee is correct: in addition to slower snowmelt, the onset of the thermal growing season is generally occurring earlier. Instead of a quick snowmelt water runoff from relatively deeper snowpacks, the slow snowmelt water percolation in the soil can be used not only for microbial activity to generate nutrients but also directly by vegetation.

34. 428, Heijmans et al (2022) doesn't deal with the release of nutrients in relation to spring water availability. Perhaps we cite others in our review that have relevant findings on this topic, but to me this doesn't seem to be an appropriate reference here.

Thank you for the remark. Heijmans et al (2022) is a very relevant reference for us to better understand the links between tundra vegetation and permafrost, but indeed we should have cited the work of Salmon et al. (2016).

35. 463-465, maybe you can back up this hypothesis about the role of shrubs or potentially other species groups by checking your greenness trends against the CAVM or Karami et al. (2018)?

The recommended sources, the Circumpolar Arctic Vegetation Maps and Karami et al. (2018), are static maps based on the collection of data over several years with different approaches. We see the potential of the recommendation, but it is enough work for a publication on its own.

36. 475, what exactly do you mean by "validating bio-climatic indicators"? I think you could explain your proposed course of action a bit better, and also explain how that would help understand future trends.

Our study demonstrates how the co-variability among bio-climate indicators effectively clusters greenness. It is crucial that these indicators are accurately modelled in global climate models. If the historical period in global climate models does not accurately reflect the conditions of our study period, these models will be ineffective at predicting future spatial and temporal vegetation changes in the ice-free regions of Greenland. We will revise the manuscript to make this clearer.

37. The implications section reads like a rather surprising selection of several implications, of which I am not really sure if all the main ones are represented, and whether the ones that are now discussed most extensively are in fact the most important ones. For example, a lot of attention is dedicated to PBAPs and fog, but no mention is made of carbon dynamics or surface energy balance feedbacks. Even if this is deliberate, it would be good to highlight why specific implications are discussed while others are not. You do mention some of these aspects in the limitations, but they are of course also relevant from an implications perspective.

Thank you for the remark! We find relevant to keep recent literature that links primary biological aerosol particles (PBAPs) with the cloud formation in the Arctic and the potential of generating fog conditions due to decreasing sea ice as part of the Discussion. We mentioned a few other implications although not directly such as the feedback of the vegetation canopy on the surface feedback, shifts in cloudiness due to increased PBAPs, the cooling of the surface due to surface evaporation and ecological shifts on the animal community. However, we acknowledge that other important implications such carbon dynamics and surface albedo feedback have not been addressed in detail and we will include those in the revised manuscript.

38. 506 – 510, I would expect that such episodes of warm, humid conditions should be evident from your PCA analysis, so I do not see the point of mentioning the role of this particular episode as a limitation?

We agree that this was mentioned in a confusing manner, we will rewrite accordingly.

39. 517-520, needs references for the claims made. I would like to add that while permafrost thaw can indeed release moisture or lead to ponding, deeper thaw fronts also often lead to deeper infiltration and surface drying (Liljedahl et al., 2016). This section could use more nuance and backing.

Thank you for mentioning the possibility of deep thaw fronts which could lead to deep infiltration and surface drying as described by Liljedahl et al. (2016). Indeed, the complexity of these feedbacks is high and we will provide a more nuanced perspective in a revised version.

40. 525 – 530, I do not want to send you back to the drawing board, but I am interested why elevation was added to your analysis, while aspect and slope were not. You rightfully stress their importance and I would (perhaps naively!) assume that it would not be such an enormous effort to include them in your analysis as well?

Good point and indeed, for a wide perspective it would be useful to add slope and aspect for which we have the data and the analysis ready. We will add these results to the revised version.

41. Rather than reiterate what you did, you could summarize the actual findings and try to align better with the original aims (perhaps mention which set of variables or which variables show the strongest associations?) and mention the key advance you have made? This would make the conclusion more informative.

Thank you for the suggestion. We will briefly summarize our findings and keys advancements in the field in the revised version.

**Technicalities & Language**
1. 10 "summer spectral vegetation". This is an unusual term, it would be good to rephrase it or explain it so that there can be no ambiguity about what it means.
2. 18 "by 22.5% increase" should be "by 22.5%". I also recommend to be more explicit about what you mean by "the distribution of vegetation". Do you mean that the vegetated area of Greenland (determined here as summer NDVI > 0.15?) expanded in area by 22.5%? Perhaps you want to rewrite this sentence.
3. 25, what do you mean by "regional Greenland"? Perhaps that specific regions of Greenland are warming three times faster.
4. 31, add "and" instead of comma between "composition" and "alterations".
5. 48, is it really necessary to mention the specific methods of Gamm et al ( "using [..], [...] and [...]")? This is not done for other papers that you cite?
6. 53-55, this reads like a repetition of L. 43-44.
7. 62, I do not think "snow cover melt" is a very generally used term. Maybe write "snowmelt timing" or "snow melt rate", depending on what you mean exactly?
8. 71, maybe write "large amounts of snow" rather than "large amounts of snow coverage", since from what I understand snowpacks were also very deep, not just spatially extensive.
9. 81-82, example of a grammatically confusing sentence.
10. 83-86, implications for phytoplankton seem beyond the scope of your study system and I do not see the added value of discussing it here (it seems more of an implication rather than an example of the importance of subsurface flow to terrestrial vegetation).
11. 105, add "the" between "to" and "CARRA".
12. 132, "and thereafter is then continued" should be "and is thereafter continued".
13. 133, add "is" between "mask" and "spectrally".

14. Figure 5) Final sentence in the caption: Do you mean that the trend was considered significant if the 90% CI of the estimate did not overlap 0? This is what I am used to. Similar for Fig. 6
15. 376, replace "evidence" with "shows"?
16. Table 2) perhaps a no brainer, but it would be good to explain what the fraction mean; is this % of total area of that ecoregion?
17. 446, change "favourable areas" into "a more favourable area".
18. 498, change "as" into "as in"

Thank you for these valuable edits which we will incorporate in the revised manuscript.

**References**
Martin, A. C., Jeffers, E. S., Petrokofsky, G., Myers-Smith, I., & Macias-Fauria, M. (2017). Shrub growth and expansion in the Arctic tundra: an assessment of controlling factors using an evidence-based approach. Environmental Research Letters, 12(8), 085007.
Bjorkman, A. D., Myers-Smith, I. H., Elmendorf, S. C., Normand, S., Rüger, N., Beck, P. S., ... & Weiher, E. (2018). Plant functional trait change across a warming tundra biome. Nature, 562(7725), 57-62.
Miura, T., Turner, J. P., & Huete, A. R. (2012). Spectral compatibility of the NDVI across VIIRS, MODIS, and AVHRR: An analysis of atmospheric effects using EO-1 Hyperion. IEEE Transactions on Geoscience and Remote Sensing, 51(3), 1349-1359.
Liljedahl, A. K., Boike, J., Daanen, R. P., Fedorov, A. N., Frost, G. V., Grosse, G., ... & Zona, D. (2016). Pan-Arctic ice-wedge degradation in warming permafrost and its influence on tundra hydrology. Nature Geoscience, 9(4), 312-318.
Mekonnen, Z. A., Riley, W. J., Berner, L. T., Bouskill, N. J., Torn, M. S., Iwahana, G., ... & Grant, R. F. (2021). Arctic tundra shrubification: a review of mechanisms and impacts on ecosystem carbon balance. Environmental Research Letters, 16(5), 053001.

References:

Bengtsson, L., Andrae, U., Aspelien, T., Batrak, Y., Calvo, J., de Rooy,W., Gleeson, E., Hansen-Sass, B., Homleid, M., Hortal, M., Ivarsson, K.-I., Lenderink, G., Niemelä, S., Nielsen, K. P., Onvlee, J., Rontu, L., Samuelsson, P., Muñoz, D. S., Subias, A., Tijm, S., Toll, V., Yang, 575 X., and Køltzow, M. Ø.: The HARMONIE–AROME model configuration in the ALADIN–HIRLAM NWP system, Monthly Weather Review, 145, 1919–1935, https://doi.org/10.1175/MWR-D-16-0417.1, 2017.

Liu, Y.,Wang, P., Elberling, B., and Westergaard-Nielsen, A.: Drivers of contemporary and future changes in Arctic seasonal transition dates for a tundra site in coastal Greenland, Global Change Biology, 30, e17 118, https://doi.org/10.1111/gcb.17118, 2024

Lorenz, E. N. (1956). Empirical orthogonal functions and statistical weather prediction (Vol. 1, p. 52). Cambridge: Massachusetts Institute of Technology, Department of Meteorology.

Madson, A., Dimson, M., Fortini, L. B., Kawelo, K., Ticktin, T., Keir, M., ... & Gillespie, T. W. (2023). A near four-decade time series shows the Hawaiian Islands have been browning since the 1980s. Environmental Management, 71(5), 965-980. https://doi.org/10.1007/s00267-022-01749-x

Masson, V., Le Moigne, P., Martin, E., Faroux, S., Alias, A., Alkama, R., ... and Voldoire, A. (2013). The SURFEXv7. 2 land and ocean surface platform for coupled or offline simulation of earth surface variables and fluxes. Geoscientific Model Development, 6(4), 929-960.

Musselman, K. N., Clark, M. P., Liu, C., Ikeda, K., and Rasmussen, R.: Slower snowmelt in a warmer world, Nature Climate Change, 7, 214–219, https://doi.org/10.1038/nclimate3225, 2017.

Myers-Smith, I. H., Kerby, J. T., Phoenix, G. K., Bjerke, J. W., Epstein, H. E., Assmann, J. J., John, C., Andreu-Hayles, L., Angers-Blondin, S., Beck, P. S., Berner, L. T., Bhatt, U. S., Bjorkman, A. D., Blok, D., Bryn, A., Christiansen, C. T., Cornelissen, J. H. C., Cunliffe, A. M., Elmendorf, S. C., Forbes, B. C., Goetz, S. J., Hollister, R. D., de Jong, R., Loranty, M. M., Macias-Fauria, M., Maseyk, K., Normand, S., Olofsson, J., Parker, T. C., Parmentier, F.-J. W., Post, E., Schaepman-Strub, G., Stordal, F., Sullivan, P. F., Thomas, H. J. D., Tømmervik, H., Treharne, R., Tweedie, C. E., Walker, D. A., Wilmking, M., and Wipf, S.: Complexity revealed in the greening of the Arctic, Nature Climate Change, 10, 106–117, https://doi.org/10.1038/s41558-019-0688-1, 2020

Niwano, M., Box, J. E., Wehrlé, A., Vandecrux, B., Colgan, W. T., & Cappelen, J. (2021). Rainfall on the Greenland ice sheet: Present-day climatology from a high-resolution non-hydrostatic polar regional climate model. Geophysical Research Letters, 48(15), e2021GL092942, https://doi.org/10.1029/2021GL092942

Pearson, K. (1901). LIII. On lines and planes of closest fit to systems of points in space. The London, Edinburgh, and Dublin philosophical magazine and journal of science, 2(11), 559-572 , https://doi.org/10.1080/14786440109462720

Pourmohamad, Y., Abatzoglou, J. T., Belval, E. J., Fleishman, E., Short, K., Reeves, M. C., ... & Sadegh, M. (2024). Physical, social, and biological attributes for improved understanding and prediction of wildfires: FPA FOD-Attributes dataset. Earth System Science Data, 16(6), 3045-3060. https://doi.org/10.5194/essd-16-3045-2024

Salmon, V. G., Soucy, P., Mauritz, M., Celis, G., Natali, S. M., Mack, M. C., & Schuur, E. A. (2016). Nitrogen availability increases in a tundra ecosystem during five years of experimental permafrost thaw. Global Change Biology, 22(5), 1927-1941.

Silva, T., Abermann, J., Noël, B., Shahi, S., van de Berg, W. J., and Schöner, W.: The impact of climate oscillations on the surface energy budget over the Greenland Ice Sheet in a changing climate, The Cryosphere, 16, 3375–3391, https://doi.org/10.5194/tc-16-3375-2022, 2022.

Schmidt, N. M., Reneerkens, J., Christensen, J. H., Olesen, M., and Roslin, T.: An ecosystem-wide reproductive failure with more snow in the Arctic, PLoS Biology, 17, e3000 392, https://doi.org/10.1371/journal.pbio.3000392, 2019.

Schyberg, H., Yang, X., Køltzow, M., Amstrup, B., Bakketun, m., Bazile, E., Bojarova, J., Box, J. E., Dahlgren, P., Hagelin, S., Homleid, M., Horányi, A., Høyer, J., Johansson, m., Killie, 750 M., Körnich, H., Le Moigne, P., Lindskog, M., Manninen, T., Nielsen, E. P., Nielsen, K., Olsson, E., Palmason, B., Peralta, A. C., Randriamampianina, R., Samuelsson, P., Stappers, R., Støylen, E., Thorsteinsson, S., Valkonen, T., and Wang, Z.: Arctic regional reanalysis on single levels from 1991 to present. Copernicus Climate Change Service (C3S) Climate Data Store (CDS), https://doi.org/10.24381/cds.713858f6, accessed on 15-12-2022, 2020.

Skakun, S., Justice, C. O., Vermote, E., and Roger, J.-C.: Transitioning from MODIS to VIIRS: an analysis of inter-consistency of NDVI data sets for agricultural monitoring, International Journal of Remote Sensing, 39, 971–992, https://doi.org/10.1080/01431161.2017.1395970, 2018.

Vermote, E., Justice, C., Csiszar, I., Eidenshink, J., Myneni, R., Baret, F., Masuoka, E., Wolfe, R., Claverie, M., and Program, N. C.: NOAA Climate Data Record (CDR) of Normalized Difference Vegetation Index (NDVI), Version 5, https://doi.org/10.7289/V5ZG6QH9, access date: 2022-05-06, 2018.

Vermote, E., Franch, B., Roger, J.-C., Murphy, E., Becker-Reshef, I., Justice, C., Claverie, M., Nagol, J., Csiszar, I., Meyer, D., Baret, F., Masuoka, E., Wolfe, R., Devadiga, S., Villaescusa, J., and Program, N. C.: NOAA Climate Data Record (CDR) of Surface Reflectance, Version 1, https://doi.org/10.25921/gakh-st76, access date: 2023-07-06, 2022.

Wilks, D. S. (2011). Statistical methods in the atmospheric sciences. Academic press.

---

## Author Comment (AC2)

We would like to thank the editor for the opportunity to publish our manuscript at EGUsphere and to the nine referees for their thorough evaluations with constructive comments that will improve the manuscript greatly. In the following, we will address the referees' comments point by point. We mark the comments given by the referee in red, provide our answers and comments in black, and indicate how we will address the amendments in the manuscript that we plan to submit upon editor's decision in green. Note, that we add the concrete amendments planned not at all points as this in parts make the replies less readable and only where we will add/adapt a section directly and concisely. In many parts we only conceptually explain how we will adapt content, while the exact implementation remains until the revised version.

We would like to announce that should the manuscript advance to the revision stage; Verena Haring from the Department of Biology at the University of Graz will be included as a co-author in recognition of her valuable assistance with the point-by-point responses.

Tiago Silva, on behalf of all co-authors.

**RC2**: 'Comment on egusphere-2024-2571', Anonymous Referee #2, 21 Oct 2024

Dear Silva et al. & the editors of Copernicus Biogeosciences,
Thank you for the invitation to review this manuscript. It's a great privilege to contribute to our scientific community. Please see the text below for my review of the manuscript, "Bio-climatic factors drive spectral vegetation changes in Greenland" by Tiago Silva et al. This study seeks to identify bioclimatic drivers of changes in greenness and greenness distribution across Greenlands ice-free terrestrial ecosystem. Understanding the impacts of climate change on this ecosystem is extremely important, particularly in the context of recent studies highlighting changes in vegetation ("Arctic greening") and permafrost dynamics. The authors do a good job summarizing the major points of current literature in these regions and highlighting the importance of their study.
The authors seek to assess these drivers by combining remotely sensed NDVI as observed from AVHRR and VIIRS between 1991 - 2023 with a gridded climate data set, the Copernicus Arctic Regional Reanalysis (CARRA). The authors use Principal Component Analysis (PCA) to identify correlations between "greenness" and a matrix of bioclimatic variables. Additionally, they use non-parametric methods to identify trends in bio-climatic indicators and assess their directionality in magnitude over time across 5 sensibly delineated ecoregion across the terrestrial Greenland ecosystem.

**General Comments**
I commend Silva et al. for their ambitious analysis of a substantial amount of data from a sensitive ecosystem of broad scientific interest. For this reason, it is my opinion that the study's aim is well suited for the readership of Copernicus Biogeosciences and is an important undertaking. However, I have major concerns about the implementation of methods and the interpretation of results. Most importantly, there is a critical misalignment between the stated goals of the study and the methods used to achieve these goals (as well as the title of the paper).

To summarise my concerns: The authors sought "to gain a deeper understanding of the spatio-temporal patterns of spectral vegetation changes across ice-free regions of Greenland (ln 90)" and "examine the combined effects of bio-climatic indicators ranging from sub-surface factors (such as soil water availability) to above-surface factors (such as the thermal growing season, heat stress, and frost) with summer spectral greenness (ln 91 - 95)." However, the authors provide contradictory statements about the goals of the PCA. Throughout the paper they explicitly state that they use PCA to assess drivers of *changes* in NDVI over time within a pixel, as well as having used PCA to assess drivers in changes of greenness *distribution*. Reviewing the methods and results of the PCA, it seems that the dimensionality reduction algorithm was actually used to assess bio-climate indicators that correlate with average summer spectral greenness ("greenness distribution"). I elaborate on these concerns below.

**Specific comments**
Major Concerns
Thank you for the comprehensive statement. We will break the explanation of point 1 into several sub-points.
1) There is a critical misalignment between the stated goals of the study and the methods used. In Section 3.4, the authors mention that "PCA was used to investigate the combined influence among bio-climatic indicators on summer greenness _changes_" (ln 249-250; _emphasis added_). However, in the Results section, it is stated that "PCA was used to investigate the combined influence among bio-climatic indicators with summer greenness" (ln 307), which suggests an analysis of greenness levels rather than _changes_ in spectral greenness.
This is a very valid point, and we apologize for having caused confusion. Our intention was the same in both sentences. We recall that the PCA encompasses data since 1991 to 2023. This means that the PCA outcome is a statistical result of the biosphere-atmosphere-cryosphere interaction on the course of the three decades in our study. By colouring each score with its corresponding greenness, we show that the densely vegetated/greenest regions are clustered by the first two principal components because of "the combined influence among bio-climatic indicators with summer greenness". This result is only achieved by considering the spatio-temporal changes of all bio-climatic indicators, where greenness is included.
This discrepancy is further supported by the caption for Figure 4, which notes that the biplots' scores "are colour-coded based on the summer spectral greenness as in Figure 1," where spectral greenness is defined as the "averaged spectral greenness (based on the period 1991-2023) for June, July, and August."
We apologize for the misunderstanding caused in this specific sentence from the Figure 4 caption. The caption was not meant to indicate that the scores are color-coded based on the summer averaged greenness of 1991-2023, but rather to direct the reader to the greenness scale for better interpretation of the colormap used. In Figure 1 the reader would also recall the omittance of the scale of greenness. Therefore, we will include scales in each sub-panel of Figure 4 and adapt the caption.
Additionally, greenness is included in the PCA but defined differently as "seasonally averaged monthly NDVI," a quantity briefly mentioned in Section 3.1.

Monthly averaged NDVI is used in Figure 1 to show the evolution of spectral greenness in summer in each ecoregion.

In Section 3.1 we state that "we calculated a seasonally averaged NDVI, hereafter referred to as spectral greenness and interchangeably as vegetation." We will revise our text to make the methodology clearer by changing the description in Section 3.2 to: Spectral greenness was compiled for summer months, in order to capture the period with maximum solar radiation in Greenland and to avoid snow cover.

The authors highlight that PC1 and PC2 "largely capture and explain Greenness distribution" (ln 320-321), suggesting a focus on greenness levels rather than changes.

The biplot shows that the scores with high spectral greenness are grouped in areas of low elevation (PC1) with varying degrees of influence in precipitation and snow patterns (PC2). Therefore, the outcome of our PCA not only captures where vegetation prevails but also where vegetation, especially at low laying areas, develops.

It may be possible that the inclusion of "changes" in lines 249-250 was unintentional. However, the broader context suggests that the issue extends beyond a simple wording error. The title ("Bio-climatic factors drive spectral vegetation changes in Greenland"), the abstract (ln 10-15: "GrowDays... emerged as the pivotal factor across all ecoregions...to promote vegetation growth."), and the discussion (e.g., ln 417-419: "Our [PCA] results suggest that in the northern ecoregions, the reduction in soil ice during summer...is enabling vegetation growth, leading to northward expansion of vegetation."

Thanks for highlighting this most important confusion. Indeed, we look both at the state as well as the changes. Our Results aim to investigate spatio-temporal changes in Greenness and its co-variability with atmospheric and snow indicators based on values from 1991 to 2023. The monthly averaged Greenness state is only shown in Figure 1. We will revise our text to make this clearer.

and ln 433-435: "The combined effect of soil nutrients with increased soil water availability in spring (SoilWaterMAM) and T2mMAM, promotes early plant growth. Therefore, leaves are more developed in early summer, which in association with increased T2mJJA and longer periods of solar radiation, allow for greener vegetation.") all imply a focus on changes in greenness values over time.

Our interpretations will be supported by additional references in the revised version.

As the analysis currently stands, PCA is used to assess the variation in climate variables, which is then visually compared to average summer greenness from 1991-2023 with biplots. Separately, the authors explore trends in vegetation expansion using Mann-Kendall tests and thresholds of NDVI between two discrete periods (1991 – 2007 and 2008 - 2023). Despite a lack of generative or predictive models linking these two goals, the authors then interpret PCA loading vectors as "explaining" changes in greenness and greenness distribution. It is also not clear to me how the authors made these interpretations; I speculate this was done by visual comparison of the maps of PCs in the supplementary material with the maps of greenness distribution and greenness change over time in Figure 6.

We hope that with the explanations provided above, it is clearer to the referee how PCA and trend analysis were interpreted. The use of generative or predictive models goes beyond the scope of this study. For that, we would need the output of models

considering sub-surface processes, for example, vegetation and microbial dynamics, to better cover essential aspects for quantitative greenness distribution and the associated changes.

We will combine information of several figures (e.g., Fig. 4 and Fig.6c) to reinforce our results and interpretations.

2) Loading vectors should not be interpreted causally in the way the authors have. While it is true that alignment between two loading vectors indicate correlation and orthogonal vectors are uncorrelated, PCA is a function purely on a matrix of features without explicit regard for response variables. Since PCA is generally used for dimensionality reduction, data compression, or exploratory analysis, its application to infer causal relationships between bio-climatic factors and greenness requires further qualification. If the goal is to assess the relative importance of climate variables on changes in greenness, a causal (or at least an interpretable predictive) model is required.

Thank you for the remark. We agree that Greenness is not a response variable due to the reasons mentioned in point 1. Indeed, causal interpretation is difficult and not justified. Wherever we find evidence that is in line with literature, we will in the revised manuscript more carefully connect the evidence of the PCA with reasoning based on literature.

3) The inclusion of "seasonally averaged" spectral greenness as a feature in the PCA and then coloring the scores in the biplots of Figure 4 based on average summer spectral greenness over the growing seasons (1991-2023) raises concerns about circular reasoning. Further clarification on how this aspect was handled could help alleviate these concerns.

This is a misinterpretation of our analysis, as the PCA is performed for all pixels available in every ecoregion between 1991 and 2023. The corresponding colouring indicates the Greenness of each pixel in a particular year. The co-variability of Greenness with the remaining components is shown on loading vector 6 and the colouring of the scores helps to better understand how Greenness is distributed along PC1 and PC2. We will rewrite the caption and implement scalebars in every subpanel to avoid misunderstandings.

4) Generally, the methods are not described in enough detail. In addition to my confusion about the methods as described above:
4a) I agree with a note from another reviewer, the calibration procedure addressing potential systematic biases between AVHRR and VIIRS NDVI should be elaborated.

The NOAA Climate Data Record (CDR) of AVHRR NDVI - Version 5 and the NOAA CDR of VIIRS NDVI - Version 1 are developed by Eric Vermote and colleagues (Vermote et al. 2018 and 2022) for NOAA's CDR Program. Both records have been processed considering the same atmospheric features as in Miura et al. (2012) and both processed records are posterior to Miura et al. (2012) proposed correction. Also, the correction proposed by Miura et al. (2012) is not assessed in polar regions, which may contribute to additional uncertainties.

We follow similar approaches of recent literature (e.g., Madson et al. 2023, Pourmohamad et al. 2024) that make use of the full AVHRR NDVI and VIIRS NDVI without additional corrections. As stated in Section 2.2, Vermote and colleagues for NOAA's CDR Program use MODIS to spectrally calibrate AVHRR (Vermote et al., 2018) and VIIRS (Skakun et al., 2018). As NOAA does not provide an overlapping period for AVHRR and VIIRS, we are unable to compare both processed products and quantify biases in polar regions. Nevertheless, we will make sure that we add to the discussion that the potential mismatches between AVHRR and VIIRS NDVI products cannot be discarded and, in a revised version we will provide the greenness trends before and after the sensor change in order to assess potential mismatches between sensors, bearing in mind that differences can also rise from other sources such as the interannual variability of the atmospheric conditions before and after 2014.

**4b) The calculation of "seasonally averaged NDVI" is somewhat unclear. I assume this involves averaging monthly NDVI across the growing season, but further explanation would be helpful.**

Yes, the Spectral Greenness is seasonally averaged monthly NDVI, as described in Section 3.1 and 3.2. We plan to add a more comprehensive explanation of the procedure in the revised version, for instance:

To calculate the NDVI for each month, we started by averaging the NDVI retrievals that we obtained each month ($monthly\ NDVI = \frac{\sum_{i=1}^{n} NDVI_i}{n}$), when NDVI > 0.15. However, before 2014, the AVHRR algorithm was less strict in its data quality control compared to VIIRS from 2014 onwards, which results in more data points (n) before 2014. With n representing the total number of data points per month for NDVI calculation (see Figure S1 for n interannual variability), a higher n previously leads to lower monthly NDVI values.

To address temporal heterogeneities, we adjusted the data from the AVHRR period with the number of data points acquired during the VIIRS period. From 2014 to 2023, we identified the minimum, maximum, and average number of good quality data points for each summer month. Using these three numbers, we were able to generate a consistent variability range for calculating monthly NDVI. Hence, the NDVI values from 1991 to 2013 were recalculated by considering a similar reduction of data points as from 2014 to 2023. Figure 2 illustrates the effect of the range of NDVI values using these recalculations to estimate the interannual vegetation extent. This procedure assumes that the environmental conditions influencing the number of data points between 1991 to 2013 are similar to those between 2014 and 2023.

**4c) Given the potential impact of cloud cover and other factors on NDVI observations, more information on how observation frequency (described as "n" in Section 3.1) was used to assess uncertainty and uneven sampling would strengthen the analysis. This seems like it was at least tangentially covered given the brief mention of this in Section 3.1 and the first figure in the Supplementary Materials -- but more explanation of the procedures is needed.**

As mentioned in Section 2.2, the climate data record for AVHRR and VIIRS NDVI is available on a daily basis. In addition to the area with the total number of available pixels (Fig. S1), we will provide the already generated maps of the 32-year average and

standard deviation of annual number of available observations across ice-free Greenland.

4d) More details on the PCA and Mann-Kendall implementations would also be valuable. For example, when using scikit-learn for PCA, describing the optimizer and input data shape would help ensure transparency, as some solvers are better optimized for particular data configurations. Similarly, the choice of the standard Mann-Kendall test variant in pyMannKendall should be justified, especially regarding serial autocorrelation, which is an important consideration in trend analysis. While MK tests are the current state of the art for landscape-scale analysis like this, pyMannKendall offers options that seek to account for autocorrelation, and discussing whether this was assessed in the data would clarify the robustness of the trend analysis.

Thank you for the request. We did not consider relevant the inclusion of the optimizer and input data shape, but we will promptly add it to the revised manuscript.

We did not consider autocorrelation for the atmospheric and snow variables from summer to summer, but we acknowledge that it should have been considered for greenness. We will address this in the revision.

**Minor Concerns & Technical Corrections.**
In addition to minor concerns pointed out by another reviewer, there are some instances of speculation that are not supported by the PCA analysis in the results section which should removed, or moved to the discussion section and include citations. These are also specific examples of where I think a inappropriate causal interpretation of loading vectors has occurred (Major Concern 2). For example:

- (ln 326) "The decreasing trend of snow rates (SnowDJF and SnowMAM) has led to SWEMAXDOY to occur earlier. Despite the increasing trend in T2mMAM, the still-low solar elevation and the still-low near-surface air-temperatures result in low melting rates of the snowpack (MeltRate). These slow melt rates favour slow meltwater percolation (SoilWaterMAM loading vector opposite to MeltRate loading vector)."
I think this sentence can remain in the Results since it comes as an interpretation of the resulting seasonal accumulated snow, SWE_MAX DOY, MeltRate and SoilWaterMAM trend maps, together with the PCA loading vectors. We will add all the necessary trend maps supporting our results into the Supplementary Material of the revised manuscript.

- (ln 329) "Additionally, the earlier onset of the thermal growing season allows vegetation to produce energy via photosynthesis, particularly in the ecoregions in lower latitudes with adequate 330 sun exposure (Onset loading vector opposite to Greenness loading vector)."
Thank you for the remark. This sentence will be moved to the Discussion.

- (ln 333) "Therefore, increases of RainRatioJJA promote high greenness (aligned loading vectors), as vegetation in such environmentally harsh places likely developed mechanisms to effectively retain/absorb liquid water whenever possible."

Thank you for the remark. This sentence will be moved to the Discussion.

This sentence is a tautological argument:
(ln 465) "The wide-spread summer spectral greening occurs as a result of greener vegetation as certain sites."
Greening shown in Fig. 6a) occurs for several reasons. In this sentence from the Discussion, we meant to say that greening is essentially occurring in regions that already comprise vegetation (white regions in Figure 6c), whereas in other regions greening is observed due to green vegetation expansion (turquoise regions in Figure 6c).

The importance of solar radiation exposure is described as important in several places, including the conclusions, but are not included explicitly in the PCA or other analyses (ln 327, 435, 534).
The exposure to solar radiation is not considered as the NDVI is only available when there is solar exposure. However, we make use of relevant metrics of the atmospheric circulation patterns in the vicinity of Greenland that promote cloudless conditions (e.g., positive phase of the Greenland Blocking Index, GBI). That is why in Figure 1, we correlated summer greenness with summer GBI, where we report high positive correlations across all ecoregions. Therefore, we cannot discard the role of the interannual variability of atmospheric circulation patterns on greenness, as the previous decade was composed by more frequent cloudless conditions in summer (Silva et al. 2022). We will include additional references in the Discussion regarding this aspect.

Figure 4 - It would be helpful to readers if the PC1 axis was flipped for Ecoregion 2 and 4 so that the quadrants with higher greenness scores were all in the same vicinity in the biplots across Ecoregions.
Thanks for the remark! We will flip the axis for the same orientation across ecoregions.

The color palettes in Figures 5 and 6 rely on a reader's ability to distinguish red and green, which is a common color-blindness.
The colormaps were checked prior to submission following the Copernicus manuscript preparation style and Coblis – Color Blindness Simulator. All figures are supposed to be colour-blind friendly, except for monochromacy.

Grammar checks needed throughout.
Thanks for the remark! We will revise and improve the grammar.

References:

Bengtsson, L., Andrae, U., Aspelien, T., Batrak, Y., Calvo, J., de Rooy,W., Gleeson, E., Hansen-Sass, B., Homleid, M., Hortal, M., Ivarsson, K.-I., Lenderink, G., Niemelä, S., Nielsen, K. P., Onvlee, J., Rontu, L., Samuelsson, P., Muñoz, D. S., Subias, A., Tijm, S., Toll, V., Yang, 575 X., and Køltzow, M. Ø.: The HARMONIE–AROME model

configuration in the ALADIN–HIRLAM NWP system, Monthly Weather Review, 145, 1919–1935, https://doi.org/10.1175/MWR-D-16-0417.1, 2017.

Liu, Y.,Wang, P., Elberling, B., and Westergaard-Nielsen, A.: Drivers of contemporary and future changes in Arctic seasonal transition dates for a tundra site in coastal Greenland, Global Change Biology, 30, e17 118, https://doi.org/10.1111/gcb.17118, 2024

Lorenz, E. N. (1956). Empirical orthogonal functions and statistical weather prediction (Vol. 1, p. 52). Cambridge: Massachusetts Institute of Technology, Department of Meteorology.

Madson, A., Dimson, M., Fortini, L. B., Kawelo, K., Ticktin, T., Keir, M., ... & Gillespie, T. W. (2023). A near four-decade time series shows the Hawaiian Islands have been browning since the 1980s. Environmental Management, 71(5), 965-980. https://doi.org/10.1007/s00267-022-01749-x

Masson, V., Le Moigne, P., Martin, E., Faroux, S., Alias, A., Alkama, R., ... and Voldoire, A. (2013). The SURFEXv7. 2 land and ocean surface platform for coupled or offline simulation of earth surface variables and fluxes. Geoscientific Model Development, 6(4), 929-960.

Musselman, K. N., Clark, M. P., Liu, C., Ikeda, K., and Rasmussen, R.: Slower snowmelt in a warmer world, Nature Climate Change, 7, 214–219, https://doi.org/10.1038/nclimate3225, 2017.

Myers-Smith, I. H., Kerby, J. T., Phoenix, G. K., Bjerke, J. W., Epstein, H. E., Assmann, J. J., John, C., Andreu-Hayles, L., Angers-Blondin, S., Beck, P. S., Berner, L. T., Bhatt, U. S., Bjorkman, A. D., Blok, D., Bryn, A., Christiansen, C. T., Cornelissen, J. H. C., Cunliffe, A. M., Elmendorf, S. C., Forbes, B. C., Goetz, S. J., Hollister, R. D., de Jong, R., Loranty, M. M., Macias-Fauria, M., Maseyk, K., Normand, S., Olofsson, J., Parker, T. C., Parmentier, F.-J. W., Post, E., Schaepman-Strub, G., Stordal, F., Sullivan, P. F., Thomas, H. J. D., Tømmervik, H., Treharne, R., Tweedie, C. E., Walker, D. A., Wilmking, M., and Wipf, S.: Complexity revealed in the greening of the Arctic, Nature Climate Change, 10, 106–117, https://doi.org/10.1038/s41558-019-0688-1, 2020

Niwano, M., Box, J. E., Wehrlé, A., Vandecrux, B., Colgan, W. T., & Cappelen, J. (2021). Rainfall on the Greenland ice sheet: Present-day climatology from a high-resolution non-hydrostatic polar regional climate model. Geophysical Research Letters, 48(15), e2021GL092942, https://doi.org/10.1029/2021GL092942

Pearson, K. (1901). LIII. On lines and planes of closest fit to systems of points in space. The London, Edinburgh, and Dublin philosophical magazine and journal of science, 2(11), 559-572 , https://doi.org/10.1080/14786440109462720

Pourmohamad, Y., Abatzoglou, J. T., Belval, E. J., Fleishman, E., Short, K., Reeves, M. C., ... & Sadegh, M. (2024). Physical, social, and biological attributes for improved understanding and prediction of wildfires: FPA FOD-Attributes dataset. Earth System Science Data, 16(6), 3045-3060. https://doi.org/10.5194/essd-16-3045-2024

Salmon, V. G., Soucy, P., Mauritz, M., Celis, G., Natali, S. M., Mack, M. C., & Schuur, E. A. (2016). Nitrogen availability increases in a tundra ecosystem during five years of experimental permafrost thaw. Global Change Biology, 22(5), 1927-1941.

Silva, T., Abermann, J., Noël, B., Shahi, S., van de Berg, W. J., and Schöner, W.: The impact of climate oscillations on the surface energy budget over the Greenland Ice Sheet in a changing climate, The Cryosphere, 16, 3375–3391, https://doi.org/10.5194/tc-16-3375-2022, 2022.

Schmidt, N. M., Reneerkens, J., Christensen, J. H., Olesen, M., and Roslin, T.: An ecosystem-wide reproductive failure with more snow in the Arctic, PLoS Biology, 17, e3000 392, https://doi.org/10.1371/journal.pbio.3000392, 2019.

Schyberg, H., Yang, X., Køltzow, M., Amstrup, B., Bakketun, m., Bazile, E., Bojarova, J., Box, J. E., Dahlgren, P., Hagelin, S., Homleid, M., Horányi, A., Høyer, J., Johansson, m., Killie, 750 M., Körnich, H., Le Moigne, P., Lindskog, M., Manninen, T., Nielsen, E. P., Nielsen, K., Olsson, E., Palmason, B., Peralta, A. C., Randriamampianina, R., Samuelsson, P., Stappers, R., Støylen, E., Thorsteinsson, S., Valkonen, T., and Wang, Z.: Arctic regional reanalysis on single levels from 1991 to present. Copernicus Climate Change Service (C3S) Climate Data Store (CDS), https://doi.org/10.24381/cds.713858f6, accessed on 15-12-2022, 2020.

Skakun, S., Justice, C. O., Vermote, E., and Roger, J.-C.: Transitioning from MODIS to VIIRS: an analysis of inter-consistency of NDVI data sets for agricultural monitoring, International Journal of Remote Sensing, 39, 971–992, https://doi.org/10.1080/01431161.2017.1395970, 2018.

Vermote, E., Justice, C., Csiszar, I., Eidenshink, J., Myneni, R., Baret, F., Masuoka, E., Wolfe, R., Claverie, M., and Program, N. C.: NOAA Climate Data Record (CDR) of Normalized Difference Vegetation Index (NDVI), Version 5, https://doi.org/10.7289/V5ZG6QH9, access date: 2022-05-06, 2018.

Vermote, E., Franch, B., Roger, J.-C., Murphy, E., Becker-Reshef, I., Justice, C., Claverie, M., Nagol, J., Csiszar, I., Meyer, D., Baret, F., Masuoka, E., Wolfe, R., Devadiga, S., Villaescusa, J., and Program, N. C.: NOAA Climate Data Record (CDR) of Surface Reflectance, Version 1, https://doi.org/10.25921/gakh-st76, access date: 2023-07-06, 2022.

Wilks, D. S. (2011). Statistical methods in the atmospheric sciences. Academic press.

---

## Author Response (AR1)

In the following, we will address the referees' comments point by point. We mark the comments given by the referee in red, provide our answers and comments in black, and indicate how we addressed the amendments in the manuscript in green.

Tiago Silva, on behalf of all co-authors.

**RC1**: 'Comment on egusphere-2024-2571', Rúna Magnússon, 20 Oct 2024

Dear authors, dear editor
Thank you for inviting me to review "Bio-climatic factors drive spectral vegetation changes in Greenland", by T. Silva et al. for Biogeosciences. The manuscript explores the role of a wide range of bio-climatic factors in explaining satellite-derived vegetation dynamics in Greenland. The authors aimed to identify which sub-surface and above-surface climate factors were associated with greening in Greenland, and how such associations differed among ecoregions and latitudinal and altitudinal gradients. They report that increases in the duration of the thermal growing season show the strongest association with greening, with additional influences of snowpack dynamics, and differential strength of association across regions and altitudinal and latitudinal gradients.

This study is relevant in the context of rapid ongoing climate change in the Arctic, observed dynamics of "Arctic greening" and their implications for the future functioning of tundra ecosystems. The authors analyze a substantial amount of data for a large region, using various sources and environmental disciplines in a holistic and, generally, appropriate way. The manuscript is well within the scope of Biogeosciences and presents a relevant and timely case. I do, however, have several major concerns about some of the methodological choices and the structuring and argumentation of the work. I advise a round of thorough revision and rewriting of substantial parts of the manuscript before it can be considered for publication. This has resulted in a rather lengthy review report, but I would also like to stress that many of the points I raise are interrelated or specific examples of the major points, so I hope it is not discouraging. I am sure the ms will find a good home in a respected journal.
I have performed this review together with 7 MSc students for an open review course assignment at Wageningen University. Their help has been valuable, and they appreciated the opportunity to learn from this ambitious and relevant paper, and to contribute to the scientific publishing process. We have all enjoyed this activity and we wish you all the best as the manuscript comes to full maturity!

Rúna Magnússon,
with input from Annika Robben, Djordy Potappel, Aron den Exter, Muriël de Vries, Rikuto Shinagawa, Yente Reniers and Yorick Kwakkel.

Thank you very much for your comments. We address each of the points raised in more detail below.

**Major comments**

1. I hope the authors can make clarify how the potential mismatches between AVHRR and VIIRS NDVI products (e.g. masking differences) have been accounted for during statistical analysis and trend detection. Explanations on how this was done are sparse and not sufficiently clear to understand the implications.
Beside adding the shaded min-max range in Fig. 2 (that I also don't fully understand the procedure behind, can this be clarified?),
how did you prevent the use of two different records and sensors from affecting your temporal trends? And especially, how do you prevent this from unduly influencing the comparison between 2008-2023 and 1991-2007, that you describe in L. 380-392? This appears to be based on counts of NDVI > 0.15, where differences in bandwidth and snow/water/cloud detection easily become problematic. Miura et al. (2012) may be an appropriate source to evaluate the validity of trend detection across two satellite platforms, and you may want to statistically test for absence of trend breaks coinciding with the switch from one platform to another.

Thank you for raising these concerns.

This is a very important point, namely the homogeneity of the two NDVI time series to each other and what this means for a calculated trend.

The NOAA Climate Data Record (CDR) of AVHRR NDVI - Version 5 and the NOAA CDR of VIIRS NDVI - Version 1 are developed by Eric Vermote and colleagues (Vermote et al. 2018 and 2022) for NOAA's CDR Program. Both records have been processed considering the same atmospheric characteristics as in Miura et al. (2012) and both processed records are posterior to Miura et al. (2012) proposed correction. However, the correction proposed by Miura et al. (2012) is not assessed in polar regions, which may contribute to additional uncertainties in our study.

Unfortunately, no overlap periods are available for the parallel measurements of the two satellite sensors. Therefore, no systematic differences can be determined. However, as we state in *NOAA Climate Data Record for Normalized Difference Vegetation Index* there is work that has calibrated the utilized AVHRR product with MODIS (e.g. Franch et al., 2017) and thus improved the internal homogeneity of AVHRR, as well as work that has established the homogeneity of VIIRS with MODIS (Skakun et al., 2018) to improve the consistency of the NDVI datasets. This does not yet achieve perfect homogeneity, which we explain in the description of both products in *NOAA Climate Data Record for Normalized Difference Vegetation Index as follows*:

According to AVHRR and VIIRS technical reports, the NIR channel is centred at different wavelengths (830 nm vs. 865 nm). As there is no overlapping period available in the NOAA CDR, potential mismatches between AVHRR and VIIRS NDVI cannot be discarded.

We have also revised the same subsection in general to better understand the processing of two NDVI data sets and the problems of data homogeneity. We also reformulate in *Spectral greenness* how spectral greenness is derived from the AVHRR and VIIRS NDVI to better explain how the shaded area later shown in Figure 2 is calculated:

As estimates integrated through time are less likely to be influenced by temporal sampling artefacts at high latitudes than metrics based on maximum NDVI (e.g., Myers-Smith et al. 2020), we started by calculating monthly integrated NDVI. Also, since our focus is on green vegetation, only daily NDVI pixel values with higher or

equal to 0.15 are considered. Then, we divide the monthly integrated NDVI by the total number of monthly observations (n, see Figure S1 for the interannual variability of n) to obtain the monthly NDVI. However, before 2014 and as described in Subsection 2.2, the AVHRR algorithm was less strict in its data quality control compared to VIIRS from 2014 onward, resulting in higher n before 2014 that lowers monthly NDVI. To address temporal heterogeneities, we adjusted n from the AVHRR period with the number of monthly observations acquired during the VIIRS period. From 2014 to 2023, we identified the minimum, maximum and average number of observations for each month. Hence, using these three quantities, we generated a consistent variability range from 1991 to 2013 to recalculate monthly NDVI, considering a similar number of observations as from 2014 to 2023. This procedure assumes that the environmental conditions (i.e. snow-cover, clouds and shadow) between 1991 to 2013 are similar to those between 2014 and 2023. The maps for the average number of monthly observations and the associated standard deviation for AVHRR and VIIRS period before and after the adjustment regarding n are shown in Figures S2-S5, respectively.

We investigate at the start of *Results* how summer spectral greenness statistically relates with climate oscillations (e.g., Greenland Blocking Index) for AVHRR, VIIRS and the entire study period. We use these climate oscillation time-series, that are homogenous and independent of spectral greenness, as a reference to evaluate systematic inconsistencies that may arise due to sensor change.

It should be noted that prevailing weather patterns during summer months, like the North Atlantic Oscillation (NAO) and the Greenland Blocking Index (GBI), are highly correlated with spectral vegetation (Fig. S7). Therefore, summer weather patterns can accelerate or delay the maximum green vegetation extent given their link with temperature and precipitation. Correlations between green vegetation extent and summer GBI are investigated for three periods: AVHRR (1991-2013), VIIRS (2014-2023) and the full period (1991-2023), and are shown in Table S1. Positive and significant correlation coefficients ranging between 0.5 and 0.8 are found between ecoregion 1 and 4, generally with higher correlations for VIIRS than for AVHRR period. Green vegetation extent in ecoregion 5 is poorly correlated with the prevailing weather patterns during summer.

While the AVHRR 22-year trend evidence general expansion of green vegetation, the VIIRS 9-year trend evidence decreases, particularly in West Greenland (Table S2). However, due to high variability and small sample size, most trends in both periods are not significant.

We address in *Study limitation and future research*, our concerns about the reliability of long-term time integrated NDVI analysis

The NDVI datasets employed in this study are sourced from two satellite products processed by NOAA, each utilizing a different type of sensor. Due to the absence of a temporal dataset overlap, the assessment of uncertainties was limited and potential for mismatches between the datasets cannot be discarded. This lack of a common calibration period raises concerns about the reliability of long-term time integrated NDVI analysis.

In the end, we follow similar approaches of recent literature (e.g., Madson et al. 2023, Pourmohamad et al. 2024) that make use of the full AVHRR NDVI and VIIRS NDVI without additional corrections.

2. Your methodology is ambitious and extensive, which is laudable. It does however lead to many choices during the processing of the data, and not all of these have been properly backed or described yet. Examples include (1) the use of 0.15 as an NDVI threshold without a reference, (2) the described use of CryoClim data that only go to 2015 without any visible inclusion of these data throughout later analyses, (3) why has only altitude, and not for example slope aspect, been included into the study? These, and further examples, are given in the minor comments. I suggest that the authors critically go through every step in the methodology and check whether all choices are described in sufficient detail for an independent reader to reproduce the study, and that choices are back-up either by literature, data or statistics. If needed, details on processing can be described in a supplementary methods section to prevent disruption of the flow of the main text.

Thanks for your kind words endorsing our ambitions! We are sorry that relevant information is missing here. This is essential to guarantee reproducibility in future interdisciplinary studies.

The citation for the choice of the NDVI threshold of 0.15 to derived spectral greenness is now expanded in *Spectral Greenness*: Arctic regions are characterized by sparse vegetation, that typically exhibit markedly low NDVI values, often as low as 0.15 (e.g., Gandhi et al. 2015; Liu et al. 2024), with dense shrubs above 0.5 (e.g., Walker et al. 2005), and signal saturation at around 0.7 (e.g., Myers-Smith et al. 2020).

Regarding the CryoClim, we expanded *Copernicus Arctic regional reanalysis* noting that: The data providers assure that the data for the period post-2015 have been produced and arranged in collaboration with the CryoClim developers at the Norwegian Meteorological Institute.

We have taken up the idea of including slope and aspect in the analysis. The results can be found in *Coastal, latitudinal and altitudinal dependence on trends* along with supplementary figures.

The relationship between GrowDays and topographical features such as slope and aspect were further explored. As the surface slope is highly correlated with surface elevation, trends in GrowDays tend to significantly decrease with steepness. The dependence between GrowDays and surface aspect is rather complex, without a predominant slope orientation promoting GrowDays, in general. However, latitudes immediately south of Maniitsoq Ice Cap show increases of GrowDays in slopes with southwest orientation. On the East coast, a western slope orientation is particularly pronounced along Jameson Land, whereas northeast exposure appears favourable north of ecoregion 5.

The dependence of the slope orientation for greenness changes is partly in alignment with the dependence of the slope orientation for GrowDays. Greenness trends increased in two latitudinal bands facing southeast in ecoregion 1 and 2. In Jameson Land a similar tendency for more greening is found towards southwest, while east facing slopes are preferred towards the northern part of ecoregion 5.

3. From L. 227 onwards, it reads as if the distinction between methods, results and interpretation of results (discussion) is lost. For example: results and maps are presented in the methods in L. 227-247. New information on choices of processing, variable selection and statistical tests (Pearson correlations) are introduced in the results in L. 275-305, L. 311-314 L. 380-384 and many other places. Throughout the entire (lengthy) results section, interpretations are added that go beyond the statistical results of your own methods. Lines 320 -350 for instance are very speculative for a results section, and other paragraphs and show similar interpretation or speculation. These would be better suited for the discussion and require backing by references. Please rewrite the methods-results-discussion in such a way that: (1) all methodological choices and tests are explained in the methods (2) only numerical and statistical outcomes are presented in results (with a minimum interpretation to make the results understandable, e.g. writing out abbreviations and description of patterns) and (3) interpretation and relation to unmeasured mechanisms such as permafrost, latent heat processes or photosynthesis are only kept for the discussion.

Thank you for pointing this out. We rewrote the Methods, Results and Discussion as suggested by the referees. To make the differentiation between these sections more evident, we:

1. Rewrote several parts in the subsection *Statistical Methods* and removed the above-mentioned methodological choices from the *Results*.
2. Rewrote several parts in *Key findings and interpretation in the context of the current* literature, removing the above-mentioned interpretations from the *Results* and back up our statistical outcomes with experimental studies.
3. Rewrote the *Results*, properly backing our statistical outcomes with supplementary figures. We apologize if this last point may be seen as too generic, but we show below along the Minor points the amendments made regarding rewriting of the methods-results-discussion, as suggested by the referees.

Several of these revisions are mentioned and expanded in the minor points.

4. In the results section and abstract, observed greenness dynamics are attributed to processes such as nutrient dynamics and permafrost. This gives the reader the impression that such variables were included or that you can at least confidently attribute greening dynamics to such processes. Given the set of bioclimatic factors that were included, however, I doubt whether you can make such claims. These processes can be touched upon in the discussion, with support from literature, but should not be presented in a way that readers might think that these are actual conclusions from this study. I also think that to properly discuss their role in the discussion, you will need to evaluate several lines of reasoning more critically: are the subsurface products (soil water and soil ice) that you include, given the limited representation of subsurface dynamics in the used reanalysis products, actually representative of permafrost conditions or hydrology? How can you better argue the role of snowmelt rates in relation to microbial activity and nutrient dynamics, especially to an audience that may not be familiar with works such as musselman et al.? Because at first it is very counterintuitive that shallower snowpacks melt more

slowly and with the current explanation provided, this line of argumentation is very hard to follow. I suggest you evaluate to what extent your bioclimatic variables are representative of processes such as permafrost dynamics and melt rates and nutrient dynamics, discuss their potential roles in the discussion section, and refrain from making any hard statements about their role in the abstract/results/conclusion sections.

Thank you for valuable comment! To better distinguish our conclusions from interpretations and to follow the advice of the referees:

1. We rewrote the *Abstract*, given more emphasis to our results than certain interpretations.

2. We rewrote parts of *Introduction* to better explain how soil water physically relates to the snowpack characteristics and, in particular, to show the sequence of processes by which a shallow snow cover can influence microbial activity:

   A relevant characteristic of the snowpack is that deep snow requires more energy to equalise the cold content and the liquid water holding capacity to subsequently initiate and sustain melt than shallow snowpacks (Colbeck 1976; Musselman et al. 2017). (...) Concurrently, meltwater from relatively shallow snow percolates the soil more efficiently during the ablation period, in contrast with fast snowmelt that quickly saturates the soil surface and runs off (Stephenson and Freeze, 1974). See also our detailed answer under minor comment.

3. And also, the importance of climate oscillations: Grimes et al. (2024) has recently shown that the doubling of vegetation across ice-free Greenland is linked with warming. The warming observed in Greenland over recent decades has been associated with more frequent and intense weather patterns that promote widespread clear-sky conditions and the advection of relatively warm air masses from southern latitudes along Western Greenland (Barrett et al., 2020). Weather patterns can be related to indices by analysing specific atmospheric variables over time and space. For instance, the North Atlantic Oscillation is driven by surface pressure configurations in the North Atlantic (Hurrell et al., 2003), and the Greenland Blocking Index by the geopotential height in the mid-troposphere over Greenland (Hanna et al., 2016). Both indices are commonly utilized in climate studies to deduce influences on various components of the climate system in Greenland and vicinity (e.g., Bjørk et al. 2018; Olafsson and Rousta 2021).We added more details on CARRA (*Copernicus Arctic regional reanalysis)* regarding the multi-layer surface model (SURFEX) and its schemes as dependent on the surface type: SURFEX is a multi-layer surface model that computes specific schemes dependent on the surface type (e.g., vegetation, soil, snow), allowing soil water phase changes and enabling runoff over frozen and unfrozen soil. This helps to better represent areas with permafrost and ice surfaces in Greenland as they are not well described in the present version of HARMONIE-AROME.

4. We added to *Study limitations and future research directions*, the limitations of CARRA on the representation of permafrost and nutrient dynamics, thus better contextualizing the explanatory value of these variables for the vegetation

changes in Greenland: (...) better representation of the permafrost extent and active layer thickness along with the inclusion of dynamic tundra vegetation models within CARRA could be beneficial in improving our knowledge on interactions among atmosphere, vegetation, carbon and nitrogen cycling, water and permafrost dynamics.

**Minor comments**

1. The writing could be improved by splitting up some very long compound sentences into shorter ones. I provide some examples in the "technicalities", but I recommend a thorough re-reading for writing style and grammar.

Thank you for this very relevant point. In addition to the examples mentioned in "technicalities", we revised our sentence construction to improve readability throughout the manuscript.

2. The abstract ends with the conclusion that you "identify a set of bioclimatic variables" and that you provide a "basis to validate bioclimatic indicators from climate models". Your conclusions section states more or less the same. I suggest that you reflect more specifically on how exactly your findings help to achieve this (more to the point). This will hopefully also better explain how you advance the field, since the role of growing season onset and snowmelt timing are already well established in Arctic ecological studies.

Thank you for pointing out that we had to further develop our abstract and conclusions to show how our work advances the field. We hope that the revised *Abstract:*

[revised manuscript text omitted]

3. 35 – 43. Several references seem out of place in this paragraph. I suspect you mean Bjorkman et al. (2018) instead of Metcalfe et al. (2018), since Metcalfe et al. (2018) does not deal with the type of findings you describe at all, and Anne Bjorkman's paper does. Sturm et al. (2001) is a rather old and case-specific (albeit popular) reference for shrubification of the Arctic. ITEX papers (e.g. Elmendorf et al., 2012) or syntheses (e.g. Mekonnen et al., 2021; Martin et al., 2017; Myers-Smith et al., 2011) would be more appropriate.

Thank you for identifying misplacement of references. We followed your advice and adapted this accordingly.

4. 66 – 68, this proposed increase in nutrient availability under deeper snow is at odds with your statements in the abstract, results and discussion that shallower snowpacks should melt more slowly. It should be clear from the introduction onwards which snowpack properties can be expected to facilitate faster or slower melt, and how would relate to nutrient cycling. If the literature on the influence of snow dynamics on microbial turnover and nutrient availability is ambiguous in itself, then I would refrain from making any statements about nutrients as a mediating effect between snow dynamics and greenness.

Thank you for raising this point. We improved a few paragraphs in the *Introduction* better explaining the relationship between snowpack characteristics and snowmelt, rate and how they relate to nutrient cycling. The most relevant revised paragraphs in this regard are:

*Introduction*

Increased snow depth during the cold season usually causes increased plant growth in the following summer, as more snow provides insulation, less frost damage and, depending on the snowpack characteristics, increase in water availability (e.g., Lamichhane 2021; Migała et al. 2014; Wang et al. 2024).

A relevant characteristic of the snowpack is that deep snow requires more energy to equalise the cold content and the liquid water holding capacity to subsequently initiate and sustain melt than shallow snowpacks (Colbeck 1976; Musselman et al. 2017). As a result, deep snow often subsists for longer periods, potentially delaying the start of the growing season, which can hinder plant growth (Schmidt et al. 2019). On the other hand, the insulation provided by deep snow has also been demonstrated to promote increased microbial decomposition, enhancing the nutrient supply for the following growing season (e.g., Cooper 2014; Pedron et al. 2023; Xu et al. 2021). The higher amount of energy input needed to melt deep snow means that it melts later but also faster, which can cause nutrient loss through increased runoff.

Concurrently, meltwater from relatively shallow snow percolates the soil more efficiently during the ablation period, in contrast with fast snowmelt that quickly saturates the soil surface and runs off (Stephenson and Freeze, 1974). These slow snowmelt rates allow water to remain in the soil for extended periods, which is critical

for activating soil microbe communities. These microbes then produce nutrients that are vital for vegetation growth (Glanville et al. 2012).

*Results*:

These shallow snowpacks seem to be linked to more water content in the soil in spring (SoilWaterMAM loading vector opposite to SWEMAX loading vector). Additionally, the earlier snow depletion and thus earlier onset of the thermal growing season relates to enhanced spectral greenness (Onset loading vector opposite to Greenness loading vector).

In the *Discussion*:

For most ecoregions in ice-free Greenland, we find that snowpacks are becoming shallower, and consequently melt slowly, but earlier in the season. This feature was mentioned by Musselman et al. (2017) and is attributed to global warming. Musselman et al. (2017) explains that in Western North America regions with shallower snow are experiencing snow season contractions. Shallower snow is susceptible to snow season contraction because shallow snow requires less energy to initiate melt than deeper snow. This earlier start of the ablation period occurs at a slower rate due to a combination of near-surface warming with relatively low solar altitude angles. In contrast, for deep snowpacks that require more energy to initiate runoff, it is also more likely for the snowmelt water to refreeze within the snowpack (Dingman, 2015). Therefore, early season slow snowmelt rates in shallow snowpacks allow for efficient soil water percolation and subsequent water storage (Stephenson and Freeze, 1974). The successful percolation of liquid water into soil plays a key role in tundra regions during the snow ablation period and start of the growing season, as during this time soils are generally dry due to high drainage (Migała et al., 2014).

Increased water availability in the soil could stimulate dormant microbial communities and thus increase the decomposition of soil organic matter, releasing soil nutrients (e.g., Glanville et al. 2012; Salmon et al. 2016; Xu et al. 2021). This in turn could prime the soil for earlier and more efficient vegetation growth and colonization. The increased spring soil water content (SoilWaterMAM), spring near-surface air temperature (T2mMAM), and lengthening of the thermal growing season (GrowDays) indicated in our results could therefore improve conditions for plant growth and colonization, especially in the southern ecoregions.

5. 111, here you mention the use of CryoClim data, that was chose to represent daily snow cover rather than the CARRA dataset. I do not see how this could be done since the data only goes to 2015, and this data product is not mentioned anywhere anymore in the remainder of the ms. Did you actually use it and if so, how? Perhaps it is a nice addition to incorporate data sources directly into Table 1 to resolve unclarities like this.

Thank you for the request to clarify this point. CryoClim is one of the data sources/products assimilated by CARRA that does not extend along the entire reanalysis period. Nevertheless, CARRA data providers assure that the data for the period post-2015 have been produced and arranged in collaboration with the CryoClim developers at the Norwegian Meteorological Institute.

We decided to highlight CryoClim in the data description to indicate that a remotely sensed product is used to a certain extent to represent snow-covered regions.

Although CryoClim has not been available since 2015 and thus inhomogeneity in CARRA could be assumed, a similar product has been provided by the Norwegian Meteorological Institute for the rest of the period. Additionally, van der Schot et al. (2024) reports how CARRA performs against in situ measurements until 2023 across Greenland with no obvious change after 2015. Accuracy metrics were provided as suggested in point 10.

6. 128-153: Can you give an indication of the match between AVHRRR and VIIRS? Calibration against MODIS does not seem to be the most relevant thing to mention here, since you do not use MODIS. See Miura et al. (2012), there seem to be some structural NIR differences and non-linear NDVI relationships between VIIRS and AVHRR?

Major comment 1 also highlighted revisions required in this area, please also refer to our response there. The following text is now included in the paper:

1. *NOAA Climate Data Record for Normalized Difference Vegetation Index*: According to AVHRR and VIIRS technical reports, the NIR channel is centred at different wavelengths (830 nm vs. 865 nm). As there is no overlapping period available in the NOAA CDR, potential mismatches between AVHRR and VIIRS NDVI cannot be discarded. However, AVHRR NDVI uses the MODIS Land-Sea mask and its cloud mask is spectrally adjusted using 10 years of MODIS data, with 90% match accuracy over land (Franch et al. 2017). As VIIRS will eventually replace MODIS for land science, MODIS is also used to calibrate VIIRS NDVI estimates (Skakun et al. 2018).

2. *And furthermore, in the Results*: It should be noted that prevailing weather patterns during summer months, like the North Atlantic Oscillation (NAO) and the Greenland Blocking Index (GBI), are highly correlated with spectral vegetation (Fig. S7). Therefore, summer weather patterns can accelerate or delay the maximum green vegetation extent given their link with temperature and precipitation. Correlations between green vegetation extent and summer GBI are investigated for three periods: AVHRR (1991-2013), VIIRS (2014-2023) and the full period (1991-2023), and are shown in Table S1. Positive and significant correlation coefficients ranging between 0.5 and 0.8 are found between ecoregion 1 and 4, generally with higher correlations for VIIRS than for AVHRR period. Green vegetation extent in ecoregion 5 is poorly correlated with the prevailing weather patterns during summer.

   While the AVHRR 22-year trend evidence general expansion of green vegetation, the VIIRS 9-year trend evidence decreases, particularly in West Greenland (Table S2). However, due to high variability and small sample size, most trends in both periods are not significant.

3. *Study limitations and future research directions* that The NDVI datasets employed in this study are sourced from two satellite products processed by NOAA, each utilizing a different type of sensor. Due to the absence of a temporal overlap from the data providers, the assessment of uncertainties was limited and potential for mismatches between the datasets cannot be discarded. This lack of a common calibration period raises concerns about the reliability of long-term time integrated NDVI analysis.

**7. 143, why did you use an NDVI threshold of specifically 0.15?**

In the revised manuscript we clarify this point in *Spectral greenness* and add the following: Arctic regions are characterized by sparse vegetation, that typically exhibit markedly low NDVI values, often as low as 0.15 (e.g., Gandhi et al. 2015; Liu et al. 2024), with dense shrubs above 0.5 (e.g., Walker et al. 2005), and signal saturation at around 0.7 (e.g., Myers-Smith et al. 2020).

**8. 146, can you provide a sharper definition of "interannual extent of vegetation"? To ecologists, this may be confusing since extent almost always refers to spatial extent.**

Thanks, we agree, this was formulated in a confusing manner. In the revised manuscript, we sharpen in *Spectral greenness* that Pixels with monthly NDVI equal to or greater than 0.15, representative of the area covered by green vegetation, are used to estimate the green vegetation extent.

**9. 147-155. It is very difficult for readers who are not intimately familiar with the AVHRR and VIIRS datasets to follow this paragraph, even though it is quite important for the quality of the results. Terms like "flag" and "n" may be unclear. Please provide more explicit description of exactly how the monthly max/mean/min nr. of valid pixels was used and how this translates to the CI's in Fig. 2. From reading this several times I still did not understand if any correction was applied before further analysis (and looking at Fig. S1 I would expect for that to be necessary).**

In the revised manuscript, we clarify in *Spectral greenness*: As estimates integrated through time are less likely to be influenced by temporal sampling artefacts at high latitudes than metrics based on maximum NDVI (e.g., Myers-Smith et al. 2020), we started by calculating monthly integrated NDVI. Also, since our focus is on green vegetation, only daily NDVI pixel values with higher or equal to 0.15 are considered. Then, we divide the monthly integrated NDVI by the total number of monthly observations (n, see Figure S1 for the interannual variability of n) to obtain the monthly NDVI. However, before 2014, the AVHRR algorithm was less strict in its data quality control compared to VIIRS from 2014 onward, resulting in higher n before 2014 that lowers monthly NDVI. To address temporal heterogeneities, we adjusted n from the AVHRR period with the number of monthly points acquired during the VIIRS period. From 2014 to 2023, we identified the minimum, maximum and average number of valid points for each month. Hence, using these three quantities, we generated a consistent variability range from 1991 to 2013 to recalculate monthly NDVI, considering a similar reduction of points as from 2014 to 2023. This procedure assumes that the environmental conditions (i.e. snow-cover, clouds and shadow) influencing the number of data points between 1991 to 2013 are similar to those between 2014 and 2023. The maps for the average number of monthly observations and the associated standard deviation for AVHRR and VIIRS period before and after the adjustment regarding n are shown in Figures S2-S5, respectively.

**10. 163, it would be useful to report an accuracy metric here.**

Thank you for pointing this out. In in the revised version *Bio-climatic factors*, we write that van der Schot et al. (2024) demonstrate in a recent study that the agreement is

strong between the snow water equivalent modelled by CARRA and a snow model utilizing in situ observations in both the West and East coastal regions of Greenland. They report that CARRA is capable of successfully representing snow-related indicators, with correlation coefficients exceed 0.8 and mean absolute percentage errors less than 30%.

**11. 169, why only from January onwards and not in autumn-winter previous year?**
Thank you for reflecting on the definition of rain-on-snow days. In the revised version of *Bio-climatic factors*, we state that SnowDays, in combination with RainRatio higher than 50%, are used to derive days with rain-on-snow (RainOnSnow) between January and July to investigate potential snowpack warming before the thermal growing season onset.

**12. 169-171, I have a slight doubt about the way that the melt rate is calculated here. If this basically represents the time that passes between the peak SWE and moment of complete snowmelt, and peak SWE occurs early in the winter-spring season, how representative is this timeframe really for the spring melt season and water release? Especially if heavy snowfall occurs later in spring and is followed by warming, this automatically leads to a situation where deep snow appear to melt more rapidly. As a reader, it is hard to fully grasp how such nuances in the choice of processing influence the results.**
Thank you for reflecting on the definition of snowmelt rate, which made us adapt our methodology in order to improve. We changed Table 1 in *Bio-climatic factors* and expand for clarity: mean melt rate for ablation days between SWE_MAX DOY and Onset of GrowDays and in the text below During the snow melt period, we calculated daily changes of SWE from which we derived days with negative SWE changes (SWEmeltDays) and the mean of the negative SWE changes (MeltRate).
This improved approach significantly increased the correlation of MeltRate with SWE_MAX to 0.7, reinforcing the already physically discussed relationship between snow depth and snowmelt rates. Heavy snowfall occurring later in spring does not seem to impact the co-variability between SWE_MAX and MeltRate across the entire ecoregion and the 32 years in study.
To avoid redundancy in the PCA, MeltRate is no longer a feature.

**13. 176, you mention rain, but rainfall does not seem to be included as a bioclimatic variable as far as I can see (Table 1, Fig. 3), while snowfall was, and rain fraction too. You refer to Fig S10 for statements on the role of rain, but this figure refers to "solid precipitation" which suggests that this is about snow. Since you discuss the role of rain regularly, why not include rain (total summer season liquid precipitation) as a bio-climatic variable explicitly? This would make your conclusions and discussion points on the role of rain more explicit and justifiable.**
Our apologies for the incomplete point in the previous version. Thank you, again, to reflect on our approaches. Rain is added to Table 1 and Figure 3. Also, RainJJA, along with RainRatioJJA, is added as a feature in the PCA, as we acknowledge that changes in RainRatio and Rain amount are different quantities with different impacts on the surface.

**14. You could statistically back up your choice for PCA and its assumption of linear relations. You could do this by reporting axis lengths, for instance.**

Thanks for the remark! As we standardized all variables prior to analyses, we opted for unimodal and linear species response, as PCA is better suited for low variance, small gradients and more intuitive for the interpretation of the biplots.

**15. Fig. 1, here results are presented, and completely new information comes in (NAO / GBI), so perhaps the figure should be presented later, in the results. I also miss a scale bar for greenness and it is unclear what "greenness" represents here (is this one the extent variables you calculated, or a mean, and are pixels < 0.15 included or not?).**

We apologize for the misunderstanding and for the inconsistencies of the earlier submitted figure. We now:

1. Added to the Introduction that Grimes et al. (2024) has recently shown that the doubling of vegetation across ice-free Greenland is linked with warming. The warming observed in Greenland over recent decades has been associated with more frequent and intense weather patterns that promote widespread clear-sky conditions and the advection of relatively warm air masses from southern latitudes along Western Greenland (Barrett et al., 2020). Weather patterns can be related to indices by analysing specific atmospheric variables over time and space. For instance, the North Atlantic Oscillation is driven by surface pressure configurations in the North Atlantic (Hurrell et al., 2003), and the Greenland Blocking Index by the geopotential height in the mid-troposphere over Greenland (Hanna et al., 2016). Both indices are commonly utilized in climate studies to deduce influences on various components of the climate system in Greenland and vicinity (e.g., Bjørk et al. 2018; Olafsson and Rousta 2021).

2. Opted to keep the absolute values away from the greenness scale in Figure 1 as they correspond to a 32-year averaged greenness, and it would not be useful for any further interpretation.

3. Simplified Figure 1 in the main manuscript, with the correlation maps of summer NAO and GBI moved to supplementary material.

[Figure]

*Figure 1  Ecoregions in ice-free Greenland, June, July and August averaged spectral greenness for the period 1991-2023. No scale shown in the colour bar because the aim is to illustrate spectral greenness patterns, not absolute values. Place names referenced in the study are indicated.*

4. Kept the delineation of the ecoregions and the greenness evolution during summer in the Methods, as they will support the readers to understand the geography of the ecoregions and to recognise the greenness dynamics across Greenland from June to August as well as what entails the summer averaged greenness.

16. 227-247 seem to be combined methods and results. The source for the climate oscillation data, and the rationale for including them, have not been properly covered earlier in the methods. It is also unclear how the use of oscillations relates to your study aim and research questions.

Thanks for the remark. We added to the *Introduction* how climate oscillations play a role on vegetation as indicated in minor point 15. We added to Data a new subsection *Climatic oscillation index*, where we describe the data used.

A variety of analytic approaches, such as principal component analysis (PCA) or k-means clustering, are often utilized to characterize the North Atlantic Oscillation (NAO), with input data sourced either from reanalysis or station records. Here, the NAO derived from sea-level pressure applying PCA is used. In this study, the NAO index calculated applying the leading principal component derived from sea-level pressure anomalies within the Atlantic domain (20∘N–80∘N, 90∘W–40∘E) is provided by NCAR/UCAR (Hurrell et al., 2003). This product is posited to yield a more comprehensive representation of NAO spatial patterns compared to indices based on specific terrestrial stations. Notwithstanding, it is noteworthy to acknowledge the dynamic nature of PCA-based NAO indices, being subject to ongoing refinement with the integration of new data.

The Greenland Blocking Index (GBI) is derived from 500 hPa geopotential height over the region (60∘N–80∘N, 80∘W–20∘W), retrieved from PSL/ESRL (Hanna et al., 2016). Both the NAO and GBI indices originate from the NCEP/NCAR reanalysis dataset (Kalnay et al. 1996). Consequently, these climatic oscillation indices have undergone seasonal standardization against the baseline period of 1950–2000.

In the *Results* we also show how these climate oscillations statistically related with green vegetation extent and help to explain the opposite greenness trend signal for the AVHRR and VIIRS periods as shown in minor point 6.

17. 250, Pedregosa et al does not seem like the most appropriate reference for the use of PCA. I advise to find papers that specifically deal with the considerations and strengths of using PCA in a pixel-based remote sensing context.

Thank you for the comment. Now in the *Statistical Methods*, we expand to: Principal Component Analysis (PCA, Pearson 1901; Lorenz 1956), often used on remotely sensed and environmental data (e.g., Mills et al. 2013; Yan and Tinker 2006), was employed to investigate the combined influence among bio-climatic indicators with summer greenness.

18. 259, the use of Mann-Kendall tests is state of the art, but it appears that later on you only show results for growdays and greenness, not all bioclimatic factors as suggested here? Perhaps mention only growdays and greenness then?

We performed a regression and the Mann-Kendall trend test to all bioclimatic indicators in our study. However, attempting conciseness, we only displayed GrowDays and Greenness in the main manuscript. Although other bioclimatic trends are not shown explicitly, their results are referred to throughout the manuscript (e.g., Fig S2 and S10). Therefore, we added more supplementary figures to back our results as also requested in point 23.

19. 262, please explain the use of a 90% confidence interval rather than 95%. With the vast amount of pixels at your disposal, and the relatively long timespan of the study, I would expect that the generally accepted 95% CI would be fine and I would be curious to know why you deviated from this standard.

Thanks, we followed your advice and decreased the test level to 5%, as commonly used in ecology.

20. 271-273, the statements made here need backing; how did you test whether significant long-term trends in vegetation extent were evident? Mann-Kendall test? Could sensor discrepancies play a role here?

Thank you for your comment. We expand on the sensor heterogeneity in our reply to major comment 1. We add in the *Results*: While the AVHRR 22-year trend evidence general expansion of green vegetation, the VIIRS 9-year trend evidence decreases, particularly evident in West Greenland (Table S2). However, due to high variability and small sample size, most trends in both periods are not significant. The significant long-term trends range from 2 % per decade in ecoregion 1 to approximately 6 % per decade in ecoregion 4.

21. 275-279, reads like methods and introduces a whole new aspect of the methodology. I would also provide some more explanation of why the use of detrended Pearson correlations is an appropriate method to evaluate linearity assumptions for a PCA.

Thank you for your comment. We removed the information mentioned in the comment from the *Interconnectedness among bio-climatic indicators* and added to *Statistical Methods* that: As the classic PCA requires the variables to be linearly related, we calculated Pearson correlation coefficients to investigate bio-climatic indicators by ecoregion. However, Pearson correlation assumes that the data are stationary; that is, their statistical properties do not change over time. In order to avoid serial autocorrelation, we transform the data into non-stationary time series by linearly detrending the data before performing the correlation.

22. 290 & 296, you describe how specific variables were removed from analysis a priori. This is essential information that should go into methods, and it seems at odds with your earlier statement that variables were excluded from PCA based on contribution to cumulative explained variance. I would recommend to present a single, unambiguous criterium for the inclusion of variables into PCA and figures, in the methods. Especially since the identification of useful bio-climatic indicators was an explicit aim of the study.

Thank you for your comment. We moved and rephrased the information mentioned in the comment from the *Results* and added in *Statistical Methods*: The calculated correlations are displayed in a correlation matrix, and bio-climatic indicators with similar correlations are sorted with hierarchical clustering. This helped to visually discern bio-climatic indicators with comparable statistical relationships and supported on the empirical reduction of indicators accounting for the relevant physical and the ecological processes on the tundra ecosystems, later used as part of the PCA. This will diminish "noise", redundancy and ultimately boost the clarity of interactions across atmosphere-biosphere-cryosphere.

23. 291-292 & L. 294-295, examples of interpretation of results, and no backing (figure, reference) provided to support these interpretations.

Thanks for pointing this out. We acknowledge that some of our statements were indeed too bold and misplaced. Certain statements were moved to and rephrased in the *Key findings and interpretation in the context of the current literature*, while others were rephrased, remained in the *Results* and backed up by a series of supplementary figures.

24. 311-314, I had to read this section a few times to understand the rationale and approach. So if I read correctly, you applied the PCA for all years and ecoregions separately, and then tested whether the variances explained by PC1 and PC2 were similar across the two time periods. I am not fully sure how this would demonstrate that the two NDVI records are comparable and valid in this context. The variances may be similar, but the greenness dynamics, and the associations between different variables and PC axes may not be (do I understand this correctly)? Sidenote: a lot of this information again reads like methods and not results.

Thank you for the remark. We provided information in major point 1 and minor point 6 on how we handled the NDVI records. Additionally, it was important to assess whether sensor discrepancies could have severely impacted the interactions among bio-climatic indicators, influencing loading vectors and the explained variance. Therefore, the inter-annual PCA was performed and assessed for statistically significant differences. The information referred in the comment was removed from the *Results* and added in *Statistical Methods*: Due to a change of satellite sensor from 2014 onwards, we also investigated how PCA performs interannually and whether there was a statistically significant change of the explained variance for years before and after 2014. The result is shown in Figure S8 for a set of 16 bioclimatic indicators, displaying that the two independent samples of explained variance have identical averages in all ecoregions, with a 95 % confidence level, as determined by a two-sample t-test.

25. This is a nice figure! Also here, a scale bar for greenness would help the reader understand what kind of magnitudes we are talking about, across regions.

Thank you for appreciating our charts! We added a scale bar in each subplot to display the range of greenness for the years between 1991 and 2023.

[Figure]

*Figure 2 Biplot for scores between 1991 and 2023 for each ecoregion. The loading vectors are labelled and scaled by the maximum of each principal component. The scores are colour-coded based on the summer spectral greenness, with different scales to enhance greenness. The explained variance of the first (PC1) and second (PC2) component is labelled in the corresponding axis of the subplot. The 16 bio-climatic indicators are 1: maximum snow water equivalent (SWEMAX); 2: total number of thermal growing days (GrowDays); 3 and 4: start (Onset) and termination (End) of GrowDays; 5: summer spectral greenness (Greenness); 6: rain in summer (RainJJA); 7 and 8: averaged rain ratio in summer (RainRatioJJA) and autumn (RainRatioSON); 9, 10, 11, 12: averaged 2-m air-temperature in winter (T2mDJF), spring (T2mMAM), summer (T2mJJA) and autumn (T2mSON) 13 and 14: volumetric soil water in spring and (SoilWaterMAM) autumn (SoilWaterSON); 15 and 16: volumetric soil ice in winter and (SoilIceDJF) summer (SoilIceJJA). The abbreviations of the bio-climatic indicators are described in Section 3.2 and in Table 1. The spatial pattern of the averaged 1991–2023 scores for both components in every ecoregion, including their corresponding loadings, are shown in Fig. S10-S14.*

26. 318-319, "PC2 is heavily shaped by continentality, permafrost extent and precipitation patterns, meaning that snow-related indicators, like SWEMAX and MeltRate have the highest explanatory power". I struggle to see how your variables and methods could allow you to conclude anything about continentality or permafrost. This needs to be either backed up better, or (ideally) kept for the discussion. I also do not see how this means that snow related indicators are most important (snow is something different than permafrost and continentality?).

Thank you for the remark. We indeed did not process enough data to reliably conclude about continentality or permafrost. Therefore, we rephrased our statement to According to the spatial maps of the first (PC1) and second component (PC2, Fig. S10-S14), PC1 is found to be highly controlled by the topography of the ecoregion, and is

consequently related to temperature (and through that on elevation), making GrowDays the bio-climatic indicator with the highest loading in all ecoregions, and therefore, the most significant contributor to the pattern represented by PC1. Through the analysis of the trend map for summer rainfall (Fig. S15) and the spatial maps of PC2, we found that PC2 relates to precipitation and snow patterns, with SWEMAX and RainJJA having the highest explanatory power.

27. L 320 – 350 are altogether quite speculative and many of the claims here need to be supported either by a figure, statistics or literature (and in the latter case, it is better suited for the discussion). I would advise to back up your statements much more. And please carefully evaluate whether reported drivers are really drivers, or just represent the overall role of warming (e.g. increases in rainratio cannot really be teased apart from warming effects so I do not see how you would attribute change to rainfall patterns specifically, especially if total rainfall is not included in the analysis). I think this paragraph needs a thorough rewriting.

Thank you for your comment. We rephrased the entire sub-section *Bio-climatic indicators interlinked with greenness*, moving the interpretations to *Key findings and interpretation in the context of the current literature* with proper citations of experimental studies. Rain is in the revised version included as a bio-climatic indicator as indicated in minor point 13.

Changes in several bio-climatic indicators, such as near-surface air temperature and RainRatio in summer, are indeed related to tropospheric warming. However, warming is not uniform with elevation. While changes in RainRatioJJA are generally related to elevations, changes in T2mJJA are not so clear with elevation in ice-free Greenland. This comment ended up encouraging us on the inclusion of RainJJA as a PCA feature.

28. 349-350, please consider how this relates to the aims of the study (oscillations are not introduced anywhere), report the approach in the methods, and report the test statistics either here or in the appendix.

Thank you for the comment. Climate oscillation indices are now in the *Introduction* and described in minor point 15, with a dedicated section in the *Data* and described in minor point 16.

29. 380, at this point the different terms used (here: spectral vegetation expansion) become a bit confusing. It would be nice to have a single, consistent term for each of the various manifestations of greening that you study in this paper, and present all of these early on.

Thank you for the notice. Only three terms are used along the manuscript: spectral greenness, green vegetation extent and green vegetation distribution. We added these definitions prominently upon first occurrence in *Spectral greenness*.

30. 382, see also major comments, here I was very unsure whether the differences in bandwidths and quality filtering might introduce artefacts into the comparison. Perhaps also good to remind the reader that 'greenness' here refers to the 0.15 threshold (related to comment above).

Thank you for the comment. The revisions stated in major point 1 and minor point 6 apply here, too.

31. 417-420, How can you demonstrate that soil ice has an additional role, additive to warming and rainratio? Aren't they just all sides of the same coin? Could it also be, for instance, that the northern regions still feature most frozen ground conditions in summer and that in southern regions, soils were already mostly above 0 degrees in the summer season, and that hence this dynamic is mostly evident in northern regions? I would carefully read this part of the discussion and evaluate which claims can be made with certainty, and which ones just reflect collinearity within the bio-climate variables.

Thank you for your reflection, which we implemented by more carefully discriminating between cause and effect. Now in *Key findings and interpretation in the context of the current literature*: Our study found that in the northern ecoregions, areas with "greening" in recent decades have experienced a rise in soil water content during the spring (SoilWaterMAM) along with declines in both springtime soil ice content trends (SoilIceMAM) and maximum snow depth (SWE_MAX). The rise in SoilWaterMAM is also accompanied by higher spring temperatures (T2mMAM) and earlier onset of the thermal growing season (Onset).

Despite regional trends on higher summer rainfall amounts (RainJJA), we did not find a clear link between greening and changes in RainJJA. Interestingly, summer soil water content (SoilWaterJJA) and soil ice content (SoilIceJJA) are negatively related to near-surface air temperatures in summer, which results as a consequence of surface thawing and subsequently increased evaporation caused by higher vapor pressure deficits in these northern areas.

The greening of the recently emerged vegetated areas in the northern ecoregions respond to different seasonal soil water contents. Greening in ecoregion 1 correlates best with SoilWaterMAM patterns, similar to the remaining southwestern ecoregions. Conversely, ecoregion 5 is more closely connected with SoilWaterJJA, likely due to a later onset of the GrowDays.

32. Overall, the discussion would really benefit from a thematic subdivision, for instance into different sets of climate variables, or into driving mechanisms and a section on how they differ among regions? Right now the reader easily gets lost between different lines of argumentation.

Thank you for your suggestion. We are confident that our attempt of separating our interpretations into thematic paragraphs could have improved the revised version. For instance, in *Key findings and interpretation in the context of the current literature*, with 1. Changes in green vegetation extent; 2. PCA performance and basis for interpretation; 3. Northern ecoregions; 4. Southern ecoregions; 5. Common features across ecoregions, 6. Drying in the interior of ecoregion 2; 7. GrowDays elevation dependence explained; 8. Changes in green vegetation distribution and in bio-climatic factors reported in literature.

33. 426 – 435, I found the descriptions of slower melt of shallower snowpacks very difficult to follow (and frankly, counterintuitive, but then I am not a snow physics

expert). Even if the melt rate is lower, wouldn't the timing of complete snowmelt still be earlier for shallow snow than for deeper snow? What then is the exact role of the slower melt rate and potentially better water absorption within the context of your findings? I have a feeling that similar claims could be made about the role of deeper snow and its impact on soil temperature and microbial activity (as you also state in the introduction), so I am still in the dark about the role of melt rate in nutrient availability. I would recommend rewriting this in a way that is more accessible to readers without a background in snow physics and staying closer to your own results.

We hope that the improved explanation on minor point 4 clarifies the relationship between snow depth and snowmelt rate better as well as it is addressed in *Key findings and interpretation in the context of the current literature* it reads that For most ecoregions in ice-free Greenland, we find that snowpacks are becoming shallower, and consequently melt slowly, but earlier in the season.

This feature was mentioned by Musselman et al. (2017) and is attributed to global warming. Musselman et al. (2017) explains that in Western North America regions with shallower snow are experiencing snow season contractions. Shallower snow is susceptible to snow season contraction because shallow snow requires less energy to initiate melt than deeper snow. This earlier start of the ablation period occurs at a slower rate due to a combination of near-surface warming with relatively low solar altitude angles.

In contrast, for deep snowpacks that require more energy to initiate runoff, it is also more likely for the snowmelt water to refreeze within the snowpack (Dingman, 2015). Therefore, early season slow snowmelt rates in shallow snowpacks allow for efficient soil water percolation and subsequent water storage (Stephenson and Freeze, 1974). The successful percolation of liquid water into soil plays a key role in tundra regions during the snow ablation period and start of the growing season, as during this time soils are generally dry due to high drainage (Migała et al., 2014).

Increased water availability in the soil could stimulate dormant microbial communities and thus increase the decomposition of soil organic matter, releasing soil nutrients (e.g., Glanville et al. 2012; Salmon et al. 2016; Xu et al. 2021). This in turn could prime the soil for earlier and more efficient vegetation growth and colonization. The increased spring soil water content (SoilWaterMAM), spring air temperature (T2mMAM), and thermal growing season days (GrowDays) indicated in our results could therefore improve conditions for plant growth and colonization, especially in the southern ecoregions. Therefore, it is expected that vascular plants are more developed in early summer. Such conditions in conjunction with summer weather patterns that favours increased T2mJJA and longer periods of solar radiation (Barrett et al., 2020), allowed for greener summer vegetation. The same summer weather patterns also brought more drought and heat days, without an immediate negative impact on greenness.

34. 428, Heijmans et al (2022) doesn't deal with the release of nutrients in relation to spring water availability. Perhaps we cite others in our review that have relevant findings on this topic, but to me this doesn't seem to be an appropriate reference here.

Thank you for the remark. Heijmans et al (2022) is a very relevant reference for us to better understand the links between tundra vegetation and permafrost changes, but in the revised version we rather refer to Glanville et al. 2012 and Salmon et al. (2016), which carries our point in a more central role.

**35. 463-465, maybe you can back up this hypothesis about the role of shrubs or potentially other species groups by checking your greenness trends against the CAVM or Karami et al. (2018)?**

The recommended sources, the Circumpolar Arctic Vegetation Maps and Karami et al. (2018), are static maps based on the collection of data over several years with different approaches. Our trend perspective is thus not directly comparable, which is why we prefer to remain closer to our focus.

**36. 475, what exactly do you mean by "validating bio-climatic indicators"? I think you could explain your proposed course of action a bit better, and also explain how that would help understand future trends.**

We wrote in the revised version in *Significance and implication*, that: Our study determines a set of bio-climatic indicators that have been shown relevant for spectral greenness. The statistical interlink among these indicators is confirmed in experimental studies across the Arctic (e.g., Chen et al. 2023; Gamm et al. 2018; Grimes et al. 2024; Huai et al. 2022; Migała et al. 2014; Musselmann et al. 2017; Opala et al. 2018; Schmidt et al. 2023; Stephenson et al. 1974; van der Schot et al. 2023), allowing the interpretation of our outcome to be expanded to large-scale, with apparent features dependent on the ecoregion and latitude. Such insights can now be used to validate whether the same bio-climatic indicators interdependence is captured by climate models. A consistent representation of past conditions would provide a sound basis for the use of such indicators for the study of future vegetation changes across Greenland under a changing climate.

**37. The implications section reads like a rather surprising selection of several implications, of which I am not really sure if all the main ones are represented, and whether the ones that are now discussed most extensively are in fact the most important ones. For example, a lot of attention is dedicated to PBAPs and fog, but no mention is made of carbon dynamics or surface energy balance feedbacks. Even if this is deliberate, it would be good to highlight why specific implications are discussed while others are not. You do mention some of these aspects in the limitations, but they are of course also relevant from an implications perspective.**

Thank you for the remark! We find relevant to keep recent literature that links primary biological aerosol particles (PBAPs) with the cloud formation in the Arctic and the potential of generating fog conditions due to decreasing sea ice as part of the Discussion. We mentioned other implications although not directly such as the feedback of the vegetation canopy on the surface feedback, shifts in cloudiness due to increased PBAPs, the cooling of the surface due to surface evaporation and ecological shifts on the animal community. However, we acknowledge that other important implications such as carbon dynamics and surface albedo feedback have not been addressed previously and are therefore included in the revised manuscript.

Longer thermal growing seasons are shown across Greenland between 1991 and 2023. Longer thermal growing seasons with higher air-temperatures favoured general vegetation growth and expansion have in the studied period. However, further investigation is required to comprehend the impacts on vegetation and ecosystem functioning in regions that have been facing freezing conditions due to earlier onset of the thermal growing season, exposed to heat stress conditions and experiencing changes in precipitation patterns. Given the reportedly significant decreases in snow cover, the surface albedo is lower for longer periods, facilitating more energy absorption and enhancing surface warming. The observed wide-spread greenness changes intensify the surface albedo feedback with varying effects that extend beyond the growing season and depend on the vegetation type (e.g., Blok et al. 2011; Loranty et al. 2011).

The surplus of the surface energy budget leads to surface warming and promotes surface thawing, particularly in the northern ecoregions. However, depending on the vapour pressure deficit and the vegetation canopy, the excess of surface energy can be used for latent heat release, which in turn will cool the surface (Heijmans et al., 2022). The increase in green vegetation drives at first to greater carbon sequestration. However, if the increase in vegetation causes substantial surface thaw, the net effect could trigger the release of carbon, offsetting the compensation of carbon sequestration from vegetation (Glanville et al., 2012).

**38. 506 – 510, I would expect that such episodes of warm, humid conditions should be evident from your PCA analysis, so I do not see the point of mentioning the role of this particular episode as a limitation?**

Indeed, the warm and humid episodes should be depicted in the PCA from the reanalysis output, but cloudiness does not allow surface reflectance retrievals, and therefore, partly hinders potential vegetation development.

**39. 517-520, needs references for the claims made. I would like to add that while permafrost thaw can indeed release moisture or lead to ponding, deeper thaw fronts also often lead to deeper infiltration and surface drying (Liljedahl et al., 2016). This section could use more nuance and backing.**

Thank you for mentioning the possibility of deep thaw fronts which could lead to deep infiltration and surface drying as described by Liljedahl et al. (2016). We added in *Study limitations and future research directions* that: better representation of the permafrost extent and active layer thickness together with the inclusion of dynamic tundra vegetation models within CARRA could be beneficial to deepen our knowledge on interactions among atmosphere, vegetation, carbon and nitrogen cycling, water and permafrost dynamics.

Permafrost areas will continue to likely be locations for future vegetation expansion (Chen et al., 2023), especially under the current trend of decreased summer precipitation. Moreover, permafrost thawed areas are also susceptible to fast drying (Liljedahl et al., 2016) and potentially sudden vegetation changes. Ultimately, plants can fixate along streams and small lakes as future land ice melt will continue to provide sediments and nutrients through runoff (Migała et al., 2014).

40. 525 – 530, I do not want to send you back to the drawing board, but I am interested why elevation was added to your analysis, while aspect and slope were not. You rightfully stress their importance and I would (perhaps naively!) assume that it would not be such an enormous effort to include them in your analysis as well?

Good point and indeed, for a wide perspective it would be useful to add slope and aspect for which we have the data and the analysis ready. We added these results to the revised version. The relationship between GrowDays and topographical features such as slope and aspect was further explored. As the surface slope is highly correlated with surface elevation, trends in GrowDays tend to significantly decrease with steepness. The dependence between GrowDays and surface aspect is rather complex, without a predominant slope orientation promoting GrowDays, in general. However, latitudes immediately south of Maniitsoq Ice Cap show increases of GrowDays in slopes with southwest orientation. On the East coast, a western slope orientation is particularly pronounced along Jameson Land, whereas northeast exposure appears favourable north of ecoregion 5.

The dependence of the slope orientation for greenness changes is partly in alignment with the dependence of the slope orientation for GrowDays. Greenness trends increased in two latitudinal bands facing southeast in ecoregion 1 and 2. In Jameson Land a similar tendency for more greening is found towards southwest, while east facing slopes are preferred towards the northern part of ecoregion 5.

Also, important to mention that Surface slope is transformed into sine aspect (west-east orientation) and cosine aspect (north-south orientation), given its circular orientation. Positive values in sine (cosine) aspect indicate how much the slope is facing east (north), whereas negative values indicate how much the slope is facing west (south). was added in Bio-climatic factors.

41. Rather than reiterate what you did, you could summarize the actual findings and try to align better with the original aims (perhaps mention which set of variables or which variables show the strongest associations?) and mention the key advance you have made? This would make the conclusion more informative.

Thank you for the suggestion. We briefly summarize our findings and keys advancements in the field in the revised version. This is answered in minor point 2.

**Technicalities & Language**

1. 10 "summer spectral vegetation". This is an unusual term, it would be good to rephrase it or explain it so that there can be no ambiguity about what it means.

This term comes from Myers-Smith et al. (2020), who refers to NDVI as a spectral vegetation index. Therefore, summer spectral vegetation (a.k.a. spectral greenness) is the seasonally averaged result mentioned in *Bio-climatic factors*.

2. 18 "by 22.5% increase" should be "by 22.5%". I also recommend to be more explicit about what you mean by "the distribution of vegetation". Do you mean that the vegetated area of Greenland (determined here as summer NDVI > 0.15?) expanded in area by 22.5%? Perhaps you want to rewrite this sentence.

Thank you. This sentence was rephrased.

3. 25, what do you mean by "regional Greenland"? Perhaps that specific regions of Greenland are warming three times faster.

Not the entire Greenland is warming. Therefore, "regional" gives emphasis only to warming locations.

4. 31, add "and" instead of comma between "composition" and "alterations".

Thank you. Done!

5. 48, is it really necessary to mention the specific methods of Gamm et al ( "using [..], [...] and [...]")? This is not done for other papers that you cite?

Not really. Thank you. Done!

6. 53-55, this reads like a repetition of L. 43-44.

Agreed. Thank you. Done!

7. 62, I do not think "snow cover melt" is a very generally used term. Maybe write "snowmelt timing" or "snow melt rate", depending on what you mean exactly?

Thank you. Done!

8. 71, maybe write "large amounts of snow" rather than "large amounts of snow coverage", since from what I understand snowpacks were also very deep, not just spatially extensive.

Thank you. Done!

9. 81-82, example of a grammatically confusing sentence.

Thank you. This and many other sentences were rephrased.

10. 83-86, implications for phytoplankton seem beyond the scope of your study system and I do not see the added value of discussing it here (it seems more of an implication rather than an example of the importance of subsurface flow to terrestrial vegetation).

Thank you. This was removed.

11. 105, add "the" between "to" and "CARRA".

Thank you. Done!

12. 132, "and thereafter is then continued" should be "and is thereafter continued".

Thank you. Done!

13. 133, add "is" between "mask" and "spectrally".

Thank you. Done!

14. Figure 5) Final sentence in the caption: Do you mean that the trend was considered significant if the 90% CI of the estimate did not overlap 0? This is what I am used to. Similar for Fig. 6

Correct. It was added to Figure 6 and to other figures where M-K trend test is used that the null hypothesis is that the slope is equal to zero.

15. 376, replace "evidence" with "shows"?

Thank you. Done!

16. Table 2) perhaps a no brainer, but it would be good to explain what the fraction mean; is this % of total area of that ecoregion?

Thank you. Done!

17. 446, change "favourable areas" into "a more favourable area".

Thank you. Done!

18. 498, change "as" into "as in"

Thank you. Done!

Thank you for these valuable edits which we incorporated in the revised manuscript.

**General Comments**

I commend Silva et al. for their ambitious analysis of a substantial amount of data from a sensitive ecosystem of broad scientific interest. For this reason, it is my opinion that the study's aim is well suited for the readership of Copernicus Biogeosciences and is an important undertaking. However, I have major concerns about the implementation of methods and the interpretation of results. Most

importantly, there is a critical misalignment between the stated goals of the study and the methods used to achieve these goals (as well as the title of the paper).
To summarise my concerns: The authors sought "to gain a deeper understanding of the spatio-temporal patterns of spectral vegetation changes across ice-free regions of Greenland (ln 90)" and "examine the combined effects of bio-climatic indicators ranging from sub-surface factors (such as soil water availability) to above-surface factors (such as the thermal growing season, heat stress, and frost) with summer spectral greenness (ln 91 - 95)." However, the authors provide contradictory statements about the goals of the PCA. Throughout the paper they explicitly state that they use PCA to assess drivers of *changes* in NDVI over time within a pixel, as well as having used PCA to assess drivers in changes of greenness *distribution*. Reviewing the methods and results of the PCA, it seems that the dimensionality reduction algorithm was actually used to assess bio-climate indicators that correlate with average summer spectral greenness ("greenness distribution"). I elaborate on these concerns below.

We would like to thank the reviewer for the general appreciation of our work. We can understand the confusion from the reviewer's point of view regarding the general objectives of our work. These are clearly not well reflected in the paper. We have therefore clarified exactly these points in the revised *Abstract*:
We use principal component analysis (PCA) to examine key sub-surface and above-surface bio-climatic factors influencing ecological and phenological processes preceding and during the thermal growing season in tundra ecosystems. Subsequently, we interpret spatio-temporal interactions among bio-climatic factors on vegetation and investigate bio-climatic changes dependent on latitude and topographical features in Greenland. Ultimately, we identify regions of ongoing changes in green vegetation distribution.
While we derive the spatio-temporal change in spectral greening from time series of the NDVI (trend analysis), PCA is used to reduce the dimensionality of the vector space of the explanatory bioclimatic indicators. We can then use the bio-climatic indicators with the greatest explanatory value to explain and try to understand changes in spectral greening from the temporal changes among indicators on regionalised scales of ice-free Greenland. The causal PCA outcome is treated with caution, and therefore, we added to the *Statistical Methods*:
We attempt a careful causal interpretation of the loading vectors from the first two principal components (PCs) of the PCA through biplots (Gabriel, 1971). Although these PCs account for most of the explained variance, their interpretation in terms of causality is limited by the nature of PCA as a descriptive statistical technique. For a cautious interpretation of the PCs, we examined not only the magnitude and direction of the loading vectors, but also trend maps of the involved bio-climatic indicators and literature on experimental studies
In the *Discussion*, we present and expand our interpretations in *Key findings and interpretation in the context of the current literature* and in *Significance and implication*, that: Our study determines a set of bio-climatic indicators that have been shown relevant for spectral greenness. The statistical interlink among these indicators is confirmed in experimental studies across the Arctic (e.g., Chen et al.

2023; Gamm et al. 2018; Grimes et al. 2024; Huai et al. 2022; Migała et al. 2014; Musselmann et al. 2017; Opala et al. 2018; Schmidt et al. 2023; Stephenson et al. 1974; van der Schot et al. 2023), allowing the interpretation of our outcome to be expanded to large-scale, with apparent features dependent on the ecoregion and latitude.

**Specific comments**
Major Concerns
Thank you for the comprehensive statement. We will break the explanation of point 1 into several sub-points.
1) There is a critical misalignment between the stated goals of the study and the methods used. In Section 3.4, the authors mention that "PCA was used to investigate the combined influence among bio-climatic indicators on summer greenness _changes_" (ln 249-250; _emphasis added_). However, in the Results section, it is stated that "PCA was used to investigate the combined influence among bio-climatic indicators with summer greenness" (ln 307), which suggests an analysis of greenness levels rather than _changes_ in spectral greenness.

This is a very valid point, and we apologize for having caused confusion. Our intention was the same in both sentences. The PCA encompasses data since 1991 to 2023. This means that the PCA outcome is a statistical result of the biosphere-atmosphere-cryosphere interactions over the three decades in study. By colouring each score with its corresponding greenness, we show that the densely vegetated/greenest regions are clustered by the first two principal components as a result of "the combined influence among bio-climatic indicators with summer greenness". This outcome is only achieved by considering the spatio-temporal changes of all bio-climatic indicators, where greenness is included. We changed both instances in the revised document to the combined influence among bio-climatic indicators with summer greenness

This discrepancy is further supported by the caption for Figure 4, which notes that the biplots' scores "are colour-coded based on the summer spectral greenness as in Figure 1," where spectral greenness is defined as the "averaged spectral greenness (based on the period 1991-2023) for June, July, and August."

We apologize for the misunderstanding caused in this specific sentence from the Figure 4 caption. We removed the reference to Figure 1 and simplified the sentence to The scores are colour-coded based on the summer spectral greenness, with different scales to enhance greenness. While Figure 1 shows 32 years of monthly averaged spectral greenness, the coloured scores in Figure 4 correspond to summer spectral greenness for each year between 1991 and 2023. We also added colour scales in Figure 4 to better distinguish greenness across ecoregions, as later suggested by the referee.

Additionally, greenness is included in the PCA but defined differently as "seasonally averaged monthly NDVI," a quantity briefly mentioned in Section 3.1.

Monthly averaged NDVI is used in Figure 1 to show the evolution of spectral greenness in summer in each ecoregion. This is particularly important for readers without knowledge of the greenness dynamics over summer across ice-free Greenland.

In *Spectral Greenness* we state that "we calculated a seasonally averaged NDVI, hereafter referred to as spectral greenness and interchangeably as green vegetation." And in *Bio-climatic factors* we write "Spectral greenness, T2m, RainRatio, the volumetric soil water and ice (SoilWater and SoilIce) and vapour pressure deficit (VPd) are seasonally averaged, whereas precipitation, snowfall (Snow) and rainfall (Rain) are seasonally accumulated." And later in the same section, we write "Spectral greenness was compiled for summer, in order to capture the period with maximum solar radiation in Greenland and avoid snow-covered patches"

The authors highlight that PC1 and PC2 "largely capture and explain Greenness distribution" (ln 320-321), suggesting a focus on greenness levels rather than changes.

The orientation of the loading vectors along with the greenness distribution in the biplots and the supplementary maps (Fig. S10-S14) show that the scores with high spectral greenness are grouped in areas of low elevation (PC1>0) with varying degrees of influence in precipitation and snow patterns (PC2).

It may be possible that the inclusion of "changes" in lines 249-250 was unintentional. However, the broader context suggests that the issue extends beyond a simple wording error. The title ("Bio-climatic factors drive spectral vegetation changes in Greenland"), the abstract (ln 10-15: "GrowDays... emerged as the pivotal factor across all ecoregions...to promote vegetation growth."), and the discussion (e.g., ln 417-419: "Our [PCA] results suggest that in the northern ecoregions, the reduction in soil ice during summer...is enabling vegetation growth, leading to northward expansion of vegetation." and ln 433-435: "The combined effect of soil nutrients with increased soil water availability in spring (SoilWaterMAM) and T2mMAM, promotes early plant growth. Therefore, leaves are more developed in early summer, which in association with increased T2mJJA and longer periods of solar radiation, allow for greener vegetation.") all imply a focus on changes in greenness values over time.

Thanks for highlighting this confusing point. Indeed, we look both at the state as well as the changes. Our *Results* aim to investigate spatio-temporal changes in greenness and its co-variability with atmospheric and snow indicators from 1991 to 2023. The monthly averaged greenness state is only shown in Figure 1.

The statement raised "GrowDays emerged as the pivotal factor across all ecoregions" in the comment is based on the interpretation of the biplot loading vectors and the relative importance of the loading vectors shown in the supplementary Figures S10-S14. We developed our reasoning in *Key findings and interpretation in the context of the current literature*. There, we write The rank of relative importance of individual bio-climatic indicators depends on ecoregion, with the number of days of the thermal growing season (GrowDays) being the most relevant across all ecoregions, followed by soil ice during summer (SoilIceJJA) in the northern and SoilWaterMAM in the southern ecoregions.

And later:

the early onset of GrowDays allows vegetation to be potentially more active and responsive to solar radiation, particularly in the ecoregions in lower latitudes with longer sun exposure.

We also improved our explanations on how biplots are interpreted in *Key findings and interpretation in the context of the current literature*. There, we write: For most

ecoregions in ice-free Greenland, we find that snowpacks are becoming shallower, and consequently melt slowly, but earlier in the season.

This feature was mentioned by Musselman et al. (2017) and is attributed to global warming. Musselman et al. (2017) explains that in Western North America regions with shallower snow are experiencing snow season contractions. Shallower snow is susceptible to snow season contraction because shallow snow requires less energy to initiate melt than deeper snow. This earlier start of the ablation period occurs at a slower rate due to a combination of near-surface warming with relatively low solar altitude angles.

In contrast, for deep snowpacks that require more energy to initiate runoff, it is also more likely for the snowmelt water to refreeze within the snowpack (Dingman, 2015). Therefore, early season slow snowmelt rates in shallow snowpacks allow for efficient soil water percolation and subsequent water storage (Stephenson and Freeze, 1974). The successful percolation of liquid water into soil plays a key role in tundra regions during the snow ablation period and start of the growing season, as during this time soils are generally dry due to high drainage (Migała et al., 2014).

Increased water availability in the soil could stimulate dormant microbial communities and thus increase the decomposition of soil organic matter, releasing soil nutrients (e.g., Glanville et al. 2012; Salmon et al. 2016; Xu et al. 2021). This in turn could prime the soil for earlier and more efficient vegetation growth and colonization. The increased spring soil water content (SoilWaterMAM), spring near-surface air temperature (T2mMAM), and lengthening of the thermal growing season (GrowDays) indicated in our results could therefore improve conditions for plant growth and colonization, especially in the southern ecoregions.

As the analysis currently stands, PCA is used to assess the variation in climate variables, which is then visually compared to average summer greenness from 1991-2023 with biplots. Separately, the authors explore trends in vegetation expansion using Mann-Kendall tests and thresholds of NDVI between two discrete periods (1991 – 2007 and 2008 - 2023). Despite a lack of generative or predictive models linking these two goals, the authors then interpret PCA loading vectors as "explaining" changes in greenness and greenness distribution. It is also not clear to me how the authors made these interpretations; I speculate this was done by visual comparison of the maps of PCs in the supplementary material with the maps of greenness distribution and greenness change over time in Figure 6.

The referee is partly correct on the speculation on how the results are interpreted. Additional to the referred procedure, we also use the information in Figure 6c to mask trend maps and better understand trend directions among bio-climatic indicators. This was essential to better understand how temporal changes among bio-climatic indicators are interlinked in the newly emerged vegetated areas, as described in *Key findings and interpretation in the context of the current literature*: Our study found that in the northern ecoregions, areas with "greening" in recent decades have experienced a rise in soil water content during the spring (SoilWaterMAM) along with declines in both springtime soil ice content trends (SoilIceMAM) and maximum snow depth (SWE_MAX). The rise in SoilWaterMAM is also accompanied by higher spring temperatures (T2mMAM) and earlier onset of the thermal growing season (Onset).

Despite regional trends on higher summer rainfall amounts (RainJJA), we did not find a clear link between greening and changes in RainJJA. Interestingly, summer soil water content (SoilWaterJJA) and soil ice content (SoilIceJJA) are negatively related to near-surface air temperatures in summer, which results as a consequence of surface thawing and subsequently increased evaporation caused by higher vapor pressure deficits in these northern areas.

The greening of the recently emerged vegetated areas in the northern ecoregions respond to different seasonal soil water contents. Greening in ecoregion 1 correlates best with SoilWaterMAM patterns, similar to the remaining southwestern ecoregions. Conversely, ecoregion 5 is more closely connected with SoilWaterJJA, likely due to a later onset of the GrowDays.

The use of generative or predictive models goes beyond the scope of this study.

We hope that with the explanations provided and the revisions made are clearer to the referee, particularly on how PCA and trend analysis were used and interpreted.

2) Loading vectors should not be interpreted causally in the way the authors have. While it is true that alignment between two loading vectors indicate correlation and orthogonal vectors are uncorrelated, PCA is a function purely on a matrix of features without explicit regard for response variables. Since PCA is generally used for dimensionality reduction, data compression, or exploratory analysis, its application to infer causal relationships between bio-climatic factors and greenness requires further qualification. If the goal is to assess the relative importance of climate variables on changes in greenness, a causal (or at least an interpretable predictive) model is required.

Thank you for the remark. We agree that Greenness is not a response variable due to the reasons mentioned in point 1. While we were overconfident in some of the formulations in the first submission, we adopted a more defensive wording – alluding to evidence that is in line with literature. The careful interpretation of the loading vectors from the first two principal components was mentioned in the General Comment of the referee.

3) The inclusion of "seasonally averaged" spectral greenness as a feature in the PCA and then coloring the scores in the biplots of Figure 4 based on average summer spectral greenness over the growing seasons (1991-2023) raises concerns about circular reasoning. Further clarification on how this aspect was handled could help alleviate these concerns.

Indeed, there is room for clarification. We rewrote the caption and implemented scalebars in every subpanel to avoid further misunderstandings. The PCA is performed for summer greenness pixels available in every ecoregion between 1991 and 2023. The corresponding colouring indicates the greenness of each pixel in a particular year. The co-variability of greenness with the remaining components is shown on loading vector 5 and the colouring of the scores helps to better understand how greenness is distributed along PC1 and PC2 space, and geographical space with the support of supplementary figure as mentioned in the General Comment.

[Figure]

*Figure 1 Biplot for scores between 1991 and 2023 for each ecoregion. The loading vectors are labelled and scaled by the maximum of each principal component. The scores are colour-coded based on the summer spectral greenness, with different scales to enhance greenness. The explained variance of the first (PC1) and second (PC2) component is labelled in the corresponding axis of the subplot. The 16 bio-climatic indicators are 1: maximum snow water equivalent (SWEMAX); 2: total number of thermal growing days (GrowDays); 3 and 4: start (Onset) and termination (End) of GrowDays; 5: summer spectral greenness (Greenness); 6: rain in sumer (RainJJA) 7 and 8: averaged rain ratio in summer (RainRatioJJA) and autumn (RainRatioSON); 9, 10, 11, 12: averaged 2-m air-temperature in winter (T2mDJF), spring (T2mMAM), summer (T2mJJA) and autumn (T2mSON) 13 and 14: volumetric soil water in spring and (SoilWaterMAM) autumn (SoilWaterSON); 15 and 16: volumetric soil ice in winter and (SoilIceDJF) summer (SoilIceJJA). The abbreviations of the bio-climatic indicators are described in Section 3.2 and in Table 1. The spatial pattern of the averaged 1991–2023 scores for both components in every ecoregion, including their corresponding loadings, are shown in Fig. S10-S14.*

4) Generally, the methods are not described in enough detail. In addition to my confusion about the methods as described above:

4a) I agree with a note from another reviewer, the calibration procedure addressing potential systematic biases between AVHRR and VIIRS NDVI should be elaborated.

Thank you for raising these concerns.

This is a very important point, namely the homogeneity of the two NDVI time series to each other and what this means for a calculated trend.

The NOAA Climate Data Record (CDR) of AVHRR NDVI - Version 5 and the NOAA CDR of VIIRS NDVI - Version 1 are developed by Eric Vermote and colleagues (Vermote et al. 2018 and 2022) for NOAA's CDR Program. Both records have been processed considering the same atmospheric characteristics as in Miura et al. (2012) and both processed records are posterior to Miura et al. (2012) proposed correction. However,

the correction proposed by Miura et al. (2012) is not assessed in polar regions, which may contribute to additional uncertainties in our study.

Unfortunately, no overlap periods are available for the parallel measurements of the two satellite sensors. Therefore, no systematic differences can be determined. However, as we state in *NOAA Climate Data Record for Normalized Difference Vegetation Index* there is work that has calibrated the utilized AVHRR product with MODIS (e.g. Franch et al., 2017) and thus improved the internal homogeneity of AVHRR, as well as work that has established the homogeneity of VIIRS with MODIS (Skakun et al., 2018) to improve the consistency of the NDVI datasets. This does not yet achieve perfect homogeneity, which we explain in the description of both products in *NOAA Climate Data Record for Normalized Difference Vegetation Index as follows*: According to AVHRR and VIIRS technical reports, the NIR channel is centred at different wavelengths (830 nm vs. 865 nm). As there is no overlapping period available in the NOAA CDR, potential mismatches between AVHRR and VIIRS NDVI cannot be discarded.

We also investigate at the start of *Results* how summer spectral greenness statistically relates with climate oscillations (e.g., Greenland Blocking Index) for AVHRR, VIIRS and the entire study period. We use these climate oscillation time-series, that are homogenous and independent of spectral greenness, as a reference to evaluate systematic inconsistencies that may arise due to sensor change.

It should be noted that prevailing weather patterns during summer months, like the North Atlantic Oscillation (NAO) and the Greenland Blocking Index (GBI), are highly correlated with spectral vegetation (Fig. S7). Therefore, summer weather patterns can accelerate or delay the maximum green vegetation extent given their link with temperature and precipitation. Correlations between green vegetation extent and summer GBI are investigated for three periods: AVHRR (1991-2013), VIIRS (2014-2023) and the full period (1991-2023), and are shown in Table S1. Positive and significant correlation coefficients ranging between 0.5 and 0.8 are found between ecoregion 1 and 4, generally with higher correlations for VIIRS than for AVHRR period. Green vegetation extent in ecoregion 5 is poorly correlated with the prevailing weather patterns during summer.

While the AVHRR 22-year trend evidence general expansion of green vegetation, the VIIRS 9-year trend evidence decreases, particularly in West Greenland (Table S2). However, due to high variability and small sample size, most trends in both periods are not significant.

We address in *Study limitation and future research*, our concerns about the reliability of long-term time integrated NDVI analysis

The NDVI datasets employed in this study are sourced from two satellite products processed by NOAA, each utilizing a different type of sensor. Due to the absence of a temporal dataset overlap, the assessment of uncertainties was limited and potential for mismatches between the datasets cannot be discarded. This lack of a common calibration period raises concerns about the reliability of long-term time integrated NDVI analysis.

In the end, we follow similar approaches of recent literature (e.g., Madson et al. 2023, Pourmohamad et al. 2024) that make use of the full AVHRR NDVI and VIIRS NDVI without additional corrections.

4b) The calculation of "seasonally averaged NDVI" is somewhat unclear. I assume this involves averaging monthly NDVI across the growing season, but further explanation would be helpful.

We have also revised the same subsection in general to better understand the processing of two NDVI data sets and the problems of data homogeneity. We also reformulate in *Spectral greenness* how spectral greenness is derived from the AVHRR and VIIRS NDVI to better explain how the shaded area later shown in Figure 2 is calculated:

As estimates integrated through time are less likely to be influenced by temporal sampling artefacts at high latitudes than metrics based on maximum NDVI (e.g., Myers-Smith et al. 2020), we started by calculating monthly integrated NDVI. Also, since our focus is on green vegetation, only daily NDVI pixel values with higher or equal to 0.15 are considered. Then, we divide the monthly integrated NDVI by the total number of monthly observations (n, see Figure S1 for the interannual variability of n) to obtain the monthly NDVI. However, before 2014 and as described in Subsection 2.2, the AVHRR algorithm was less strict in its data quality control compared to VIIRS from 2014 onward, resulting in higher n before 2014 that lowers monthly NDVI. To address temporal heterogeneities, we adjusted n from the AVHRR period with the number of monthly observations acquired during the VIIRS period. From 2014 to 2023, we identified the minimum, maximum and average number of observations for each month. Hence, using these three quantities, we generated a consistent variability range from 1991 to 2013 to recalculate monthly NDVI, considering a similar number of observations as from 2014 to 2023. This procedure assumes that the environmental conditions (i.e. snow-cover, clouds and shadow) between 1991 to 2013 are similar to those between 2014 and 2023. The maps for the average number of monthly observations and the associated standard deviation for AVHRR and VIIRS period before and after the adjustment regarding n are shown in Figures S2-S5, respectively.

4c) Given the potential impact of cloud cover and other factors on NDVI observations, more information on how observation frequency (described as "n" in Section 3.1) was used to assess uncertainty and uneven sampling would strengthen the analysis. This seems like it was at least tangentially covered given the brief mention of this in Section 3.1 and the first figure in the Supplementary Materials -- but more explanation of the procedures is needed.

The point is addressed and answered in the previous sub-point.

4d) More details on the PCA and Mann-Kendall implementations would also be valuable. For example, when using scikit-learn for PCA, describing the optimizer and input data shape would help ensure transparency, as some solvers are better optimized for particular data configurations. Similarly, the choice of the standard Mann-Kendall test variant in pyMannKendall should be justified, especially regarding serial autocorrelation, which is an important consideration in trend analysis. While MK tests are the current state of the art for landscape-scale analysis like this, pyMannKendall offers options that seek to account for autocorrelation, and

discussing whether this was assessed in the data would clarify the robustness of the trend analysis.

Thank you for the request. We acknowledge that we did not include all the necessary information for reproducibility, with indication of the optimizer and input data shape, but now in the *Statistical Methods* we write The PCA (Pedregosa et al., 2011) solver was selected based on the input data shape. As the number of features in the input data is much less than the number of samples (geographic pixels), a classical eigenvalue decomposition on the covariance matrix was run. and We used the non-parametric Mann-Kendall (M-K) trend test (Hussain and Mahmud, 2019) to assess trend monotonicity and significance among bio-climatic indicators. However, to acknowledge autocorrelation in the greenness data, we computed the Hamed and Rao modified M-K test (Hamed and Rao, 1998), with a variance correction approach considering all significant lags to improve trend analysis.

**Minor Concerns & Technical Corrections.**
In addition to minor concerns pointed out by another reviewer, there are some instances of speculation that are not supported by the PCA analysis in the results section which should removed, or moved to the discussion section and include citations. These are also specific examples of where I think a inappropriate causal interpretation of loading vectors has occurred (Major Concern 2). For example:

- (ln 326) "The decreasing trend of snow rates (SnowDJF and SnowMAM) has led to SWEMAXDOY to occur earlier. Despite the increasing trend in T2mMAM, the still-low solar elevation and the still-low near-surface air-temperatures result in low melting rates of the snowpack (MeltRate). These slow melt rates favour slow meltwater percolation (SoilWaterMAM loading vector opposite to MeltRate loading vector)."
We reformulated this statement in the *Results*, referring to the trend maps (seasonal accumulated snow, SWE_MAX DOY, MeltRate and SoilWaterMAM) in the Supplementary Material of the revised manuscript. The speculative part of this statement was rephrased and moved to *Key findings and interpretation in the context of the current literature*.

- (ln 329) "Additionally, the earlier onset of the thermal growing season allows vegetation to produce energy via photosynthesis, particularly in the ecoregions in lower latitudes with adequate 330 sun exposure (Onset loading vector opposite to Greenness loading vector)."
Thank you for the remark. This speculative sentence was written and moved to *Key findings and interpretation in the context of the current literature*: the early onset of GrowDays allows vegetation to be potentially more active and responsive to solar radiation, particularly in the ecoregions in lower latitudes with adequate sun exposure (Opala et al. 2018).

- (ln 333) "Therefore, increases of RainRatioJJA promote high greenness (aligned loading vectors), as vegetation in such environmentally harsh places likely developed mechanisms to effectively retain/absorb liquid water whenever possible."

Thank you for the remark. This speculative sentence was rewritten and moved to *Significance and implications*: Water droplets from fog can effectively be retained by tundra vegetation and are not accounted as a water source. This interaction between fog, vegetation and soil conditions should be better investigated particularly for coastal tundra vegetation.

This sentence is a tautological argument:
(ln 465) "The wide-spread summer spectral greening occurs as a result of greener vegetation as certain sites."
Thanks, this was misleading indeed. We adapt to: The widespread summer spectral greening could be due to encroachment of vegetation on previously bare surfaces and changes in plant community composition at certain sites (Grimes et al. 2024).

The importance of solar radiation exposure is described as important in several places, including the conclusions, but are not included explicitly in the PCA or other analyses (ln 327, 435, 534).
The exposure to solar radiation is not considered as the NDVI is only available when there is solar exposure. However, we make use of relevant metrics of the atmospheric circulation patterns in the vicinity of Greenland that promote cloudless conditions (e.g., positive phase of the Greenland Blocking Index, GBI). That is why in Figure 1, we correlated summer greenness with summer GBI, where we report high positive correlations across all ecoregions. Therefore, we cannot discard the role of the interannual variability of atmospheric circulation patterns on greenness, as the previous decade (since the 2010s) was composed by more frequent cloudless conditions in summer (Silva et al. 2022).

Figure 4 - It would be helpful to readers if the PC1 axis was flipped for Ecoregion 2 and 4 so that the quadrants with higher greenness scores were all in the same vicinity in the biplots across Ecoregions.
Thanks for the remark! We flipped the axis for the same orientation across ecoregions.

The color palettes in Figures 5 and 6 rely on a reader's ability to distinguish red and green, which is a common color-blindness.
Thanks for this advice. We value inclusivity and attempt to open up wherever we can. The colormaps were checked prior to submission following the Copernicus manuscript preparation style and Coblis – Color Blindness Simulator. All figures are supposed to be colour-blind friendly, except for monochromacy.

Grammar checks needed throughout.
Thanks for the remark! We revised and improved the grammar.

References:

[revised manuscript text omitted]

---

## Referee Report (RR1)

Dear authors, dear editor,

Thank you for your response to my earlier comments, and for all the edits to the manuscript. The authors have made substantial improvements in avoiding speculation outside of the discussion paragraph, providing essential details on the processing of data to ensure transparency and reproducibility, and have even added novel analyses and supplements. There is now a clearer explanation of the assumed mechanisms behind the observed patterns. Improvements have been made in phrasing more clearly which statements are interpretations, and which are actual findings of the study.

I see the scientific value in this large-scale and holistic evaluation of climate – vegetation growth relations that the authors provide for Kalaallit Nunaat / Greenland. I think Biogeosciences is an excellent platform for such a large-scale and interdisciplinary endeavour.

I do however still have several concerns and in my vision the manuscript is not publishable in its current form. I hope this next revision round will be helpful to get this work published and I would like to kindly ask that the authors are a bit more thorough this time. All line numbers refer to the text without tracked changes.

Please keep working on this! Wishing you lots of success!

**Major to moderate concerns**

1. I agreed with the second reviewer that the aims reported in the introduction, the methods, and the conclusions mentioned in the abstract and conclusion, were not yet fully aligned. While improvements have been made, several issues still remain. See minor comments. I particularly have a concern about the second aim: "We examine the combined effects of bio-climatic indicators ranging from sub-surface factors (such as soil water availability) to above-surface factors (such as the thermal growing season, heat stress, and frost) with summer spectral greenness." (L. 114-116). I would assume that you mean "combined effects […] ON summer spectral greenness" rather than "[…] WITH spectral greenness"? More appropriately, I would write "association with" rather than speak of "effects on" in this correlative study. In my opinion, this would be more aligned with running a PCA that includes greenness. If the aim is to study combined effects of bio-climatic indicators on greenness itself, then a multivariate method that models greenness as a response variable (such as pls regression) would be more appropriate.

2. The other reviewer and I both indicated that improvements were necessary throughout the ms in terms of sentence structure, grammar and language errors. Many improvements have been made, but there are still too many examples of grammatical errors and typo's, particularly in the newly written text. I have made many suggestions in the previous round of editing, but ultimately I believe reviewers should focus on the scientific aspects of a ms, and not provide free language editing services. Well-structured and unambiguous sentences are necessary to convey complex scientific matter to the broad and interdisciplinary readership of a journal like biogeosciences. The authors will have to do thorough language checks, ideally by a native speaker, or potentially use AI based language improvement tools. This also goes for any newly edited or added text.

3. The authors have added extended definitions of the different terms they use for various manifestations of greenness. This does not mitigate the fact that there is an excessive amount of different terms in use throughout the ms, and that in some cases two or more different terms are used for the same thing. Please add specific and singular definitions under 3.1. I would assume that you would need only about 3 terms ("greenness" -> seasonally averaged NDVI, please add time range in months. "seasonal duration of greenness" -> amount of months within the season that NDVI was > 0.15. and perhaps another term along the lines of "green pixel" to indicate pixels with NDVI > 0.15…?). Perhaps use either "greenness" or "spectral greenness" but

not both. These are suggestions of course; the bottom line is that all terms need to be defined clearly, and that there needs to be a stricter limit to the amount of different terms in use. *Example*: L. 213-2014 "Finally, we calculated a seasonally averaged NDVI, hereafter referred to as spectral greenness and interchangeably as green vegetation" -> it is unnecessarily confusing to use two very different terms for the same thing.
*Another example*: the terms "extent of green vegetation", "green vegetation extent" and "spectral vegetation extent" are all used interchangeably. Please pick one. All the different terms in use really mean and imply different things from ecological perspectives and remote sensing perspectives and correct terminology matters in this context (if necessary, consult review by isla myers-smith, 2020, already cited). Please critically evaluate the whole ms including figures labels, tables and captions for greening terminology, simplify and realign thoroughly.

4. The authors replied that they disagree with my request to add scale bars for "greenness" in figures, since it refers to a 32year average of the mean JJA NDVI. If you judge that a 32 year average is not informative, then I would wonder why you choose to depict it at all. Please either remove the figures completely, if you judge that the greenness patterns are not useful for interpretation, or add numbers. As a reader, I want to know how to place this information in the context of panarctic ndvi values and trends, data from different sensors, saturation values and the thresholds that you use for defining "greenness". It is also basic cartography and potentially a journal requirement that continuous scale bars include numerical labels.

5. The authors replied to my earlier request to implement thematic discussion points in the ms by explaining (only in the response) what the paragraphs are about. Please make sure that this rationale is actually visible within the ms itself. You could for example add sub-headers for several main lines of interpretation so that the reader knows what is going to be discussed, and what is relevant for them to read, instead of seeing one large page of text. Many readers will not read the discussion from beginning to end, but rather focus on specific aspects that are of interest to them. Please implement a more logical flow to the information presented in the discussion. (And a general request: make sure all comments are met by changes in the actual ms itself and cite the line nrs. Or explain and argue why no changes were made).

6. Paragraph 5.1, general: I now better understand the different dynamics of earlier season, slower melt of shallow snowpacks, and later season melt under warmer conditions in deep snow. Thank you for elaborating. This helped a lot. Throughout paragraph 5.1, I could not help but wonder to what extent the observed increases in soil water content in spring (MAM), and association thereof with greenness, are not simply a result of the soil's increasingly unfrozen state (due to shallow snow that melts early, as well as warming earlier in the season). Would this not also result in an increasingly unfrozen state of Greenlandic soils in spring, corresponding to higher water content and lower ice content in spring? Or can you provide additional argumentation, that clearly shows that increased spring soil water content is indeed the result of slower melt dynamics of shallower snow packs? This needs more argumentation.

**Minor comments**

1. L. 105: here "permafrost thaw" was replaced with "ground thaw", for reasons that aren't clear from the response. I find this new term unnecessarily vague, and I would recommend to change it to "permafrost thaw" or "ground ice degradation", depending on whether the authors are referring to seasonally frozen water resources in the active layer, or availability of new moisture resources from degrading, ice-rich permafrost.
2. L. 119 "greenness distribution"; seasonal or spatial distribution? Such nuances are important, especially given the many different terms and definitions of greenness that are used throughout the manuscript.

3. Line 133-136: Thank you for adding this critical information on how soil water and ice content are derived. Can you also add information on the accuracy of these subsurface components of the re-analysis products? Otherwise it remains a bit of a black box whether the reported associations between snow, soil water and greenness variables accurately reflect real-world processes, or whether they simply result from the way that the different sub-models are defined (with risk of circular reasoning).

4. L. 161: "likely signifying areas with no vegetation presence". I think this part of the sentence should be removed; vegetation can be present under clouds, cloud shadows or seasonal inundation. Or even under snow (for instance, bryophytes and evergreens under snow may not be photosynthetically active at that moment, but they are still there).

5. Table 1) one of the indicators in "SWEmax", but it just gives the snow water equivalent at any give moment as far as I can judge from the definitions. Don't you mean "SWE" here, as also referred to later in the ms? If you mean maximum SWE, add to the definition that this is a seasonal maximum?

6. In a more general sense, it is rather inconsistent among Table 1 and the text under 3.2 whether and how seasonal integration/averaging is mentioned. This is explicitly mentioned in the table for variables like T and greenness, but not VPD (for which it is only mentioned in the text). Please find a way to make this clearer. I recommend mentioning your definition of what "seasons" are before the table, then listing descriptions in the table and adding a column to the table that explicitly states whether it is an annual variable (starting when? Previous autumn or previous winter?) or a seasonal variable, and which seasons are considered or not (e.g. some are only calculated for spring/summer and others only for winter, which makes sense).

7. Line 285-286: "The combination of this region's complex topography with frequent cloud cover resulted in its exclusion from the analysis". Many outcomes for this ecoregion are still reported (Fig. 2, Fig. 4 and associated text, discussion). Please explain why such findings are still presented despite the limitations mentioned here.

8. Line 303-310: This is background information on the NAO and GBI, not a method. Please integrate into the introduction of the role of NAO/GBI in the introduction and use the methods section only to describe what you did yourself.

9. L. 313 – 315; again use of the phrase "influence with", which is grammatically incorrect and methodologically confusing. PCA cannot demonstrate influences, only associations. It also seems incorrect to me to speak of an influence when greenness is treated as one of the variables going into the PCA, rather than a response variable.

10. L. 331: "This will diminish noise"; please avoid different tenses in the methods. Everything else is in past tense, so please rewrite to past tense and check tenses throughout the manuscript for consistency within paragraph.

11. L. 354-355: ", due to the typically shallow snow cover". This is an example of an interpretation in the results that is not backed by a figure or a statistic. Please avoid such interpretation while reporting the outcomes of your analyses, or back them up with evidence. Please recheck the results section for such interpretations.

12. L. 359-360: "Correlations between green vegetation extent and summer GBI are investigated for three periods: AVHRR (1991-2013), VIIRS (2014-2023) and the full period (1991-2023), and are shown in Table S1.": example of methods, mentioned under results. Please move to methods.

13. L 360 -370: many grammatical issues (tenses, plural/singular). Please carefully check whole ms.

14. L. 375-395: some variables are removed from PCA analysis based on their degree of association with greenness. This is a methodological choice that needs to be described and backed up in the methods, based on *a priori* informed criteria. No threshold value (in correlation or p value) is mentioned at all, making the choices seem arbitrary (even though they are probably not). Please describe all choices and criteria in the methods (as I requested in the previous revision round), so that here you can stick to reporting the outcomes, and you only need to mention which variables make the benchmark for inclusion.

15. L. 396: please rewrite all instances of "influence with" and clarify in the aims whether this analysis is meant to show influences (= impossible with PCA) or associations.

16. L. 427 – 434: Here new analyses (methods) are introduced in the results. Please move to methods and strictly report outcomes in results.
17. L. 440: "strips" instead of "stripes"? Strips of what?
18. L. 471-473: another example of methodological information in the results. This is already described, in different wording, in Line 210-ish. Please integrate this information there and report only the outcomes here. Adding just a little sentence as a reminder of what you did as a guideline for the reader should be ok, but restating the whole processing approach is excessive.
19. L. 476: as I mentioned in the previous round of revisions, calling this "expansion of vegetation" is very misleading to all readers with an ecological background, since vegetation expansion is more or less exclusively used for spatial expansion. If this is about temporal expansion of photosynthetic activity, then please name it something else, like "extension of the growing season".
20. Table 2: the methods state that this is to be referred to as "changes in greenness distribution" later in the ms, but this does not seem to be implemented, since the results only mention greening "expansion" and "shrinkage", interchangeable with "vegetation" expansion/shrinkage. Or sometimes "reduction". This is an example of the major comment on greening terminology. Please use uniform terminology.
21. Discussion, general: There are many grammatical errors in the newly added text. The discussion really needs to be evaluated for language.
22. L. 499-505: This paragraph needs backing from literature (on the blocking events) and reference to figures and tables.
23. L. 518-529: Please tone down your statement that observed dynamics are a consequence of permafrost thaw (this is an interpretation, and cannot be described as a causal influence without further backing).
24. L. 531-533: Here you mention "spring (winter)" and "winter (spring)". Please elaborate on what you mean by this. Do you mean something like "in winter, and to a lesser extent in spring"?
25. L. 595: Why shrubs, specifically, and not other plant functional groups? Please explain in the ms.
26. L. 628: "longer roots". Please write "deeper roots". Roots can be long without extending deeply. Perhaps write "deeply rooted species" (not only graminoids) and cite papers on actual root development under warming rather than this model study (van der kolk) based on assumed vegetation growth parameters rather than actual observations. Suggestion: https://besjournals.onlinelibrary.wiley.com/doi/abs/10.1111/1365-2745.12718
27. L. 630: "This ecological shifts might will also affect".
28. L 660-661: "Such periods have favoured exceptional vegetation growth across western ecoregions as shown in our results". In your previous response letter you stated that cloudiness prevented you from actually inferring whether warmer and more humid conditions during such events are indeed associated with greening. So I am a bit confused now. Please refer to the specific results that support this claim.
29. L. 682-685: These are implications, not recommendations or limitations.
30. L. 688: "effects with", please change to "effects on", or rather, "associations with".
31. L. 693-694: "This slow snowmelt rate allows the ground to retain more liquid water during the ablation period". I find this too much of an interpretation to belong in the conclusion. See also major comment nr 6. If you want to keep it here, please add something like "We interpret this as […]" so that readers (who sometimes only read the conclusion!) do not assume that this is an actual finding resulting from your study design.
32. Fig. S7: This seems like a copy of a previous main text figure? Also here, no scale bar for greenness and the caption still mentions place names, which aren't in the figure. Please recheck.
33. Fig. S14-S22: please change "confident levels" to "confidence levels".

---

## Author Response (AR3)

Dear Editor,

We would like to sincerely thank the editor and the referees for their constructive comments and insights. We appreciate the time you have taken to provide this detailed feedback. From this iteration of review, we can see where we have still needed to improve upon the manuscript, and in this submission, we hope to have accomplished this. We acknowledge that the referee raises important points, and we attempt in the following to account for them and hopefully convince that our efforts led to a mature manuscript.

In the following, we address the referees' comments point-by-point. We marked the comments given by the referee in red, provide our answers and comments in black, and indicate how we address the amendments in the manuscript in green.

Tiago Silva, on behalf of all co-authors.

**Report from Referee #1 submitted on the 22nd April**

Dear authors,

Thank you for all the adjustments that you have made, bringing the manuscript closer to a state where it is publishable. Rather than providing final minor comments or accepting the manuscript, I would like to draw your attention towards several persistent issues that have not been sufficiently resolved in the last iteration. I am sorry that I cannot provide more positive news, but I do not think the manuscript is in a publishable form yet.

I kindly ask you to critically evaluate my previous round of comments again. Below, I have listed just some cases in which I felt that they had not been seriously implemented. I therefor stopped reading, so I will provide examples and ask that you have the entire manuscript and previous comments checked thoroughly. Comments refer to tracked changes version.

Best of luck!

1) There are still too many examples of improperly formatted sentences. Please have the manuscript read and checked by an independent person, ideally a native speaker.

We return a once again polished version, where the native speakers in the author-team went through the entire manuscript. Indeed, we found instances of improperly formatted sentences and we apologize for the inconveniences. Thanks for pointing this out and for your patience on that matter.

Example: "permafrost thaw", "surface thaw" and "ground thaw" are still used interchangeably in the ms.

In the previous revision, minor comment 1 referred to a term in the Introduction. However, the current comment highlights terms in the Abstract and Discussion, which were retained due to referee 1's major comment 4 in the first revision round. That comment raised valid concerns about the representativeness of the reanalysis product for permafrost extent and dynamics. In response, we had added this explanation to subsection 2.1:

"SURFEX is a multi-layer surface model that computes specific schemes depending on the surface type (e.g., vegetation, soil, snow), allowing soil water phase changes and enabling runoff over frozen and unfrozen soil. This better represents areas with permafrost and ice surfaces in Greenland, which are not well described in the current version of HARMONIE-AROME."

We have now added the following sentence to subsection 2.1 in LN140: Since CARRA does not represent permafrost, lacking inter-annual classifications of its types and extent, we used the term frozen surface instead of permafrost when discussing the CARRA output.

We revisited all the instances about surface and ground thaw and modified when related to literature about permafrost, for example, LN 665: Initially, increased vegetation leads to greater carbon sequestration. However, substantial permafrost thaw potentially caused by the vegetation increase could also release carbon, offsetting the compensation from vegetation-based carbon sequestration (Glanville et al., 2012). Although north Greenland lies within the continuous permafrost zone, we avoid using the term permafrost when referring to the CARRA output. For example, LN 563: "Interestingly, trends in summer soil water content (SoilWaterJJA) and soil ice content (SoilIceJJA) are both negatively correlated to near-surface air temperatures in summer (T2mJJA). This association could result from surface thawing and subsequently increased evaporation caused by higher vapour pressure deficits in these northern areas (Fig. S22)."

Other instances, explicitly about CARRA output and southern Greenland regions, were kept with the term surface thaw. For example, LN 662: "A surplus in the surface energy budget results in surface warming and promotes surface thaw…"

Example: use of "/" still occurs in places.

Thank you for pointing this out. As the comment came without reference, we were not able to clearly identify the location of the typo(s). Using the find function make us assume that this comment refers to the following sentence in LN 601 in the former subsection 5.1.6: "Although there are no significant trends in SWE_MAX in ecoregion 2, subsurface runoff from ground thaw and meltwater from nearby snow/ice bodies likely contribute to the increase in SoilWaterMAM in the area.".

We agree that this writing style is confusing and rewrote this sentence to: While no significant trends in SWE_MAX are observed within ecoregion 2, the presence of subsurface runoff from snow and ice likely plays a critical role in maintaining or increasing SoilWaterMAM in the region.

Example: "to up almost 6%" (L. 588). Should be "up to".

Thank you for spotting the typo, we now rewrote: to up to almost 6% per decade in ecoregion 4.

Example: "Additionally, certain low-lying stripes near fjords are very narrow" (L. 772). After asking to change stripes to strips, here, apparently, strips has bene changed into stripes?

We apologize for inadvertently repeating the typo. The same sentence now reads: Additionally, certain low-lying strips near fjords are very narrow

We also carefully checked the entire manuscript for this confusing description and remain with 'strips' throughout.

2) While the discussion has been updated with thematic headings, its structure and selection of relevant literature and processes is still unclear. It does not seem to be fully aligned with and tailored to your research aims and methods and the reader gets lost in places.

Thank you very much for this comment regarding the coherence of the discussion section of our paper. We revisited the comments from the previous revision round and believe we made the necessary changes to enhance the readability of the manuscript. Additionally, we carefully considered your advice and thoroughly revisited each subsection of the discussion with your comments below in mind, adding sentences where applicable to further improve the clarity. We

appreciate your guidance and input. Thanks for guiding us so carefully through this process! Below, we add some clear examples after the specific points raised:

You could improve this by, for instance:

* checking whether each paragraph has a clear introductory sentence and concluding sentence, and a clearly delineated focus.

We added some introductory sentences in the Discussion. For instance, in LN 555: Upon investigating the bio-climatic factors driving greenness changes in the northern ecoregions, we found that areas related to greenness expansion appear to be associated with a rise in SoilWaterMAM along with declines in both spring soil ice content trends (SoilIceMAM) and maximum snow depth (SWE_MAX). or in LN 570: To better understand the reduction in SoilIce, which highly correlates with greening, we investigated the relationships among changes in SWE_MAX, MeltRate, SoilWaterMAM, SoilIceMAM, and greenness, and examined the levels of SoilIceDJF. We found… .

We also added some concluding sentences in the Discussion. For example, LN 576: These changes create a more favourable setting for vegetation growth, enabling some plants to expand or establish in areas where frozen conditions previously limited their presence (e.g., Shijin and Xiaoqing 2023; Yang et al. 2024). or in LN 598 These drying processes collectively force and constrain vegetation expansion toward areas closer to water bodies, where soil moisture levels can better support vegetation (e.g., Chen et al. 2023; Gamm et al. 2018).

As we showed with these examples, we carefully adapted all sections in the Discussion following this structure.

* reducing the use of abbreviations and variable names (such as soilwaterMAM) while discussing environmental processes

We agree that the use of abbreviations comes on cost of readability, despite helping conciseness. In order to find a good balance between clarity and conciseness, we stick to abbreviations after having carefully introduced their full name in the Discussion.

* mentioning the direction of associations (positive or negative) rather than just their existence.

After having gone through the manuscript, we found several places that allowed for clearer attribution of the changes: For instance, in LN 567: Greening in ecoregion 1 demonstrates a stronger positive correlation with SoilWaterMAM patterns, similar to the remaining southwestern ecoregions. or in LN 563: Interestingly, trends in summer soil water content (SoilWaterJJA) and soil ice content (SoilIceJJA) are both negatively correlated to near-surface air temperatures in summer (T2mJJA).

* making sure that it is clear for every sentence where this information originates from (many statements lack reference to a figure, table, supplement or literature).

We added further figure references in several places. For example, LN 589: Southern ecoregions with significant decreases in SWE_MAX show early SWE_MAX day of the year (DOY, Fig. S18) that leads to early Onset. (…) Despite an increase in fresh snow accumulation and a reduction in drought days during the spring, the observed declining trend in SWE_MAX for West Greenland is linked to a decrease in winter snowfall (Fig. S16)… or LN633: Grimes et al. (2024) investigated land cover changes across Greenland by using Landsat images from the late 1980s to the late

2010s, and found spatial patterns of vegetation change similar to our findings (Figure 6c). (…) Specifically, in ecoregions 2 and 5, greenness expansion is not only occurring toward the inland regions, but also upward (Fig. 6b).

* provide context on the scope of your discussion; some paragraphs only discuss climatic changes but apparently little relation to greening or to your study aims (e.g. 5.1.6 and 5.1.7; it is unclear what the reader should take away from these paragraphs).
We reviewed all the paragraphs in the Discussion section and ensured that they address changes in greenness, greenness extent, and greenness distribution, as well as their relationship with weather patterns and bio-climatic indicators. We acknowledge that these mentioned subsubsections were not appropriately framed. In order to fulfil the requirement for better consideration of greening in all discussion subchapters, we also incorporated subsubsection 5.1.6 into subsubsection 5.1.4 and combined subsubsection 5.1.7 with subsubsection 5.1.3.

**Report from Referee #1 submitted on the 20[th] February**

Dear authors, dear editor,

Thank you for your response to my earlier comments, and for all the edits to the manuscript. The authors have made substantial improvements in avoiding speculation outside of the discussion paragraph, providing essential details on the processing of data to ensure transparency and reproducibility, and have even added novel analyses and supplements. There is now a clearer explanation of the assumed mechanisms behind the observed patterns. Improvements have been made in phrasing more clearly which statements are interpretations, and which are actual findings of the study.

I see the scientific value in this large-scale and holistic evaluation of climate – vegetation growth relations that the authors provide for Kalaallit Nunaat / Greenland. I think Biogeosciences is an excellent platform for such a large-scale and interdisciplinary endeavour.

Thanks for your kind lines and we appreciate that you can see the value of our study. We are looking forward to working on the constructive comments to present a mature study.

I do however still have several concerns and in my vision the manuscript is not publishable in its current form. I hope this next revision round will be helpful to get this work published and I would like to kindly ask that the authors are a bit more thorough this time. **All line numbers refer to the text without tracked changes.**

Please keep working on this! Wishing you lots of success!

Major to moderate concerns

1. I agreed with the second reviewer that the aims reported in the introduction, the methods, and the conclusions mentioned in the abstract and conclusion, were not yet fully aligned. While improvements have been made, several issues still remain. See minor comments. I particularly have a concern about the second aim: "We examine the combined effects of bio-climatic indicators ranging from sub-surface factors (such as soil water availability) to above-surface

factors (such as the thermal growing season, heat stress, and frost) with summer spectral greenness." (L. 114-116). I would assume that you mean "combined effects [...] ON summer spectral greenness" rather than "[...] WITH spectral greenness"? More appropriately, I would write "association with" rather than speak of "effects on" in this correlative study. In my opinion, this would be more aligned with running a PCA that includes greenness. If the aim is to study combined effects of bio-climatic indicators on greenness itself, then a multivariate method that models greenness as a response variable (such as pls regression) would be more appropriate.

Thank you for your remark. We apologize for the caused misunderstanding. We agree that rewriting the aim as you suggest is an important clarification. Indeed, we examine the associations among the bio-climatic indicators with summer greenness. i.e. how these factors co-interact with greenness.

By adjusting LN116 in the following manner in the Introduction as: We examine the associations among bio-climatic indicators ranging from subsurface factors (such as soil water availability) to above-surface factors (such as the thermal growing season, heat stress, and frost) with summer spectral greenness. We also extend our study of bio-climatic changes beyond the summer by examining indicators from the preceding winter and spring and assessing their combined interactions with summer spectral greenness.

In the *Interconnectedness among bio-climatic indicators* in LN405: Note that these physical features are constant through time and were not considered when investigating the combined associations among bio-climatic indicators with greenness in the PCA.

We agree that this subtle change has clarified the focus of our work.

After revising the manuscript, we did not find any other instance where the relationship between bio-climatic indicators and greenness could lead to similar misunderstandings.

2. The other reviewer and I both indicated that improvements were necessary throughout the ms in terms of sentence structure, grammar and language errors. Many improvements have been made, but there are still too many examples of grammatical errors and typo's, particularly in the newly written text. I have made many suggestions in the previous round of editing, but ultimately I believe reviewers should focus on the scientific aspects of a ms, and not provide free language editing services. Well-structured and unambiguous sentences are necessary to convey complex scientific matter to the broad and interdisciplinary readership of a journal like biogeosciences. The authors will have to do thorough language checks, ideally by a native speaker, or potentially use AI based language improvement tools. This also goes for any newly edited or added text.

Thank you for your remark. We made yet another and even more thorough grammar and sentence structure check in the revised version and we hope to have improved the readability of our study. We apologize that the initial submissions did not meet the expectations.

3. The authors have added extended definitions of the different terms they use for various manifestations of greenness. This does not mitigate the fact that there is an excessive amount

of different terms in use throughout the ms, and that in some cases two or more different terms are used for the same thing. Please add specific and singular definitions under 3.1. I would assume that you would need only about 3 terms ("greenness" -> seasonally averaged NDVI, please add time range in months. "seasonal duration of greenness" -> amount of months within the season that NDVI was > 0.15. and perhaps another term along the lines of "green pixel" to indicate pixels with NDVI > 0.15...?). Perhaps use either "greenness" or "spectral greenness" but not both. These are suggestions of course; the bottom line is that all terms need to be defined clearly, and that there needs to be a stricter limit to the amount of different terms in use.

Example: L. 213-2014 "Finally, we calculated a seasonally averaged NDVI, hereafter referred to as spectral greenness and interchangeably as green vegetation" -> it is unnecessarily confusing to use two very different terms for the same thing. Another example: the terms "extent of green vegetation", "green vegetation extent" and "spectral vegetation extent" are all used interchangeably. Please pick one. All the different terms in use really mean and imply different things from ecological perspectives and remote sensing perspectives and correct terminology matters in this context (if necessary, consult review by isla myers-smith, 2020, already cited). Please critically evaluate the whole ms including figures labels, tables and captions for greening terminology, simplify and realign thoroughly.

Thank you for the comment and suggestion that really improves comprehensibility. We revisited every term related to greenness in the manuscript and sticked to three terms, that are now defined in the subsection *Spectral greenness*. We will keep using greenness as defined by Isla Myers-Smith et al. (2020), greenness extent and greenness distribution (LN227).

Since we perform calculations for temporal and spatial changes, in the revised version of the manuscript we indicated that: Pixels exhibiting a monthly NDVI of 0.15 or greater are indicative of monthly greenness. The area derived from this monthly greenness is defined as the greenness extent. Additionally, we calculated the summer average greenness (see subsection 3.2 for the season definition), which we will refer to greenness hereafter. We also assessed spatio-temporal changes in the greenness extent between the periods of 2008--2023 and 1991--2007. We described these comparisons as changes in the greenness distribution, where an increase in greenness distribution is characterized as an expansion and a decrease as shrinkage. In addition, we analysed temporal changes in greenness (more details about trend analysis provided in subsection 3.4), wherein positive trends denote as greening, and negative trends denote a reduction in greenness.

4. The authors replied that they disagree with my request to add scale bars for "greenness" in figures, since it refers to a 32year average of the mean JJA NDVI. If you judge that a 32 year average is not informative, then I would wonder why you choose to depict it at all. Please either remove the figures completely, if you judge that the greenness patterns are not useful for interpretation, or add numbers. As a reader, I want to know how to place this information in the context of panarctic ndvi values and trends, data from different sensors, saturation values and the thresholds that you use for defining "greenness". It is also basic cartography and potentially a journal requirement that continuous scale bars include numerical labels.

Thank you for the comment, which makes us realise that we were not clear enough in our earlier answer. We wrote in point 15 of the previous point-by-point answer that we "kept the delineation of the ecoregions and the greenness evolution during summer in the *Methods*, as they will support the readers to understand the geography of the ecoregions and to recognise the greenness dynamics across Greenland from June to August as well as what entails the summer averaged greenness."

In the revised version, we add the numbers to the scale bar correspondents to the 32-year monthly averaged spectral greenness.

[Figure]

Additionally, to avoid further misunderstandings and help with the interpretation of averaged spectral greenness in Figure 1, we now write in subsection 3.3 that: In Figure 1 we also show the 32-year monthly averaged greenness for summer months. As mentioned in subsection 3.1, the typical NDVI analysis that consist in averaging either the entire NDVI range or selecting the maximum NDVI are more prone to artifacts. Therefore, the 32-year monthly averaged greenness here shown is not necessarily based on 32 values in every pixel. This is reflected by the monthly averaged greenness over 32 years to be lower than 0.15 in many regions. While the 32-year monthly averaged greenness spatial variability can be assessed with Figure 1, direct quantification of greenness saturation should be taken with care given the interannual variability in greenness. Maps with the correlation coefficients between greenness and North Atlantic Oscillation (NAO) index and the Greenland Blocking Index (GBI) between 1991 and 2023 are shown in Figure S7.

5. The authors replied to my earlier request to implement thematic discussion points in the ms by explaining (only in the response) what the paragraphs are about. Please make sure that this rationale is actually visible within the ms itself. You could for example add sub-headers for several main lines of interpretation so that the reader knows what is going to be discussed, and what is relevant for them to read, instead of seeing one large page of text. Many readers will not read the discussion from beginning to end, but rather focus on specific aspects that are of interest to them. Please implement a more logical flow to the information presented in the discussion. (And a general request: make sure all comments are met by changes in the actual ms itself and cite the line nrs. Or explain and argue why no changes were made).

Thank you for the comment which supports the clarity of our work. We revisited the discussion in subsection *Key findings and interpretation in the context of the current literature* and add the subsubsection titles as indicated in the point 32 of the previous point-by-point document: 5.1.1. Changes in greenness extent; 5.1.2. PCA performance and basis for interpretation; 5.1.3.

Also, we added line numbers at most of our replies and indeed made sure that all information present in the response to referees also ended up in the revised manuscript.

6. Paragraph 5.1, general: I now better understand the different dynamics of earlier season, slower melt of shallow snowpacks, and later season melt under warmer conditions in deep snow. Thank you for elaborating. This helped a lot. Throughout paragraph 5.1, I could not help but wonder to what extent the observed increases in soil water content in spring (MAM), and association thereof with greenness, are not simply a result of the soil's increasingly unfrozen state (due to shallow snow that melts early, as well as warming earlier in the season). Would this not also result in an increasingly unfrozen state of Greenlandic soils in spring, corresponding to higher water content and lower ice content in spring? Or can you provide additional argumentation, that clearly shows that increased spring soil water content is indeed the result of slower melt dynamics of shallower snow packs? This needs more argumentation.

Thank you for your reflection and for pointing to a missing explanation. We acknowledge that the reduction of ice water content in spring is not elaborated in the manuscript and now added to the respective subsubsection *Changes in the Northern ecoregions* in LN555:

After investigating the relationships among changes in SWE_MAX, MeltRate, SoilWaterMAM, SoilIceMAM, and greenness, and examining the levels of SoilIceDJF, we find no significant trends in SoilIceDJF. This suggests that to a certain extent, the proportion of frozen ground has been restored during the cold season. Changes in SoilWaterMAM are moderately proportional to changes in SoilIceMAM, indicating that the increase in the liquid water content in the soil during spring primarily originates from snowmelt. Subsequently, the presence of liquid water in soil with higher thermal conductivity, coupled with shallow snow depths (and eventually snow-free conditions), allows for a more efficient exchange of energy between the surface and the atmosphere, consequently leading to ground thawing.

Minor comments

1. L. 105: here "permafrost thaw" was replaced with "ground thaw", for reasons that aren't clear from the response. I find this new term unnecessarily vague, and I would recommend to change it to "permafrost thaw" or "ground ice degradation", depending on whether the authors are referring to seasonally frozen water resources in the active layer, or availability of new moisture resources from degrading, ice-rich permafrost.

We apologize for the unnecessary change which came from an attempt to generalize but indeed, it is clearer to stick to permafrost thaw which we do in the newly revised version.

2. L. 119 "greenness distribution"; seasonal or spatial distribution? Such nuances are important, especially given the many different terms and definitions of greenness that are used throughout the manuscript.

Thank you for the remark. This point is answered in major point 3.

3. Line 133-136: Thank you for adding this critical information on how soil water and ice content are derived. Can you also add information on the accuracy of these subsurface components of the re-analysis products? Otherwise it remains a bit of a black box whether the reported associations between snow, soil water and greenness variables accurately reflect real-world processes, or whether they simply result from the way that the different sub-models are defined (with risk of circular reasoning).

We agree, even though there is not that much literature available showing the performance of CARRA for Greenland from validation studies. However, there is the study of van der Schot et al. (2024) validating snow depth/SWE of CARRA against independent observations in Greenland and showed a promising performance of CARRA simulating snow depth/SWE. This is quite an extensive validation, as snow depth and SWE are the result of several other atmospheric variables including air temperature and precipitation.

In addition to the snow depth/SWE validation from van der Schot et al. (2024), we also add a new paragraph in the subsection *Copernicus Arctic Reanalysis* to inform the reader about a set of model schemes and parameterizations implemented in SURFEX7.2. In LN139, we wrote: The snow and frozen soil parameterizations from the ISBA (Interactions between Soil, Biosphere, and Atmosphere) scheme, as described by Noilhan and Planton (1989) and implemented in the SURFEX7.2 (Masson et al., 2013), have been tested in model intercomparison campaigns across northern Europe (e.g., Luo et al. 2002; Slater et al. 2000), high latitudes (Decharme and Douville 2006), and the Alpine regions (e.g., Decharme et al. 2016).

The physical parameterizations within the ISBA have seen progressive developments over the past decades, particularly in its snowpack scheme, Crocus, which accounts for various snowpack features — such as thickness, temperature, density, liquid water content, and grain types — and incorporates physio-geographical attributes like the surface slope. Crocus has been consistently coupled with global reanalysis like ERA5 (e.g., Ramos Buarque et al., 2025) and other atmospheric models (e.g., Luijting et al. 2018). When integrated with the atmospheric model AROME, Crocus accurately reproduced the evolution of the snow surface temperature over Dome C (Antarctica) during an 11-day period (Brun et al., 2011), and it has effectively represented snowpack features in the French Alps (Vionnet et al., 2012) for more than a decade.

Regarding surface and subsurface parameterizations, ISBA scheme explicitly calculates the actual ice and water content in the soil to determine the heat capacity and thermal conductivity of the ground. The ground thermal conductivity depends on the surface and soil heat fluxes, which in turn are dependent on the soil scheme. For soil schemes with vegetation, ISBA allows roots and organic matter to favour the development of macropores which can lead to enhanced water movement near the soil surface. To our knowledge, accuracy data for SURFEX schemes when coupled with AROME-HARMONIE are not yet available.

4. L. 161: "likely signifying areas with no vegetation presence". I think this part of the sentence should be removed; vegetation can be present under clouds, cloud shadows or seasonal

inundation. Or even under snow (for instance, bryophytes and evergreens under snow may not be photosynthetically active at that moment, but they are still there).

Thanks, we correct the sentence to: Negative NDVI values are typically associated with water, clouds, or snow, with no spectrally visible vegetation.

5. Table 1) one of the indicators in "SWEmax", but it just gives the snow water equivalent at any give moment as far as I can judge from the definitions. Don't you mean "SWE" here, as also referred to later in the ms? If you mean maximum SWE, add to the definition that this is a seasonal maximum?

Thank you for the remark. The definition of SWE_MAX in Table 1 is now written as annual maximum mass of liquid water from melting the snow per unit area. This is now in agreement with the text and with the minor point 6.

6. In a more general sense, it is rather inconsistent among Table 1 and the text under 3.2 whether and how seasonal integration/averaging is mentioned. This is explicitly mentioned in the table for variables like T and greenness, but not VPD (for which it is only mentioned in the text). Please find a way to make this clearer. I recommend mentioning your definition of what "seasons" are before the table, then listing descriptions in the table and adding a column to the table that explicitly states whether it is an annual variable (starting when? Previous autumn or previous winter?) or a seasonal variable, and which seasons are considered or not (e.g. some are only calculated for spring/summer and others only for winter, which makes sense).

Thank you for your comment. We added to the description of the Table 1 whether a certain bio-climatic indicator corresponds to annual or seasonal statistics (count, average and sum). We also moved the paragraph explained the season definition before Table 1.

7. Line 285-286: "The combination of this region's complex topography with frequent cloud cover resulted in its exclusion from the analysis". Many outcomes for this ecoregion are still reported (Fig. 2, Fig. 4 and associated text, discussion). Please explain why such findings are still presented despite the limitations mentioned here.

Our previous statement refers to greenness in Southeast Greenland specifically, where we refrain from interpretation throughout the manuscript. Southeast Greenland is not defined as an ecoregion (please see Fig. 1) and therefore not represented in the above-mentioned figures. Only the output from CARRA is available and shown in Figure 5 along the Southeast coast.

8. Line 303-310: This is background information on the NAO and GBI, not a method. Please integrate into the introduction of the role of NAO/GBI in the introduction and use the methods section only to describe what you did yourself.

Thanks, we deleted this background information on the NAO and GBI from the *Methods* since climate oscillations are already well described in the Introduction.

9. L. 313 – 315; again use of the phrase "influence with", which is grammatically incorrect and methodologically confusing. PCA cannot demonstrate influences, only associations. It also

seems incorrect to me to speak of an influence when greenness is treated as one of the variables going into the PCA, rather than a response variable.

Thank you for the remark. We replaced the noun "influence" by "interaction" not only in the mentioned sentence, but in the other instances where "influence" is used in the same context. The referred part is now changed to: Principal Component Analysis (PCA, Pearson 1901; Lorenz 1956), often used on remotely sensed and environmental data (e.g., Mills et al. 2013; Yan and Tinker 2006), was employed to investigate the combined interactions among bio-climatic indicators with summer spectral greenness.

10. L. 331: "This will diminish noise"; please avoid different tenses in the methods. Everything else is in past tense, so please rewrite to past tense and check tenses throughout the manuscript for consistency within paragraph.

Thanks for the remark. We revisited the *Methods* and corrected the verb tense to past tense.

11. L. 354-355: ", due to the typically shallow snow cover". This is an example of an interpretation in the results that is not backed by a figure or a statistic. Please avoid such interpretation while reporting the outcomes of your analyses, or back them up with evidence. Please recheck the results section for such interpretations.

Thanks for the comment. The climatology of certain environmental variables is early described upon the delineation of the ecoregions in the subsection *Ecoregions*. We consider that this information was already backed up. Therefore, we add: (see subsection 3.3)

12. L. 359-360: "Correlations between green vegetation extent and summer GBI are investigated for three periods: AVHRR (1991-2013), VIIRS (2014-2023) and the full period (1991-2023), and are shown in Table S1.": example of methods, mentioned under results. Please move to methods.

Thank you for your remark. We consider this piece of information more suitable for the current *Results* section, as the statistical output is used for interpretation of the results. However, we added to the *Statistical Methods* in LN355 that: We performed correlation and trend analysis in three periods: AVHRR (1991--2013), VIIRS (2014--2023) and the full period (1991--2023) between greenness and climate oscillations to assess their statistical strength and tendency as dependent on the sensor period.

13. L 360 -370: many grammatical issues (tenses, plural/singular). Please carefully check whole ms.

Thank you for the remark. We reviewed the manuscript to correct grammar issues.

14. L. 375-395: some variables are removed from PCA analysis based on their degree of association with greenness. This is a methodological choice that needs to be described and backed up in the methods, based on a priori informed criteria. No threshold value (in correlation or p value) is mentioned at all, making the choices seem arbitrary (even though they are probably not). Please describe all choices and criteria in the methods (as I requested in the previous

revision round), so that here you can stick to reporting the outcomes, and you only need to mention which variables make the benchmark for inclusion.

We are sorry that the selection of indicators was not clear enough. The selection is covered in both *Statistical Methods* and *Interconnectedness among bio-climatic indicators*.

In the subsection *Statistical Methods* we write: The calculated correlations are displayed in a correlation matrix, and bio-climatic indicators with similar correlations are sorted with hierarchical clustering. This helped to visually discern bio-climatic indicators with comparable statistical relationships and supported on the empirical reduction of indicators accounting for the relevant physical and ecological processes on the tundra ecosystems, later used as part of the PCA. This aimed to diminish "noise", redundancy and ultimately boost the clarity of interactions across atmosphere-biosphere-cryosphere.

In subsection *Interconnectedness among bio-climatic indicators* we write:

SoilIce is largely negatively correlated with the volume of water in the soil (SoilWater). Therefore, we decided to arbitrarily use SoilIce in winter (SoilIceDJF) and summer (SoilIceJJA) and SoilWater in spring (SoilWaterMAM) and autumn (SoilWaterSON) in the further analysis. Additionally, SnowDays and DegreeDays are not used since both are highly explained by GrowDays. While DegreeDays accumulate T2m during GrowDays, SnowDays complement FrostDays and GrowDays -- together, they represent snow-free occurrences when daily T2m is negative and higher than 1°C, respectively. Strong correlations between Rain and RainRatio are found in spring and autumn, but not in summer. Consequently, we will retain both Rain and RainRatio variables exclusively for the summer. Finally, MeltRate is removed as it is physically explained by the snowpack depth.

In order to improve clarity, we added to the *Statistical Methods* in LN350*:* Certain bioclimatic indicators exhibited high correlations among them, primarily due to physical reasons. Other bioclimatic indicators corresponded to complementary quantities. Consequently, the selection of bioclimatic indicators for the PCA was made on an arbitrary basis further detailed in subsection 4.1.

15. L. 396: please rewrite all instances of "influence with" and clarify in the aims whether this analysis is meant to show influences (= impossible with PCA) or associations.

Thank you, we have implemented this in minor point 8.

16. L. 427 – 434: Here new analyses (methods) are introduced in the results. Please move to methods and strictly report outcomes in results.

Thank you for the remark. We have moved this paragraph to the methods.

17. L. 440: "strips" instead of "stripes"? Strips of what?

Thank you for the remark. We corrected the sentence to: Along the narrow ice-free strips of land in the Southeast, there is a modest increase of GrowDays (approx. 5 days per decade), at several elevations around Tasiilaq.

18. L. 471-473: another example of methodological information in the results. This is already described, in different wording, in Line 210-ish. Please integrate this information there and report only the outcomes here. Adding just a little sentence as a reminder of what you did as a guideline for the reader should be ok, but restating the whole processing approach is excessive.

Thank you for the remark. The paragraph is now simplified and found in the revised version: In order to assess which regions became greener due to greenness expansion, we detected whether a pixel met the summer greenness criterion annually from 1991 to 2023. A detailed explanation on how the study period was split into two to investigate changes in greenness distribution is found in sub-section 3.1.

19. L. 476: as I mentioned in the previous round of revisions, calling this "expansion of vegetation" is very misleading to all readers with an ecological background, since vegetation expansion is more or less exclusively used for spatial expansion. If this is about temporal expansion of photosynthetic activity, then please name it something else, like "extension of the growing season".

Thank you for the remark. Please refer to major point 3 above to see how we have addressed this. 3.

20. Table 2: the methods state that this is to be referred to as "changes in greenness distribution" later in the ms, but this does not seem to be implemented, since the results only mention greening "expansion" and "shrinkage", interchangeable with "vegetation" expansion/shrinkage. Or sometimes "reduction". This is an example of the major comment on greening terminology. Please use uniform terminology.

Table 2 caption is now written: Percentage of expansion and shrinkage of greenness distribution, and ratio (fraction of expanded by shrank area) between 2008 and 2023 with respect to the period 1991–2007 in % of the total ecoregion area. This also considers the feedback from major point 3 and minor point 19.

21. Discussion, general: There are many grammatical errors in the newly added text. The discussion really needs to be evaluated for language.

Thank you for the remark. We have carefully reviewed the manuscript.

22. L. 499-505: This paragraph needs backing from literature (on the blocking events) and reference to figures and tables.

Thank you for the comment. The paragraph written in the revised version is:

Greenness extent has increased over time across Greenland, with an increase rate of 2% per decade in ecoregion 1 to up almost 6% per decade in ecoregion 4 (Figure 2). When comparing the recent half of the time-series (2008--2023) to the earlier half (1991--2007), the distribution of greenness has also changed. In ecoregions 3 and 5, the distributions of greenness expanded to nearly double and eight times the size of the areas that shrank, respectively (Table 2). Within the time series, maximum greenness extent was observed in 2019, aligning with the end of a

period of frequent, long-lasting and intense summer atmospheric blocking conditions in the vicinity of Greenland, conditions which promoted advection of relatively warm and humid air from the North Atlantic along West Greenland (Silva et al., 2022).

23. L. 518-529: Please tone down your statement that observed dynamics are a consequence of permafrost thaw (this is an interpretation, and cannot be described as a causal influence without further backing).

Thank you for recognising our overinterpretation. We rewrote the paragraph as follows:

Our study found that areas related to expansion in the northern ecoregions appear to be associated with a rise in SoilWaterMAM along with declines in both spring soil ice content trends (SoilIceMAM) and maximum snow depth (SWE_MAX). In Northwest Greenland, including ecoregion 1, regional exceptions of widespread increases in SWE_MAX with regional delays in the onset of the thermal growing season (Onset) are found along coastal areas and are not related to greening. Conversely, areas related to greening are statistically linked with rising SoilWaterMAM, accompanied by higher spring temperatures (T2mMAM) and earlier Onset. Despite regional trends on higher summer rainfall amounts (RainJJA, Niwano et al. 2021) in northern Greenland, we did not find a clear link between greening and changes in RainJJA.

Interestingly, trends in summer soil water content (SoilWaterJJA) and soil ice content (SoilIceJJA) are both negatively related to near-surface air temperatures in summer (T2mJJA). This could result as a consequence of surface thawing and subsequently increased evaporation caused by higher vapour pressure deficits in these northern areas (Fig. S22). The greening of the recently emerged vegetated areas in the northern ecoregions respond to different seasonal soil water contents. Greening in ecoregion 1 correlates best with SoilWaterMAM patterns, similar to the remaining southwestern ecoregions. In contrast, ecoregion 5 is more closely connected with SoilWaterJJA, likely due to a later onset of the GrowDays.

24. L. 531-533: Here you mention "spring (winter)" and "winter (spring)". Please elaborate on what you mean by this. Do you mean something like "in winter, and to a lesser extent in spring"?

Our apologies for the confusing sentence. We write in the revised version: Despite an increase in fresh snow accumulation and a reduction in drought days during the spring, the observed declining trend in SWE_MAX for West Greenland is linked to a decrease in winter snowfall; conversely, for East Greenland, it is attributed to reduced spring snowfall.

25. L. 595: Why shrubs, specifically, and not other plant functional groups? Please explain in the ms.

Therefore, we may argue that the spectral greening is generally related to vascular green vegetation expansion throughout the past three decades, as early proposed by Sturm et al. (2001).

26. L. 628: "longer roots". Please write "deeper roots". Roots can be long without extending deeply. Perhaps write "deeply rooted species" (not only graminoids) and cite papers on actual

root development under warming rather than this model study (van der kolk) based on assumed vegetation growth parameters rather than actual observations. Suggestion: https://besjournals.onlinelibrary.wiley.com/doi/abs/10.1111/1365-2745.12718

Thanks for the comment. We write "deeply" instead of "longer" rooted systems and use the recommended reference.

27. L. 630: "This ecological shifts might will also affect".

Thank you for the remark. We have corrected the sentence to: These ecological shifts might also affect...

28. L 660-661: "Such periods have favoured exceptional vegetation growth across western ecoregions as shown in our results". In your previous response letter you stated that cloudiness prevented you from actually inferring whether warmer and more humid conditions during such events are indeed associated with greening. So I am a bit confused now. Please refer to the specific results that support this claim.

Thank you for the remark. We corrected the sentence to: Such periods have favoured exceptional vegetation growth across western ecoregions as shown in our results. However, surface reflectance retrievals may have been impacted by cloudiness, partly hindering the spatio-temporal changes in spectral greenness.

29. L. 682-685: These are implications, not recommendations or limitations.

Thank you for the remark. We moved this paragraph to *Significance and implications*.

30. L. 688: "effects with", please change to "effects on", or rather, "associations with".

Thank you. This instance was considered in minor point 9.

31. L. 693-694: "This slow snowmelt rate allows the ground to retain more liquid water during the ablation period". I find this too much of an interpretation to belong in the conclusion. See also major comment nr 6. If you want to keep it here, please add something like "We interpret this as [...]" so that readers (who sometimes only read the conclusion!) do not assume that this is an actual finding resulting from your study design.

Thank you for the comment. We start this statement as suggested by the referee.

32. Fig. S7: This seems like a copy of a previous main text figure? Also here, no scale bar for greenness and the caption still mentions place names, which aren't in the figure. Please recheck.

Yes, Figure 1 from the initial pre-print was moved to the supplementary material, where the correlations maps of summer greenness and climate oscillations are shown. We changed the colour bar as mentioned above in point 4 and kept the sentence in the caption with the placenames by showing them again.

[Figure]

33. Fig. S14-S22: please change "confident levels" to "confidence levels".

Thank you for the remark. The word "confident" was now changed to "confidence" in the supplementary figures with trend maps.

---

## Author Response (AR4)

Dear Editor,

We once again would like to sincerely thank the editor and the referees for their constructive comments and insights provided throughout all the revision rounds. We deeply appreciate the time everyone has taken to provide feedback. Based on the comments from the editor, we have further improved our manuscript.

In the following, we address the editor's comments point-by-point. We marked the comments given by the editor in red, provide our answers and comments in black, and indicate how we address the amendments in the manuscript in green.

Tiago Silva, on behalf of all co-authors.

The manuscript is much improved, and thank you for working on this, but there are still a few remaining points that could make the analysis stronger, including some assumptions which I feel need more justification or explanation. Please consider the following and I look forward to the revised manuscript.

The terrestrial ecosystem (ice-free area) in Greenland has -> Terrestrial ecosystems (ice-free areas) in Greenland have

We have updated according to the suggestion.

12: 'responded highly' -> responded strongly

We have updated according to the suggestion.

28: vegetation has expanded…is this vegetation cover or leaf area index?

Since our results are based on spatio-temporal changes in spectral greenness (NDVI≥0.15), the vegetation expansion is a result of newly emerging spectrally green areas. To clarify this point in the abstract, we now write: From spatio-temporal increases in spectral vegetation, we infer vegetation expansion northward and towards the interior of Greenland.

First paragraph of the introduction: I'm not asking for a change but just a general piece of writing advice. Almost never start a sentence with an author or a paper because it makes the subject of the sentence the author rather than the dynamic at hand (here Arctic greening).

Thank you very much for your very valuable writing advice. We will follow the recommended writing approach in future manuscripts.

65: 'striking'' is subjective. Best to just avoid adjectives whenever possible.

The subjective adjective has been removed. We carefully read the manuscript again and did not find any other instance with qualitative adjectives.

87: yes in general but snowpack thermal characteristics in addition to depth, like density, are also important for melt.

We completely agree. In the seasonal snowpacks, both depth and density are indeed important for generating and sustaining melt. However, while higher snow density increases the energy required to melt snow, it also increases thermal conductivity, with more efficient vertical heat transfer, accelerating melt. In order to complete the description of the thermal characteristics of the snowpack, we have added the following sentence to the Introduction:

"A relevant characteristic of the snowpack is that deep snow requires more energy than shallow snowpacks to equalise the cold content and liquid water holding capacity, an equalisation needed to subsequently initiate and sustain melt (Colbeck 1976; Musselman et al. 2017)."
However, if the snow is dense, its higher thermal conductivity can enhance internal heat transfer, partially offsetting this energy requirement. "As a result, deep snow often persists for extended periods, potentially delaying the start of the growing season and hindering plant growth (Schmidt et al., 2015)."

95: will also probably result in groundwater recharge. Honestly the snowpack section can be cut or shortened considerably. Nit necessary but the introduction is a bit long.
The detailed explanation regarding the thermal characteristics of snowpack and its relationship to soil water is based on valid feedback from the first revision round (e.g., Major Point 3, Minor Point 4, and Minor Point 33 from referee 1). Given the interdisciplinary nature of the manuscript, readers who are not familiar with snowpack characteristics may find it counterintuitive that more shallow snowpacks melt more slowly. Therefore, we previously followed the advice of Referee 1 and expanded the Introduction, and we suggest maintaining it as it is.

2.2: I'm left a bit confused about how the long NDVI records were harmonized.
We assume that 2.2 refers to the subsection 2.2 where we solely describe the remotely sensed products from the NOAA Climate Data Record for Normalized Difference Vegetation. The description on how we attempt to minimize differences due to sensor change during the study period is found in subsection 3.1 Spectral Greenness. We address this comment more in detail in the next point.

235: how does more observations lower NDVI? Is this because of snow/ice/cloud pixels? It's still not entirely clear to me how data were harmonized to account for differences in instrument and processing.
We rewrote and expanded the paragraph that was previously unclear to:
To minimize the influence of temporal sampling artifacts at high latitudes, we began by calculating monthly integrated NDVI, as these estimates are less likely to be affected than metrics based on maximum NDVI (e.g., Myers-Smith et al. 2020). Our focus is on green vegetation, so we only considered daily NDVI pixel values greater than or equal to 0.15. We then divided the monthly integrated NDVI by the total number of observations available for that month (n) to obtain the monthly averaged greenness, analogous to the calculation of the arithmetic mean.
However, as shown in Figures S1 and S3, the AVHRR NDVI dataset, despite having more observations, exhibits less spatio-temporal variability compared to the VIIRS NDVI. This discrepancy is likely due to the less strict quality control regarding environmental conditions (i.e., snow cover, clouds, and shadows) in the AVHRR algorithm, which may have led to inaccuracies in NDVI calculations, as considered in subsection 2.2. As a result, calculating the arithmetic monthly mean for the AVHRR NDVI record would produce lower monthly greenness. To address the potential misrepresentation of the environmental conditions during the AVHRR period, we chose to use a reduced n based on the monthly minimum, average, and maximum number of observations from the VIIRS NDVI record to calculate monthly greenness. From 2014 to 2023, we identified these three statistics for each month. Then, we generated a consistent variability range from 1991 to 2013 to recalculate monthly greenness, ensuring a similar number of observations as those from 2014 to 2023. Figure 2 illustrates the resulting variability range of these three quantities in relation to the calculated monthly greenness extent. This approach

assumes that the environmental conditions from 1991 to 2013 are comparable to those from 2014 to 2023. Figures S2 to S5 present the average number of monthly observations and the associated standard deviation for both the AVHRR and VIIRS periods, both before and after adjusting n.

In the Results and in Table S1, we statistically compared the calculated monthly greenness with an independent variable, such as the Greenland Blocking Index. There, we describe how the prevailing weather patterns relate to changes in spectral greenness for three periods: AVHRR (1991-2013), VIIRS (2014-2023), and the full period (1991-2023). Positive and significant correlation coefficients ranging from 0.5 to 0.8 were found between ecoregions 1 and 4, generally with higher correlations for the VIIRS period than for the AVHRR period.
Ultimately, we stated in subsection 5.3, Study Limitations and Future Research Directions, that "The NDVI datasets used in this study come from two NOAA satellite products, each employing a different sensor type. The absence of overlapping temporal datasets limited our uncertainty assessment, and the potential for mismatches between the datasets cannot be disregarded. This lack of a common calibration period raises concerns about the reliability of long-term time-integrated NDVI analysis."

243: a bit confused by the 0.15 NDVI threshold, especially in the artic where soil reflectances or intermittent standing water (depending on ecosystem) can result in situations where vegetation is present but measured NDVI is quite low. Is there a reference for the 0.15 value?
The NDVI threshold selected for analysing spectral greenness is based on the literature referenced in the first sentence of subsection 3.1 Spectral Greenness. We have now rewritten and expanded that first sentence as: Arctic regions are characterized by sparse vegetation, which often results in notably low NDVI values, sometimes as low as 0.15, as observed by Liu et al. (2024) at the start of the growing season on Disko Island and by Gandhi et al. (2015) in scrublands. In contrast, areas with dense shrubs in tundra regions typically exhibit NDVI values above 0.5 (e.g., Walker et al. 2005), with signal saturation occurring around 0.7 (e.g., Myers-Smith et al. 2020).
In subsection 2.2, we provide a brief overview of the typical interpretation of the NDVI range. To highlight the NDVI mentioned by the editor, we now include the sentence: "Negative NDVI values are typically associated with water, clouds, or snow, indicating the absence of spectrally visible vegetation." As a result, areas with intermittent standing water and scattered vegetation, such as wet tundra or regions near water bodies, are often inadequately represented in the NDVI analysis.
Later, in subsection 5.3 Study Limitations and Future Research Directions, we further elaborate on the limitations of the NDVI analysis.

'focuses' is more common but 'focusses' isn't wrong
We have rewritten the verb to its more common form.

Fig. 1; the ecoregions make sense but it wasn't fully clear how they were deliniated. Also, is spectral greeneness reflectance in the green or NDVI or a different metric? I just checked the table and it means mean monthly NDVI....it will help the reader to quickly explain what this means in figure legends because greenness can mean different things and can get confusing.
We described in subsection 3.3 that physio-geographic features such as adjacent seas, ocean currents, and ice caps, with direct and indirect control on heat and moisture transport were

considered on the delineation of the ecoregions. In this way, ecoregions were not susceptible to climate-sensitive metrics (e.g., summer averaged air temperature).

We agree with the editor and acknowledge that the brief description of spectral greenness in Table 1 is indeed incomplete. We have updated the text that now reads seasonally averaged monthly NDVI≥0.15, as described in subsection 3.1

'shrinkage' -> decline

Instead of "shrinkage," we have updated the definition in subsection 3.1, where "decline" represents a reduction in the spatio-temporal changes of greenness distribution instead of "shrinkange". In contrast, positive spatio-temporal changes in greenness distribution are defined as "expansion."